# An introduction of the Three-Dimensional Precipitation Particles Imager (3D-PPI)

Jiayi Shi[1], Xichuan Liu[1,2], Lei Liu[1,2], Liying Liu[3], Peng Wang[3]

[1]College of Meteorology and Oceanography, National University of Defense Technology, Changsha, China
[2]Key Laboratory of High Impact Weather (special), China Meteorological Administration, Changsha, China
[3]Aerospace NewSky Technology Co., Ltd, Wuxi, China

*Correspondence*: Xichuan Liu (liuxichuan17@nudt.edu.cn)

**Abstract.** A Three-Dimensional Precipitation Particles Imager (3D-PPI) is presented as a new instrument for measuring the size, fall velocity, and three-dimensional shape of the precipitation particles. The 3D-PPI consists of three high-resolution
cameras with telecentric lenses and one high-speed camera with one non-telecentric lens. The former records the high-resolution images of falling particles from three angles, based on which the three-dimensional shapes of particles can be restored by using a 3D reconstruction algorithm, and the observation volume is large enough to obtain the particle size distribution (PSD). The latter records the images of the falling precipitation particles with 200 frames per second, based on which the falling velocity of particles can be calculated. The field experiment of the 3D-PPI and OTT PARSIVEL disdrometer (OTT) was conducted at Tulihe, China, more than 880,000 snowflakes were recorded during a typical snowfall case lasting
for 13 hours, and the results show that the PSD obtained by the 3D-PPI and OTT have good agreement. It shows a potential application in atmospheric science, polar research, and other fields.

## 1 Introduction

Precipitation microphysics refers to the interactions and processes associated with the scale of individual precipitation
particles. Microphysics and microstructure, i.e. the distribution of particle properties such as size, shape, density, and mass, together determine the state and evolution of clouds and precipitation at this scale and are intrinsic to the process of cloud and precipitation evolution (Taylor, 1998). The microphysical properties of spherical or ellipsoidal raindrops have been relatively well studied. Research on ice-phase precipitation (such as snowflakes) microphysics is complicated by the complex geometry of individual snowflakes. Minor variations in the air temperature and humidity can cause significant changes in the shape of
ice crystals, resulting in a wide variety of crystal shapes ranging from simple plates and columns to complex dendrites or needles (Bailey and Hallett, 2012). Aggregation combines individual crystals into complex and random shapes of snowflakes. Despite the challenges associated with measuring snowflake shapes, the significance of this work is substantial. The accurate measurement of snowflake shapes is paramount for advancing our understanding of atmospheric sciences, including the formation of ice and mixed-phase clouds, as well as precipitation processes (Morrison et al., 2020; Pu et al., 2021).
Specifically, accurate snowflake shapes are critical for triple-frequency radar retrievals, as they directly influence the interpretation of radar echoes and the assessment of snow's microphysical properties (Mason et al., 2019). Besides, precise

shape descriptions of snowflakes will significantly improve the radar-based quantitative winter precipitation estimation (Notaroš et al., 2016).

The absence of accurate information on the 3D shape of precipitation particles leads to errors in the parameterization of physical processes in cloud precipitation and quantitative precipitation estimation (QPE) using weather radar. The assumption that true snowflakes are ellipsoidal may lead to inaccurate scattering matrix calculations, and hence incorrect determination of snow water equivalent (Tyynelä et al., 2011). The use of the ellipsoid approximation is only valid for smaller particles, and shape properties play an increasingly important role in scattering calculations as the snowflake scale increases (Olson et al., 2016). Between isovolumetric spheres and hexagonal columns, more accurate snowflake models are needed. In addition, an assessment of the sensitivity of high-frequency falling snow characteristics using an idealized simulated snowflake model indicates the necessity for a scattering database of snowflakes, in which three highly variable shapes should be taken into consideration (Kneifel et al., 2010). Ideal ice crystal models were created in the form of dendritic, thin plate, stellar plate, crown prism, and hollow column, and the scattering effects of these geometries were calculated using the Discrete Dipole Approximation (DDA) approach (Kim, 2006). The results indicate that the scattering characteristics of these ideal snowflakes are highly sensitive to their shapes. This further emphasizes the necessity for accurate shape modeling (Kim et al., 2007).

Different in-situ instruments were invented to measure the precipitation particles. The OTT PARSIVEL disdrometer (Loffler-Mang and Joss, 2000) can obtain the horizontal size (fall velocity) of particles according to the decrease (duration) of the laser signal by attenuation. However, the one-dimensional measurement concept requires additional assumptions to correctly size irregularly shaped particles such as snowflakes (Battaglia et al., 2010), the shape of particles cannot be obtained. The Two-Dimensional Video Disdrometer (2DVD) can obtain the 3D particle shape by using two line-scan cameras with an angle of 90°, the sampling area is 10×10 cm$^2$ (Bernauer et al., 2016). It should be noted that particle shape estimates may still be subject to bias due to horizontal winds (Helms et al., 2022). The Multi-angle Snowflake Imager (MSI) can obtain the three-dimensional shape and fall velocity of individual snowflakes by using four line-scan cameras with an angle of 45°, a limitation lies in its restricted sampling area, allowing the measurement of only one snowflake within a narrow field of view (Minda et al., 2017).

In addition to line-scan cameras, several planar camera instruments have been developed. The Snowfall Video Imager (SVI) employs a camera and a light source to record the images of snowflakes in free fall. Its subsequent evolution, the Precipitation Imaging Package (PIP), employs advanced digital image processing algorithms to enhance the precision and resolution of snowflake imaging. (Newman et al., 2009; Pettersen et al., 2020a). The Precipitation Imaging Package (PIP) provides physically consistent estimates of snowfall intensity and volume equivalent densities from high-speed images, although its equivalent density parameterization requires further refinement for extremely high snow-to-liquid ratio snowfall events (Pettersen et al., 2020b). The video precipitation sensor (VPS) can obtain the shape and fall velocity of hydrometeors when the particles fall through the sampling volume, the camera is exposed twice in a single frame, which allows the double exposure of particle images to be recorded, and the size and fall velocity of particles can obtained simultaneously (Liu et al., 2014; Liu et al., 2019). The Video In-situ Snowfall Sensor (VISSS) consists of two cameras with LED backlights and

telecentric lenses, it can measure the shape and size of snowflakes in a large observation volume with high pixel resolution (43 to 59 $\mu m \cdot px^{-1}$) (Maahn et al., 2024). Nevertheless, it is challenging to resolve highly fine structures of snowflakes with only two angles of observation.

The Multi-Angle Snowflake Camera (MASC) simultaneously captures three high-resolution images of falling hydrometeors from three different viewpoints (Garrett et al., 2012), which provides the conceptual possibility of 3D reconstruction of the observed snowflakes. The Visual Hull (VH) algorithm was used to reconstruct the 3D shape of snowflakes through multi-angle imaging, and the addition of two more cameras to MASC has been shown to improve reconstruction results (Kleinkort et al., 2017). Incorporating the rich dataset from the Multi-Angle Snowflake Camera (MASC), the 3D-GAN model is adeptly trained to reconstruct the intricate 3D architecture of snowflakes, thereby unlocking new dimensions in the

study of snowfall microphysics (Leinonen et al., 2021). Furthermore, the MASCDB, is a comprehensive database of images, descriptors, and microphysical properties of individual snowflakes in free fall, as presented by, showcases the MASC's exceptional potential for contributing to the field of atmospheric science by providing an extensive and detailed resource for studying the microphysical properties of snowflakes (Grazioli et al., 2022).

Currently, instruments are needed that not only provide a finer 3D structure of the snowflake but also capture enough

particles per unit time to estimate the PSD. This study presents a new instrument: the Three-Dimensional Precipitation Particle Imager (3D-PPI), the instrument design and main components are introduced in Sec.2, the calibrations of the camera and image binarization are described in Sec.3, and detailed algorithms including image processing, particle matching, particle localization, and 3D reconstruction are presented in Sec.4, the preliminary results of field experiment are discussed in Sec.5, The last part summarizes the main results and future work of 3D-PPI.

**2    Instrument design**

The 3D-PPI contains three identical high-resolution cameras with telecentric lenses (numbered Cam0, Cam1, and Cam2 in this paper) and one high-speed camera with a non-telecentric lens (numbered Cam3). Four high-brightness LED arrays are used as light sources to illuminate the observation volume. Additionally, a ZYNQ driver circuit has been developed to control the cameras, and light source, as well as to transmit the raw images to the PC terminal. To improve the instrument's working

efficiency, a capacitive rain sensor is adopted as a trigger, the cameras only work when the precipitation occurs. The sensor detects any moisture on the instrument's surface, and its heating element ensures that snow is melted and appropriately sensed. To mitigate the effects of wind on camera alignment, we have designed the 3D-PPI with a protective housing that effectively shields the cameras from wind disturbances. The concept drawing, prototype, and snowflake images of 3D-PPI are shown in Fig. 1.


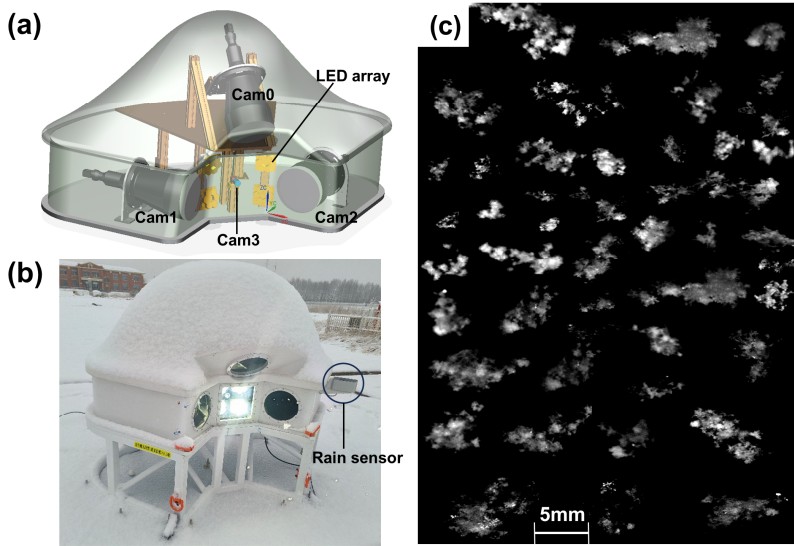

**Figure 1. (a)** Concept drawing of the 3D-PPI. **(b)** The prototype 3D-PPI deployed at Tulihe, China., (photo by J. Y. Shi). **(c)** Randomly selected snowflakes were observed on 6 April 2024 between 0453 UTC and 11:13 UTC.

The high-speed camera is positioned horizontally at the center, while the Cam1 and Cam2 are positioned, at 45 degrees, on either side of the high-speed camera in the same horizontal plane, and the Cam0 is positioned at 45 degrees vertically above (Fig. 1a). The high-brightness LED array light sources are situated on the same side as the cameras to illuminate the observation volume. The three telecentric lenses and LED lighting beams of 3D-PPI are illustrated in Fig. 2. To clarify, the three dimensions of observation volume (OV) of one high-resolution camera is a × b × d (170mm × 125mm × 88mm), which represent the length, width, and depth of field of view respectively. The intersection of three OVs of three high-resolution cameras forms the effective OV of 3D-PPI, the volume size is 775 cm$^3$. Only the particles falling within the effective OV of 3D-PPI can be simultaneously captured by the three high-resolution cameras, and then their 3D shapes can be 3D reconstructed.

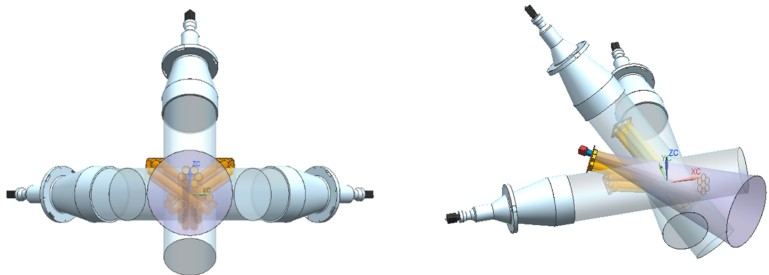

**Figure 2.** The two views of three telecentric lenses and LED light beams.

The high-resolution camera is a Sony IMX304 with a resolution of 4096 × 3000 gray-scale pixels and a frame rate of 5 fps. The high-speed camera is a Sony IMX287 with a resolution of 720 × 540 gray-scale pixels and a frame rate of 200 fps. The detailed technical parameters are shown in Table 1 and are divided into two parts, the image sensor and the lens.

**Table 1:** Technical specifications of the cameras.

|  |  | High-resolution camera | High-speed camera |
|---|---|---|---|
| Image sensor | Type | CMOS,  Global shutter | CMOS,  Global shutter |
|  | Model | Sony IMX304 | Sony IMX287 |
|  | Single pixel size [μm] | 3.45 × 3.45 | 6.9 × 6.9 |
|  | Resolution [px] | 4096×3000 | 720×540 |
|  | Pixel resolution [μm·px$^{-1}$] | 41.6 | 265.4 (450mm distance) |
|  | Size of the field of view (a × b) [mm] | 170 × 125 | 191 × 143 (450mm distance) |
|  | Frame rate [fps] | 5 | 200 |
|  | Effective exposure time [μs] | 20 | 20 |
| Lens | Type | Telecentric lenses | 25mm non-telecentric lenes |
|  | Aperture | F6.5 | F2.4 to F16 |
|  | Magnification of lens | 0.083 (constant) | 0.026 (450mm distance) |
|  | Distortion | 0.044% | 0.16% |

For the high-resolution camera, the single pixel size is 3.45 μm, and the magnification of the lens is 0.083, meaning that the pixel resolution is 41.2 μm·px$^{-1}$. Telecentric lens distortion is 0.044% and allowed to be ignored. For the high-speed camera, the single pixel size is 3.45 μm, and the magnification is 0.026 at a working distance of 450mm, meaning that the pixel resolution is 265.4 μm·px$^{-1}$.

The utilization of telecentric lenses eliminates the sizing error caused by the uncertain distance between the snowflakes and the cameras. Unlike non-telecentric lenses, which produce larger images of nearby objects and smaller images of distant objects (Fig. 3a), telecentric lenses are based on the principle of parallel light imaging, resulting in identical objects at different distances from the lens having the same size in the image (Fig. 3, c)., This difference in imaging characteristics will lead to distinct methods for three-dimensional reconstruction (Fig. 3b, d), which will be discussed in detail in Sec. 4. With an optical distortion of 0.044%, the telecentric lens effectively minimizes distortion, establishing a linear correspondence between image coordinates and world coordinates. This alignment greatly simplifies camera calibration, it will be elaborated upon in Sec. 3.

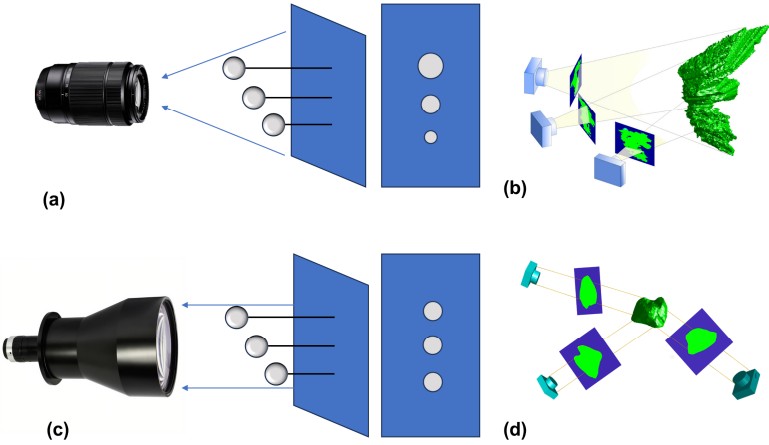

**Figure 3.** Illustration of imaging characteristics and reconstruction of a snowflake using three cameras. **(a)** non-telecentric lens imaging characteristics; **(b)** non-telecentric lenses back-projected to obtain three visual cones (Kleinkort et al., 2017); **(c)** telecentric lens imaging characteristics; **(d)** telecentric lenses back-projection to three visual columns.

The 3D-PPI has 4 light source arrays, each consisting of 6 high brightness LEDs organized into a cluster to create an array lighting system. Each LED is 5W with a total power of 120W. This LED lighting system integrates a high-brightness LED chip, substrate, heat sink, casing, leads, protective features, and other functionalities into a single unit. At its core lies the LED chip, utilizing white high-brightness LED chips. Each LED is equipped with a converging lens, facilitating the creation of a cone-shaped beam of light. Once triggered, the LEDs will continue to illuminate, providing consistent lighting throughout

the exposure period. The design and physical representation of the LED array lighting system is illustrated in Fig. 4. The LED light sources are configured in parallel, allowing for a single power supply connection that distributes power to the entire array. Each LED can operate in either constant current mode or trigger mode. In constant current mode, the LEDs provide stable and uniform light intensity, which enhances the uniformity of illumination within the OVs.

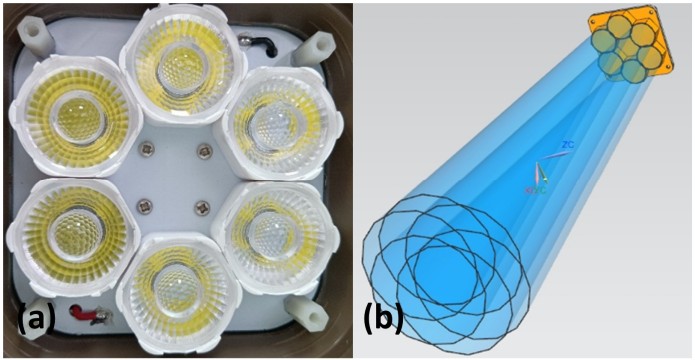

**Figure 4. (a)** LED array lighting group design; **(b)** Cone-shaped beam formed by LED array lighting group.

# 3    Calibration

## 3.1 Calibration of cameras

Camera calibration is the basis for obtaining three-dimensional (3D) spatial information from two-dimensional (2D) images. There is a one-to-one correspondence between the spatial scene points and their image points in the image, and their positional relationship is determined by the camera imaging geometric model (Cheng and Huang, 2023). The parameters of this geometric model are called camera parameters, which can be determined by experiment and computation, and the process of solving the parameters experimentally and computationally is called camera calibration. The geometric model for telecentric lens imaging is described in detail in Appendix A, where four coordinate systems (WCS, CCS, ICS, and PCS) are defined and their transformations are derived. The purpose of camera calibration in this section is to estimate the projection matrix $KM_0$, $KM_1$, and $KM_2$ for three high-resolution cameras (Cam0, Cam1, and Cam2 respectively), which enables the transformation of 3D spatial points in the WCS to their corresponding pixel coordinates on the image plane in PCS. Appendix A derives a model that assumes linear, telecentric imaging without distortion. Based on this model, Eq. (A.6) is transformed to:

$$\begin{bmatrix} \dfrac{\beta}{S_v}r_{11} & \dfrac{\beta}{S_v}r_{12} & \dfrac{\beta}{S_u}r_{13} & \dfrac{\beta}{S_u}t_x+u_0 \\ \dfrac{\beta}{S_v}r_{21} & \dfrac{\beta}{S_v}r_{22} & \dfrac{\beta}{S_v}r_{23} & \dfrac{\beta}{S_u}t_y+v_0 \end{bmatrix} \begin{bmatrix} X_w \\ Y_w \\ Z_w \\ 1 \end{bmatrix} = \begin{bmatrix} u \\ v \end{bmatrix} \tag{1}$$

Where $\beta$ is the telecentric camera magnification; $S_u$ and $S_v$ are the length and width dimensions of the individual image; $r_{ij}$, $t_x$ ($t_y$) denote the matrix elements of the rotation and translation process from the WCS to the CCS respectively, and these parameters relate solely to the relative position of the cameras; and $u_0$ and $v_0$ are the horizontal and vertical coordinates of the camera's main point offsets, which may change over time.

The coordinate points ($X_w$, $Y_w$, $Z_w$) in WCS are projected to the coordinate points ($u$, $v$) in the PCS through the 2 × 4 matrix (also denoted as $KM$) on the left side of Eq. (1). In  particular, the three telecentric lenses are in the same WCS, and each lens corresponds to a PCS.

To obtain the projection matrices, the following steps are executed:  Firstly, make a 3D checkerboard and establish the WCS. Attach three high-precision 2D checkerboards to three mutually perpendicular flat boards to form a 3D checkerboard. The three plane intersections are used as the WCS origin O, and the two plane intersections are used as the X, Y, and Z axes respectively (Fig. 5b). The 3D checkerboard is placed in a position that defines the WCS. This position is within the effective OV to ensure that three high-resolution cameras can capture checkerboard corner points simultaneously. Secondly, physically measure the precise coordinates of all checkerboard corner points in the WCS ($X_{wj}$, $Y_{wj}$, $Z_{wj}$) ($j$ denotes the index of corner points), and identify the corresponding pixel coordinates ($u_j$, $v_j$) of $j^{th}$ corner points in the PCS of each camera image. Thirdly, substituting these coordinates into Eq. (1) yields Eq. (2) for each camera image:

$$KM \cdot W_J = C_J \quad \Leftrightarrow \quad KM \cdot \begin{bmatrix} X_{w1} & X_{w2} & \cdots & X_{wJ} \\ Y_{w1} & Y_{w2} & \cdots & Y_{wJ} \\ Z_{w1} & Z_{w2} & \cdots & Z_{wJ} \\ 1 & 1 & \cdots & 1 \end{bmatrix} = \begin{bmatrix} u_1 & u_2 & \cdots & u_J \\ v_1 & v_2 & \cdots & v_J \end{bmatrix} \tag{2}$$

Where the $J$ is the number of chosen points for each camera image. Its value is determined based on the number of corner points of the image captured by each camera, and may be different for each camera image; $W_J$ is a matrix of 4 rows and $J$ columns denoting the $J$ coordinates of the points in the WCS; $C_J$ is a matrix of 2 rows and $J$ columns denoting the $J$ coordinates of the points in the PCS.

As can be seen from Fig. 5b, the value of $J$ is much larger than 4, so Eq. (2) is equivalent to overdetermined linear systems. Further, the least squares method is used to optimally estimate the projection matrices ($KM$) for each high-resolution camera. It is worth noting that during the solution (optimal estimation) it is important to make sure that no row of $W_J$ (except the row with value 1) can be the same value. In other words, the selected points cannot be in the same plane in the WCS. Therefore, the reason for selecting a 3D checkerboard rather than a 2D checkerboard becomes evident.

To assess the accuracy of the estimated projection matrices ($KM$) of three cameras, we calculated the average reprojection error, which is calculated as the mean distance between the identified 2D image points ($u_j$, $v_j$) and their corresponding projected points $KM \cdot (X_{wj}, Y_{wj}, Z_{wj})^T$. The results show that the average reprojection error for three high-resolution cameras is 0.32 pixels.

However, even small movements of the high-resolution camera can alter the projection matrix. This requires the instrument to be more robust. To mitigate the effects of wind on camera alignment, the instrument housing has been specifically designed for stability. Additionally, a semiconductor air conditioner has been installed in the housing, which will prevent minor camera expansion caused by temperature fluctuations.

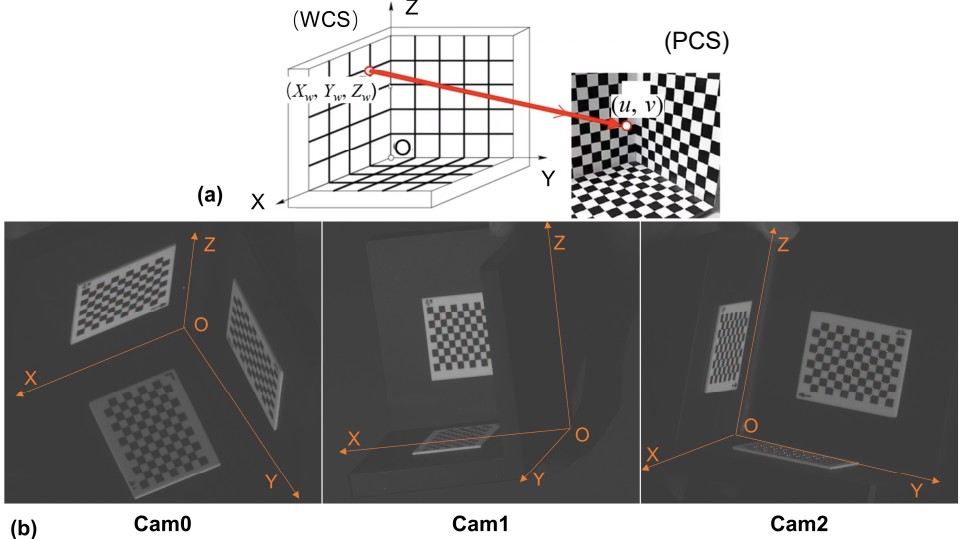

**Figure 5. (a)** 3D checkerboard calibration principle; **(b)** WCS with three camera views.

### 3.2 Estimation of pixel resolution

The purpose of this section is to experimentally determine the accurate pixel resolution of each high-resolution camera. The process utilizes 15 spheres with absolute sphericity ranging from 1 to 25 millimeters in diameter, which are dropped into the OV of 3D-PPI (Fig, 6a). The materials of the spheres have a similar refractive index to the snowflakes. Theoretically, any spherical object captured in an image appears as a perfect circle, which is an advantage of using spheres. The ratio of the actual diameter of the sphere ($D_a$) to its diameter in pixels as represented in the image ($D_p$) constitutes the pixel resolution. The $D_a$ is

measured using a caliper, with units in millimeters, and the $D_p$ is counted manually, with units in pixels. The scatterplot about the $D_a$ and corresponding $D_p$ for 15 ceramic spheres and the linear least-squares fit straight line plotted together (Fig, 6b), resulting in:

$$D_a\,[\mathrm{mm}] = (0.0416 \pm 0.0001) \cdot D_p\,[\mathrm{px}] + 0.0259 \tag{3}$$

for Cam0,

$$D_a\,[\mathrm{mm}] = (0.0410 \pm 0.0003) \cdot D_p\,[\mathrm{px}] + 0.1647 \tag{4}$$

for Cam1, and

$$D_a\,[\mathrm{mm}] = (0.0410 \pm 0.0003) \cdot D_p\,[\mathrm{px}] + 0.0835 \tag{5}$$

for Cam2.

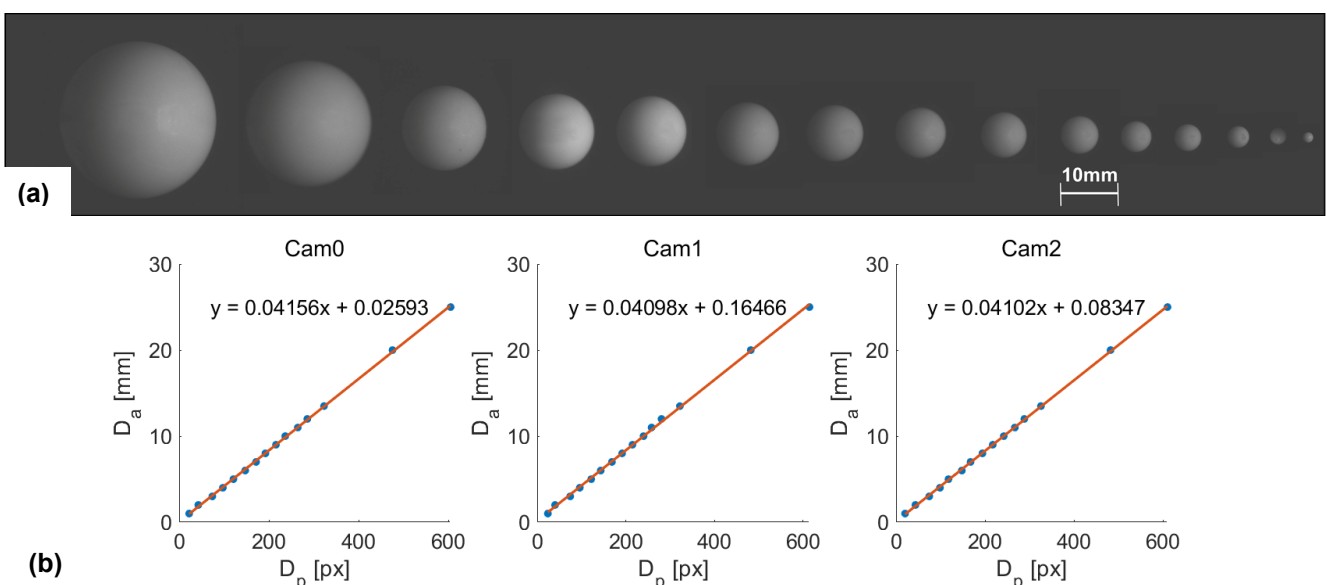

**Figure 6. (a)** The 15 reference spheres ranging from 1 to 25 millimeters in diameter captured by the high-resolution camera; **(b)** The scatterplot about the $D_a$ and corresponding $D_p$ for 15 ceramic spheres and the linear least-squares fit straight line. The results of the linear least squares fit are also shown in the legends.

The reciprocals of the slopes of the three fitted lines are 0.04156, 0.04098, 0.04102 mm·px$^{-1}$, and closely align with the 0.0416 mm·px$^{-1}$ specification from the manufacturer of the high-resolution cameras. The non-zero intercepts observed in the linear fits, ranging from about 0.02 px to 0.16 px, can be attributed to several factors, including systematic errors introduced during the calibration process due to lens distortion or misalignment, as well as image processing artifacts from the binarization method that may affect edge detection.

### 3.3 Calibration of image binarization

The image captured by the high-resolution camera is binarized through adaptive thresholding. Adaptive thresholding is a widely used technique for image binarization, where the threshold for converting pixel values to binary is determined based on local characteristics of the image rather than a single global threshold. This method enhances the accuracy of foreground particle detection, particularly in images with complex backgrounds and uneven illumination. The adaptive threshold at a pixel location $(u,v)$, denoted as $T(u,v)$, represents the threshold value used to classify the pixel at position $(u,v)$ into either foreground or background. It is computed using the local mean $\mu(u,v)$ from a specified neighborhood around that pixel, adjusted by a sensitivity coefficient $C$:

$$T(u,v) = \mu(u,v) - C \tag{6}$$

Where the sensitivity coefficient $C$, typically constrained between 0 and 1, plays a crucial role in modulating the threshold. A smaller $C$ value treats relatively lower brightness pixels as foreground, while a larger $C$ value causes more pixels to be classified as background. Therefore, a well-determined $C$ can enhance binarization performance by effectively distinguishing between foreground objects and background noise, thereby improving the segmentation results.

Calibration of image binarization is aimed at determining the optimal value of $C$. Firstly, drop the spheres of each diameter mentioned in Sec. 3.2 times into the observation volume, and acquire 20 clear images for each diameter (similar to Fig. 6a). Secondly, use the adaptive thresholding algorithm to convert the sphere image into the binarized image, and then calculate $D_{max}$ and $D_{eq}$ (in the image, the $D_{max}$ is the distance between the two farthest points of the particle profile, and the $D_{eq}$ is the diameter of the circle equal to the area of the particle profile. The $D_{max}$ and $D_{eq}$ are equal only when the image of the sphere is perfectly circular). Thirdly, convert $D_{max}$ and $D_{eq}$ from pixels to millimeters through pixel resolution calculated in Sec. 3.2 and calculate the average value of $D_{max}$ and $D_{eq}$ for 20 times for each diameter. Ultimately, calculate the mean relative error (MRE) for different sensitivity coefficients $C$ using Eq. (7). And the results in Fig.7a show that a $C$ value of 0.4 resulted in the smallest MRE, indicating that the optimal value for $C$ is 0.4.

$$\text{MRE} = \frac{1}{2}\left(\frac{1}{n}\sum_{i=1}^{n}\left|\frac{D_{ai} - \bar{D}_{maxi}}{D_{ai}}\right| + \frac{1}{n}\sum_{i=1}^{n}\left|\frac{D_{ai} - \bar{D}_{eqi}}{D_{ai}}\right|\right) \tag{7}$$

Where n denotes the number of spheres of different diameters, which is 15; $D_{ai}$ denotes the $i^{th}$ diameter of the sphere; $\bar{D}_{maxi}$ and $\bar{D}_{eqi}$ denote the average value of $D_{max}$ and $D_{eq}$ for 20 times for $i^{th}$ diameter respectively.

With $C$ set to 0.4, the 15 spheres in Fig. 6a were effectively binarized, as shown in Fig. 7b. Regarding the $D_{max}$ measurement results (Fig. 7c, e), smaller spheres (9 mm and below) tend to show that the measurements are slightly greater than the true values, while larger particles exhibit that the measurements are slightly lower than the true values. The maximum relative error is about 14%, and the MRE of $D_{max}$ is about 4%. As for $D_{eq}$ measurement results (Fig. 7d, f), almost all diameter measurements underestimate the true values. The maximum relative error is about -7% and the MRE of $D_{eq}$ is about 3%. Since

the measurement errors of $D_{eq}$ for all spheres are lower than the true values, they can be utilized for systematic error correction.

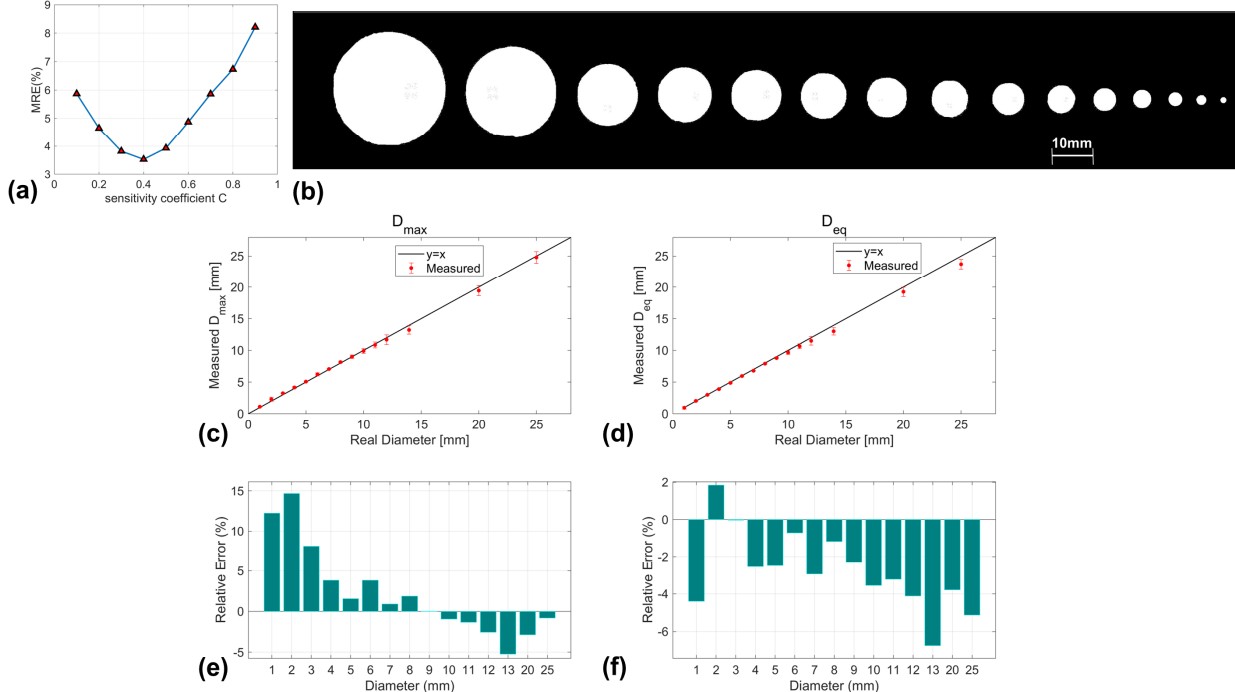

**Figure 7 (a)** Variation of MRE with sensitivity coefficient $C$; **(b)** The binarized images of 15 spheres; The average values of measurements of $D_{max}$; **(c) - (f)** The average values and relative errors of 20 $D_{max}$ and $D_{eq}$ measurements for per diameter sphere.

## 4    Algorithms

### 4.1  Image processing

        For High-resolution camera images with the resolution of 4096×3000, no background removal and denoising are required due to low image noise and little background interference. The processing steps are as follows: (i) Image binarization. The image is binarized through adaptive thresholding mentioned in Sec.3.3. (ii) Particle detection. Firstly, detect the connected regions in binarized images. Secondly, combine regions into a single particle when the distance between the closest points of

connected regions in a single image is detected to be less than 2 mm apart. This step is necessary because a single particle may sometimes be perceived as two separate particles due to its position near the edge of the image processing threshold. Thirdly,

discard the particles with an area smaller than 20 pixels (Equivalent to 0.035 mm², $D_{max}$ is about 0.2mm), which enables the removal of pixel noise (no larger than 20 pixels) from the image, to prevent these noises from being mistakenly detected as small snowflakes. Fourthly, discard the particles at the edge of the image. If the connected region of a particle contains points located at the edges of the image, the particle is considered not to be fully captured, and it should be discarded.

For the high-speed camera images with a resolution of 720×540, there is non-negligible noise and background interference, therefore, two more steps are required before image binarization and particle detection: (i) Background removal and enhancement. Background artifacts captured by high-speed cameras in natural settings can be influenced by varying lighting conditions, lens surface contamination, or other factors that change over time. To address this issue, a real-time background detection method is employed. Specifically, 1024 images are randomly selected every minute to calculate the average grayscale value, representing the minute-by-minute background changes. These background variations are then subtracted from all images taken within that minute to effectively remove the background interference. It is further necessary to enhance the contrast of the image by stretching the grey scale histogram to better distinguish between background and foreground particles. (ii) Image denoising. The median filtering is used to remove the remaining. Further, the image binarization and particle detection methods are the same as the previous high-resolution camera image processing methods.

The two types of image processing processes and the results of each step are shown in Fig. 8.

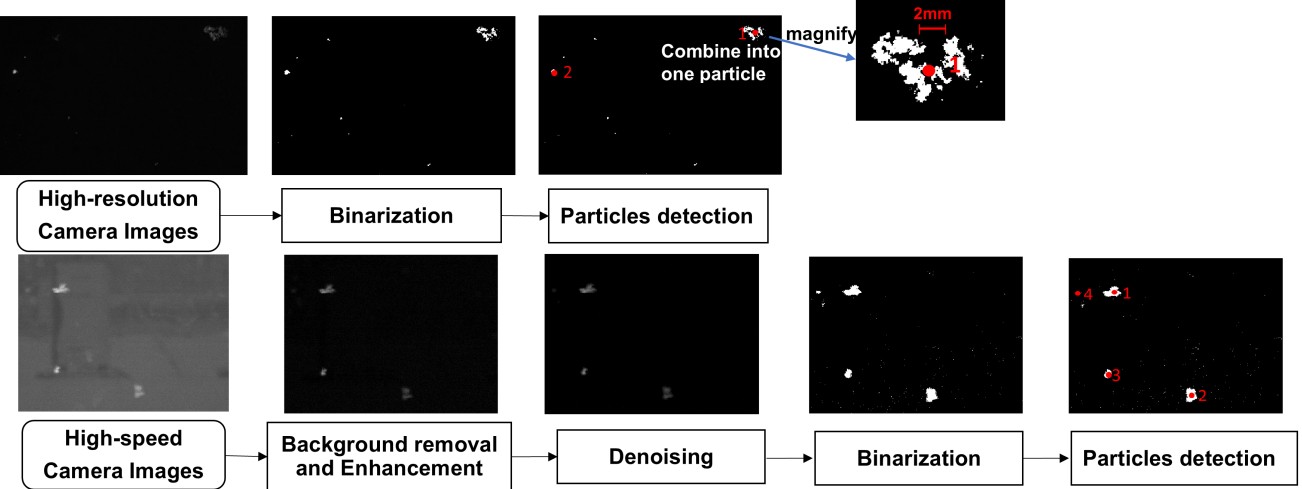

**Figure 8** Flowchart of two types of image processing.

## 4.2 Particle matching and localization

### 4.2.1 Particle matching

In the observation volume of 3D-PPI, there might be numbers of particles with similar sizes, colors, shapes, and textures, which poses a challenge for particle matching from the images captured by three cameras. In this work, we propose a particle matching algorithm that addresses this issue by focusing on the spatial positions of particles in three images, as derived from

the projection matrices obtained through precise calibration of the trinocular telecentric camera system. For one particle, the matching algorithm is implemented in detail as follows (Fig. 9):

(i) Detect the centroid coordinates of the particle P $(u, v)$ in the PCS from Cam0 (Fig. 10a).

(ii) Using the projection matrix $KM_0$ of Cam0, the underdetermined linear equations corresponding to P are solved to obtain a straight line L in WCS. Specifically, it is equivalent to solving $W_1$ in Eq. (2) with $KM$ and $C_1$ known, where the solution to $W_1$ is not unique and all solutions are the L in the WCS. The L represents all points in 3D space that can be projected onto P by $KM_0$, in other words, the line L is the back-projection of the point P in WCS.

(iii) Project the L onto the planes of Cam1 and Cam2 by multiplying the projection matrices $KM_1$ and $KM_2$, respectively, with L, resulting in the line segments on each of the image planes of Cam1 and Cam2 (Fig. 10b, c). The exact derivation of the formulae in (ii) and (iii) are described in detail in Appendix B.

(iv) Detect the particles that the line segments pass through on Cam1 and Cam2 respectively. If the line segments do not pass through any particles in Cam1 or Cam2, it is a failed matching, meaning that the particle does not appear in the effective OV.

By performing the above matching for each particle detected by cam0, the location of this particle in Cam1 and Cam2 can be found. Fig. 10 shows the three particles detected in Cam0 and the matching of each particle in Cam1 and Cam2.

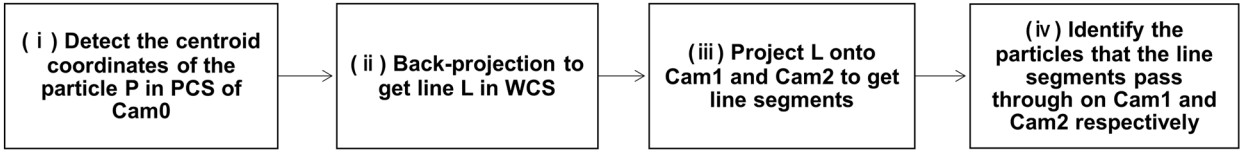

**Figure 9.** Flowchart of a particle matching

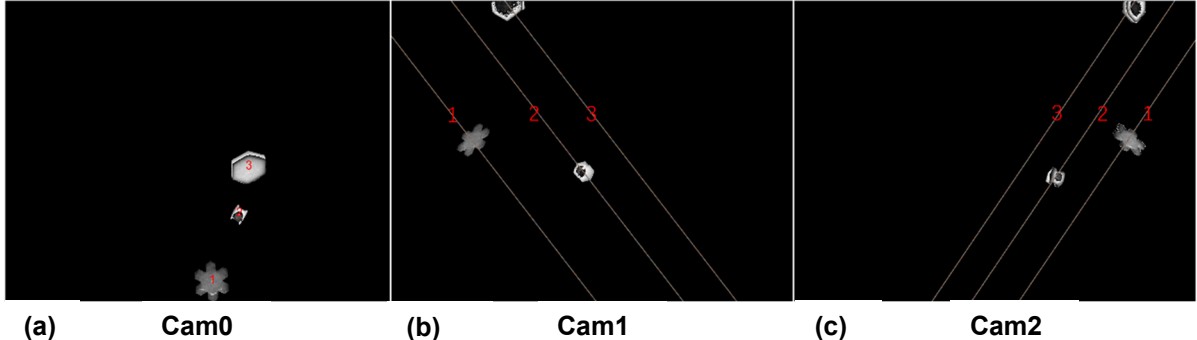

**Figure 10.** The three particles detected in Cam0 can only appear on the corresponding three line segments in Cam1 and Cam2 respectively

4.2.2 Particle localization

After completing camera calibration and particle matching, 3D reconstruction for each particle needs to be performed. However, since each particle only occupies a small space in the effective OV, we propose a method that involves preliminarily locating particles in WCS before proceeding with subsequent 3D reconstruction. This method leverages the positions of single

particles in three images to identify the minimal cuboid capable of containing the particles, thereby accurately pinpointing the particles' localizations.

For a single particle, the pixel coordinates of the centroid point of the particle in Cam0, Cam1, and Cam2, respectively, have been detected and the subsequent 3D spatial localization steps in the WCS are as follows (Fig. 11):

(i) Find the back-projection line $L_1$ of point $P_1$ by $KM_0$. The underdetermined linear equation corresponding to $P_1$ is solved to obtain a straight line $L_1$ in WCS. This implementation principle is similar to the second step of particle matching mentioned above. Eq. (B.1) and Eq. (B.2) in Appendix B explain this process.

(ii) The lines $L_1$ are projected onto the planes of Cam1 by multiplying the projection matrices $KM_1$, resulting in line segment $L_2$, which is represented as a 2-row by 1-column matrix. This corresponds to the segment of the $L_{p1}$ within the image boundaries as described in Eq. (B.3).

(iii) Find the point $P_2'$ on $L_2$ that is closest to $P_2$. Due to the irregular shape of the particle, $P_2'$ does not necessarily coincide with $P_2$.

(iv) Follow the same approach as in step 1, determine the back-projection line $L_3$ of point $P_2'$ by $KM_1$.

(v) Localize the 3D coordinates of the intersection of $L_1$ and $L_3$ in the WCS, that is $P_c$, which is the centroid of the target cuboid, and further determine the side lengths of the cuboid. From the previous steps, $L_1$ and $L_3$ are destined to intersect in the WCS, and the intersection point is regarded as the centroid of the rectangle, whose side lengths can be determined by converting from the pixel dimensions in the particle image to the actual physical dimensions in the WCS.

(vi) Finally, verify that the projection of the $P_c$ point through $KM_2$ in Cam2 is near the $P_3$ point and within the particle contour, otherwise, it is a failed localization.

The particle's position in the WCS should be inside the region of the cuboid determined by localization, which will next be discretized into numerous smaller voxel grids to finely perform the 3D reconstruction.

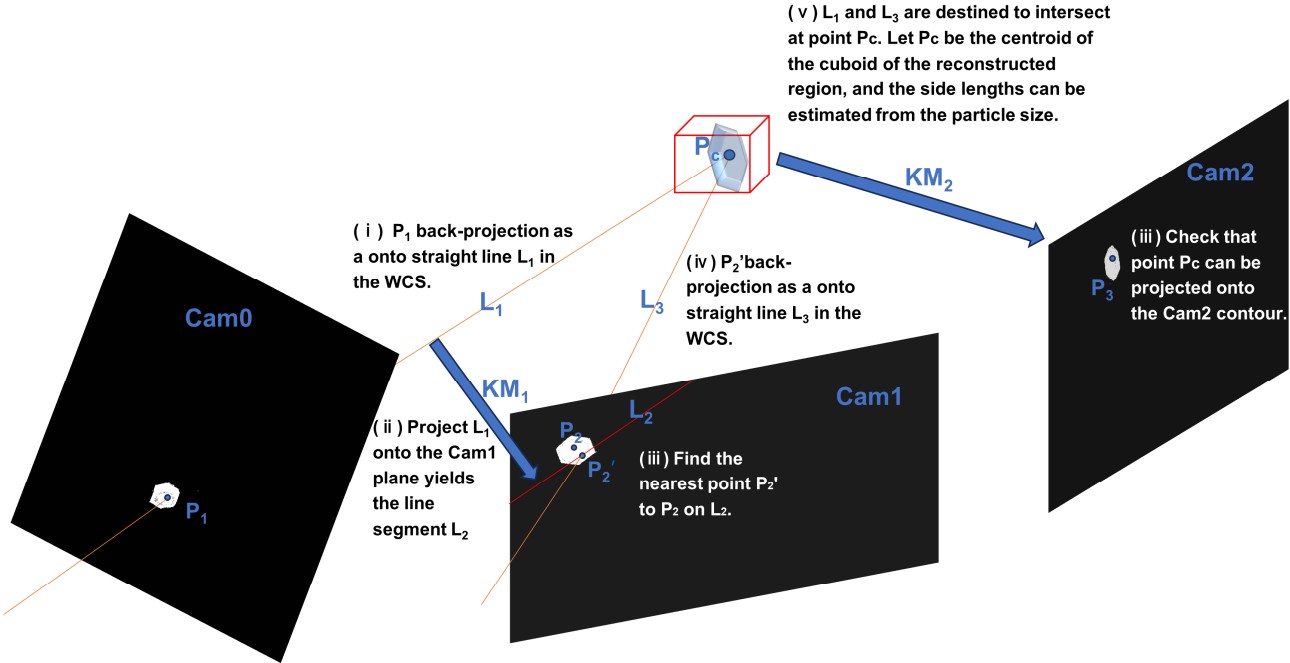

**Figure 11.** Steps to localize the particle position in the WCS according to the image particle contours

## 4.3 Three-dimension (3D) reconstruction

The Visual Hull (VH) method is a technique used to reconstruct the three-dimensional shape of snowflakes by utilizing silhouettes, which are the outlines of the snowflakes as seen from different camera angles. In this process, multiple cameras are positioned around the snowflake at various viewpoints (Cam0, Cam1, and Cam2 in 3D-PPI). Each camera captures the silhouette of the snowflake, and these silhouettes are carefully calibrated to ensure that they accurately represent the snowflake's geometry. A cone of silhouette is created by back-projecting the set of points of view of the previously detected silhouettes into the corresponding image planes in front of the cameras (Fig. 3b), and the intersection of these cones gives the visual hull (Hauswiesner et al., 2013). Since concave features do not affect the silhouette obtained from each image, a limitation of the visual hull method is its inability to capture concave features. Based on high-resolution contour images captured by three high-resolution cameras of 3D-PPI, and the projection matrix for each camera, we propose to apply the visual hull method to reconstruct the 3D shapes of snowflakes. The use of telecentric cameras allows the visual solid cones formed by back-projection to become visual solid columns (Fig. 3b, d).

The algorithm operates as follows: given a multi-viewpoint contour image and projection matrices, it ascertains whether the pixel or voxel corresponding to a spatial point on each contour map is part of the object's contour. The resulting model represents a sample of the smallest convex set that encloses the object's true shape, precluding the depiction of indentations.

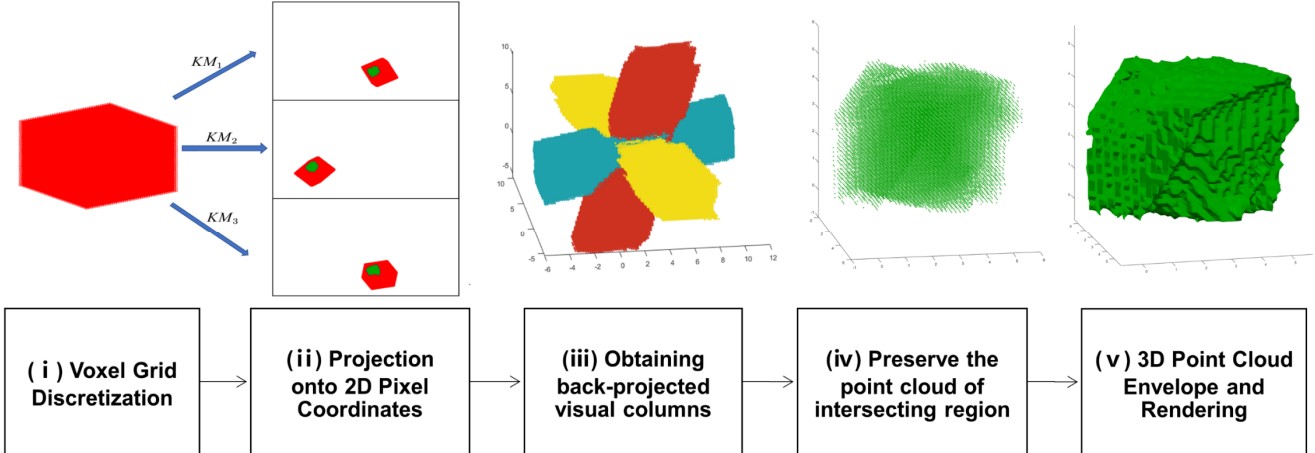

**Figure 12.** Flowchart of 3D construction algorithm

Initially, we employ the preliminary particle localization method described in Sec. 4 to estimate the particle's approximate spatial position. The further procedure for obtaining a 3D point cloud and reconstructing the 3D model of precipitation particles is outlined through the following refined steps (Fig. 12): (i) Voxel Grid Discretization: Subdivide the space into a voxel grid with a predefined resolution. For each voxel, extract the 3D coordinates of the upper left corner point. This grid will serve as a framework for subsequent steps. (ii) Projection onto 2D Pixel Coordinates: Utilize the projection matrix to project the 3D voxel coordinates of each voxel onto the three-angle contour, transforming them into 2D-pixel point coordinates. (iii) Obtaining back-projected visual columns (by three sets of point clouds): Mark the 3D voxel coordinates of points that can be projected onto the contours of each of the three images, that is, obtain three contours of back-projected visual columns. (iv) Preserve the point cloud of the intersecting region: Retain a point cloud within the convergence region of the three optical axes, identifying the spatial locus where the 3D reconstructed object is situated. (v) 3D Point Cloud Envelope and Rendering: Apply the triangular sectioning algorithm to extract the visual envelope of the 3D point cloud. Subsequent rendering steps will then be used to construct the 3D reconstruction model of the precipitation particle.

## 5   Preliminary results of field experiment

### 5.1 Case studies of snowfall case

To evaluate the performance of the 3D-PPI, the prototype of 3D-PPI was deployed at Tulihe, China (50.692°N, 121.652°E; 733ma.m.s.l.) from January 1st, 2024, and an OTT PARSIVEL laser disdrometer (OTT for short) was installed 10 meters apart for comparison. According to the conclusion of wind field simulation in Appendix C, the 3D-PPI is installed facing towards the south. A typical snowfall case lasting for 13 hours, from 1900 UTC on March 28th, 2024 to 0759 UTC on March 29th, 2024 was observed and analyzed.

The PSD calculated from OTT counting is as follows (Zhang et al., 2019):

$$\text{PSD}(D_i) = \frac{1}{S \cdot T \cdot \Delta D_i} \sum_{j=1}^{32} \frac{n_{ij}}{V_j} \tag{8}$$

Where PSD $(D_i)$ (mm$^{-1}$·m$^{-3}$) is the number concentration of particles per unit volume per unit size interval $\Delta D_i$ for snowflake size $D_i$ (mm); $n_{ij}$ is the number of snowflakes within size bin $i$ and velocity bin $j$; $T$ (s) is the sampling time (60 s in this study), and $V_j$ (m/s) is the falling speed for velocity bin $j$; $S$ (m²) is the effective sampling area (0.18 m×0.03 m).

The time-averaged PSD calculated from 3D-PPI counting over a specified period is as follows:

$$\text{PSD}(D_i) = \frac{N_i}{\triangle D_i \cdot N_{\text{ima}} \cdot V_i} \tag{9}$$

Where $N_i$ is the number of particles in the $i$th size bin; $N_{\text{ima}}$ is the number of acquired images over a period of time; The size descriptor $D$ for 3D-PPI is $D_{\max}$ or $D_{\text{eq}}$ in this paper; $V_i$ (m³) is the valid OV of the Cam0 after edge correction. Since we discard particles at the edges of the image in Sec. 4.1, $V_i$ is a function of $D_i$, shown in Eq. (10). The $a$, $b$, and $d$ represent the length (0.17m), width (0.125m), and depth (0.088m) of the field of view respectively.

$$V_i = (a - 2D_i) \cdot (b - 2D_i) \cdot d \tag{10}$$

Considering that the sampling rate of a high-resolution camera is 5 fps, and the time for a snowflake to pass through the field of view is less than 0.2s, the probability of capturing the same snowflake in two consecutive frames is very low.

During the snowfall case, three high-resolution cameras of 3D-PPI recorded 552383 and 328792 snowflakes over two days. The time-averaged PSD in the same period obtained by 3D-PPI (using two types of size descriptors) and OTT are compared (Fig. 13a, b). The PSDs measured by OTT and 3D-PPI using $D_{\text{eq}}$ as a size descriptor are highly consistent, however, they deviate significantly from those using $D_{\max}$ as a size descriptor. The PSDs of particles with a $D_{\text{eq}}$ of about 0.4 mm were highest for both chosen days.

The trends of the number density of particles observed by the two instruments were similar, the correlation coefficients are 0.94 and 0.96 for the two days. In comparing the temporal plots (Fig. 13c, d, e, f), certain periods (19:00 to 19:50, 20:50 to 22:00, and after 23:30 UTC on March 28; 01:30 to 04:30, and 06:00 to 07:59 on March 29) exhibited a smaller number of particles per unit volume, with larger average sizes and greater difference between $D_{\text{eq}}$ and $D_{\max}$. This indicates a higher degree of aggregation and potentially more complex shapes of individual snowflakes during these times. Conversely, other periods (19:50 to 20:50 and 22:00 to 23:30 UTC on March 28; 00:00 to 01:30, and 04:30 to 06:00 on March 29) showed a larger number of particles per unit volume, smaller average sizes, and reduced difference between $D_{\text{eq}}$ and $D_{\max}$.

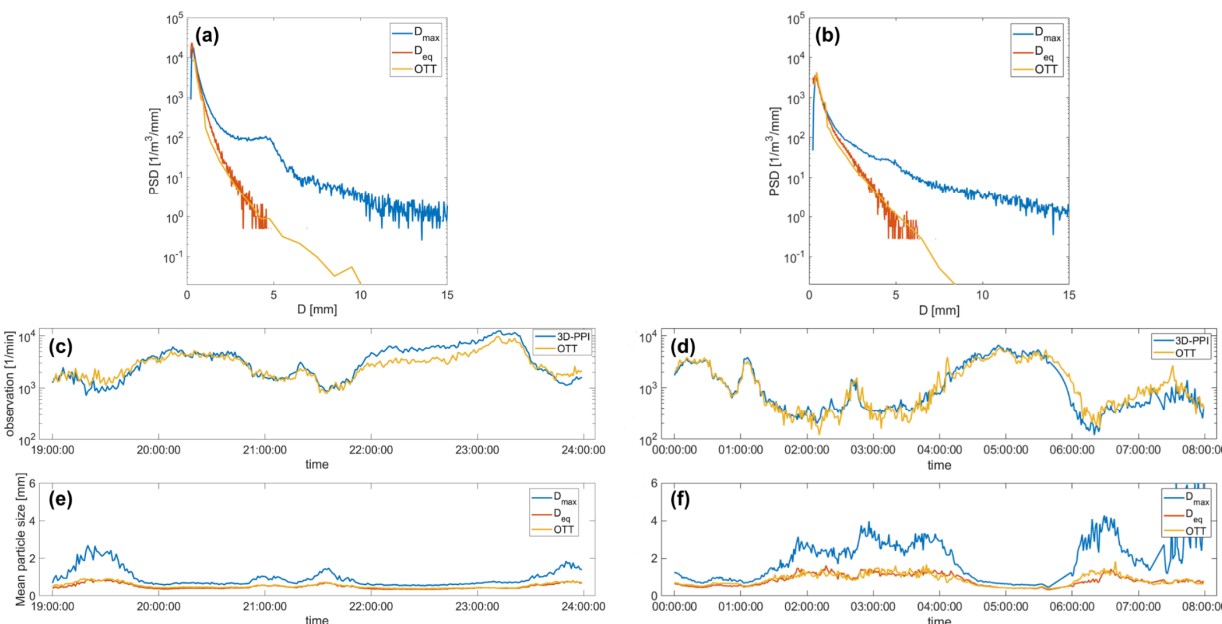

**Figure 13.** This typical continuous snowfall was split into two days and plotted separately (left and right). The 1-minute particle size distribution (PSD) of the $D_{max}$ (blue) and $D_{eq}$ (orange) for 3D-PPI and OTT (yellow) for the snowfall case (first row). Temporal plot of average particle counts per unit volume per minute over two days (second row). Temporal plot of average particle size per minute over time (third row).

## 5.2 Three-dimensional shape of snowflakes

Fig. 14 shows the reconstructed 3D shapes of 6 snowflakes collected on 6 April 2024. For each snowflake, three images were obtained from three high-resolution cameras (Cam0, Cam1, and Cam2) and the results of the 3D shape were reconstructed by utilizing the visual hull method (Fig. 14). To characterize the 3D shape of each snowflake, four parameters are calculated: volume $V$, maximum size $D_{max}$ (diameter of the smallest enclosing circle), aspect ratio $AR$ (ratio of the longest and shortest axes of the smallest outer ellipsoid), sphericity ($Sp$). $Sp$ is derived from the $V$ and $S$ (surface area of 3D reconstructed hull) and characterizes the degree to which 3D particles approach the sphere:

$$Sp = \frac{4\pi(\frac{3V}{4\pi})^{\frac{2}{3}}}{S} \tag{11}$$

The 3D shapes of snowflakes ranging in volume from over 400 mm³ (Fig. 14b) to as small as less than 20 mm³ (Fig. 14f) all can be reconstructed. In the algorithm when two connected regions are close together, they are considered as the same snowflake, so the reconstructed snowflake will appear as a separated small part that is not connected to the main body, in which case $D_{max}$ is meaningless (Fig. 14 e, f). From the results, it can be found that the visual hull approach can effectively and precisely execute the 3D reconstruction for snowflakes with highly realistic, intricate, and varied shapes and compositions, as well as diverse sizes and sphericity. The analysis of individual snowflake cases is meaningless, and here is just to show that

3D-PPI already has the capability of 3D reconstruction. Therefore, the next step is to embed the 3D reconstruction and pre-algorithms into the instrument to realize real-time, automated, and batch 3D reconstruction of snowflakes, to statistically characterize the distribution of the 3D shape of a large number of snowflakes.

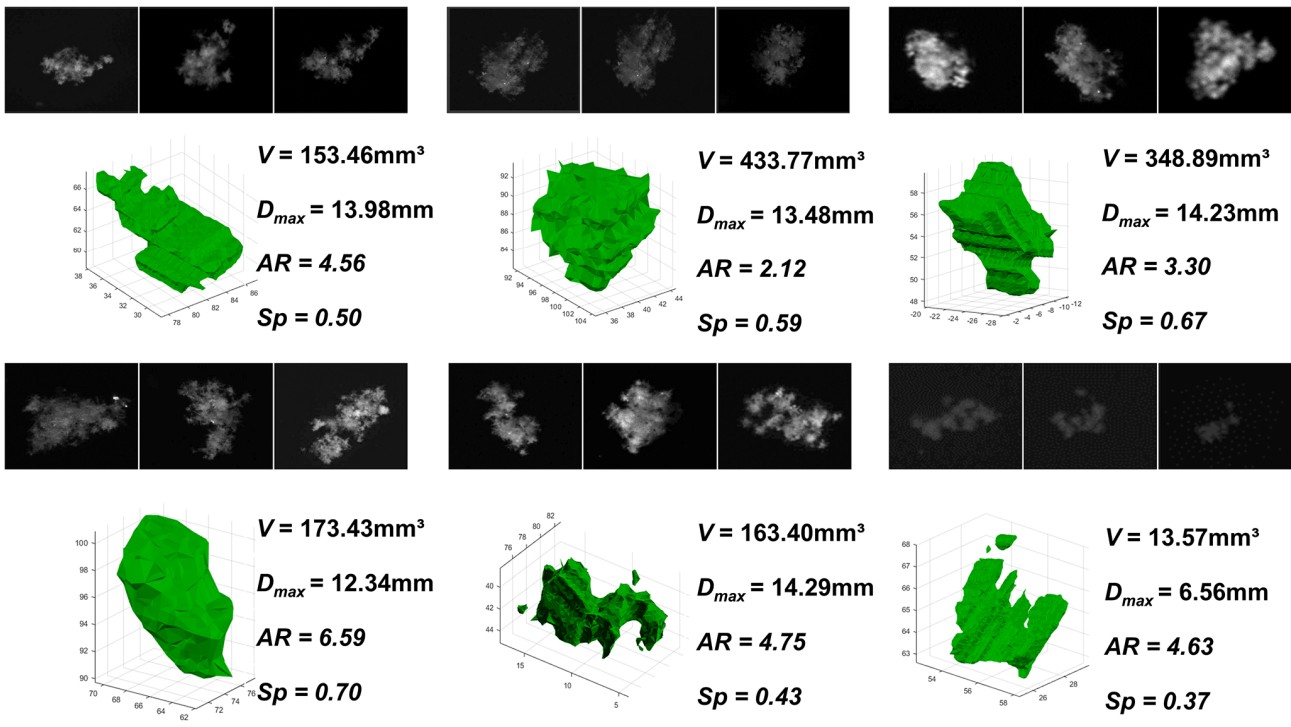

**Figure 14.** Several typical snowflakes captured by 3D-PPI in the field and the corresponding 3D reconstruction results. For each reconstruction, the computed $V$, $D_{max}$, $AR$, and $Sp$.

## 5.3 Fall velocity of snowflakes

The exposure time for a single frame is 20 μs, which renders the motion blur of the particles negligible. Additionally, the
time interval between two consecutive frames is 5 ms, allowing the same particle to be captured multiple times, thus enabling accurate velocity calculations. The same particle from consecutive frames is merged into a single image in Fig. 15a to enhance the visualization of its movement. The speed calculation schematic is shown in Fig. 15b.

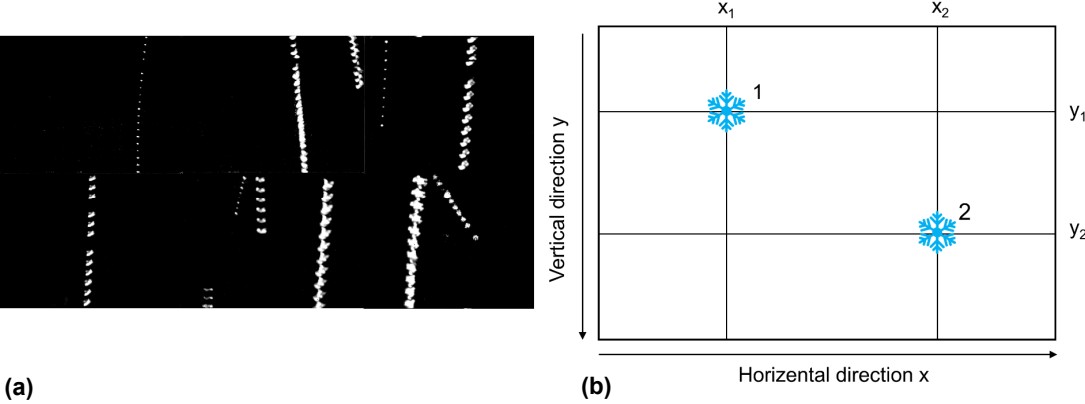

**(a)**            **(b)**

**Figure 15.** Processed high-speed camera images, and then the same particles from consecutive frames are merged into a single image **(a)**. Speed Measurement Schematic **(b)**.

When there are multiple particles in a high-speed camera image, particle matching is required. The same particle is regarded as the same particle when the following two principles are satisfied: (i) The pixel coordinates of the centroid of the contours in the particle images in consecutive frames are similar (the falling velocity of the snowflake is generally not more than 4m/s, so the distance between the vertical pixel coordinates of the same particle centroid in two adjacent image frames is not more than 100 pixels); (ii) Given that the size of the particles captured in two consecutive frames does not change significantly, the $D_{max}$ and $D_{eq}$ of the particles are similar, generally not more than 20% (the $D_{max}$ or $D_{eq}$ value of the particle in the next frame deviates within ±20% of the previous particle). Each particle may have recorded anywhere from 2 to 20 $D_{max}$, $D_{eq}$, pixel horizontal coordinates, and pixel vertical coordinates. The standard deviation of $D_{max}$, $D_{eq}$, and the difference between the horizontal and vertical coordinates of each particle is calculated, and if the standard deviation of any of the particle's quantities is too large, the particle is treated to be an invalid particle, and it will be discarded. $D_{max}$ and $D_{eq}$ of each particle are taken to be the maximum of these values, which excludes cases where the particle is not captured because it is at the edge of the image. The horizontal velocity ($V_h$) component and vertical velocity component ($V_v$) are calculated as follows:

$$V_h = \frac{x_2 - x_1}{\Delta t} \cdot \alpha \tag{12}$$

$$V_v = \frac{y_2 - y_1}{\Delta t} \cdot \alpha \tag{13}$$

Where, $x_2$, $x_1$ and $y_2$, $y_1$ denote the horizontal and vertical coordinates of the same particle in two consecutive frames; $\Delta t$ denote exposure interna which is generally 5ms; $\alpha$ denotes the pixel resolution of the high-speed camera at the working distance, which is 265 $\mu m \cdot px^{-1}$.

From 0800 UTC to 0830 UTC on 6 April 2024, the Cam3 of 3D-PPI recorded 77042 valid snowflakes, and for each snowflake, horizontal and vertical velocities were calculated. The distributions of horizontal and vertical velocity components as a function of $D_{eq}$ are further plotted as a scatter density plot and compared to the results measured by OTT at the same period, which is shown in Fig.16. The color scale denotes the number of snowflakes measured in the corresponding bins.

OTT's $D$ and $V$ binning is uneven, whereas here 3D-PPI is set to even binning with $D$ and $V$ intervals of 0.1mm and 0.1m/s, respectively. The red and black solid lines in Fig. 16b, d represent the empirical curves of the falling velocity and diameter of unrimed aggregates and densely rimed dendrites, respectively (Locatelli and Hobbs, 1974).

445    The empirical velocity of unrimed aggregates is:

$$V_1 = 0.81D^{0.16} \tag{14}$$

The empirical velocity of densely rimed dendrites is:

$$V_2 = 0.62D^{0.33} \tag{15}$$

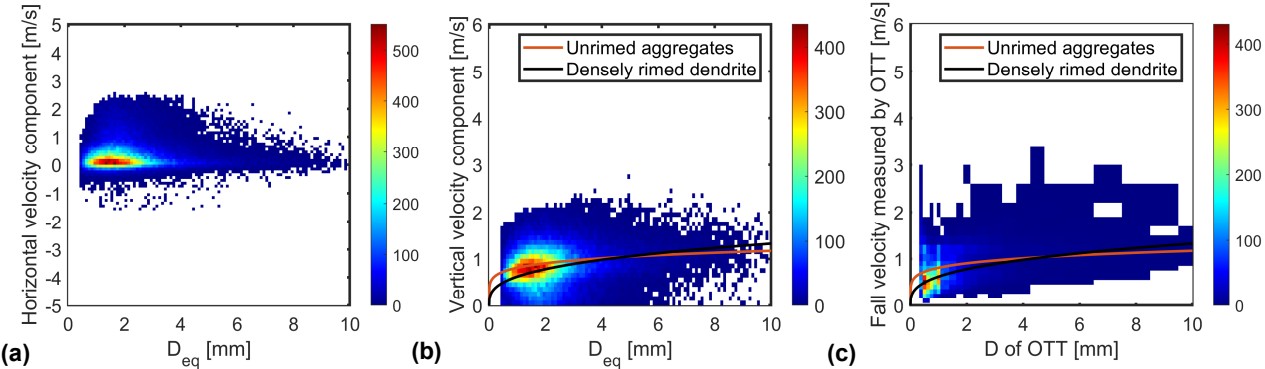

**(a)**                    **(b)**                    **(c)**

450

**Figure 16.** Distribution of horizontal **(a)**, and vertical **(b)** snowflake velocities with $D_{eq}$ measured by 3D-PPI. Distribution of falling velocity with diameter measured by OTT **(c)**. The color scale denotes the number of snowflakes measured.

Installing the instrument facing south means that the horizontal velocity seen by the high-speed camera Cam3 corresponds to East-West. The average value of the horizontal velocity measured by 3D-PPI is +0.41m/s, and the standard deviation is 455    0.73m/s (Fig. 16a). The overall distribution of particle horizontal velocities ranges between ±3m/s, and more than 70% of the snowflakes have a horizontal velocity distribution between ±1m/s. Positive velocities predominate over negative ones, largely influenced by the prevailing westward winds. The average value of the vertical velocity component measured by 3D-PPI is 0.74m/s and the standard deviation is 0.62m/s, while, the average value and standard deviation of the velocities measured by OTT are 0.69m/s and 0.31m/s. The diameters of snowflakes measured by 3D-PPI (OTT) were concentrated in the range of 0.5 460    to 2.2 mm (0.4 to 1 mm). The vertical velocities were concentrated in the 0.6 to 1.1 m/s (0.3 to 0.7 m/s). The diameter and velocity values measured by 3D-PPI are larger and more dispersed than those measured by OTT. Overall the vertical velocity of snowflakes increases with the increasing diameter, and the observed data are in good agreement with the two empirical velocity relationships (Locatelli and Hobbs, 1974). It should be noted that a small number (7% of the total sample) of small snowflakes have almost 0 vertical velocity or are negative (not shown due to the positive vertical axis range of Fig.16b). This 465    phenomenon may be attributed to localized updrafts caused by thermal convection or wind shear. Additionally, lighter or

smaller snowflakes may be temporarily suspended in the air due to turbulence, resulting in a recorded velocity of almost zero or upwards.

## 6 Conclusion

The design of a Three-Dimensional Precipitation Particles Imager (3D-PPI) has been introduced in this paper. The 3D-PPI consists of three high-resolution cameras (4096×3000, 5fps) with telecentric lenses and one high-speed camera (720×540, 200fps) with a non-telecentric lens. Three high-resolution cameras are oriented at a 45° angle relative to the optical axis of the high-speed camera, forming an intersecting observation volume of 775 cm$^3$. The high-resolution cameras feature a pixel resolution of 41.5 μm·px$^{-1}$ and are precisely synchronized by clock control which is sufficient to obtain fine shapes of snowflakes larger than 0.2mm, and the large field of view of 170 mm×125 mm enables it to capture enough snowflakes to estimate PSD accurately. The high-speed camera allows for the calculation of velocity accurately. Besides, the utilization of telecentric lenses eliminates the sizing error caused by the uncertain distance between the snowflakes and the cameras.

For three high-resolution cameras, a calibration method using the 3D checkerboard was proposed. By shooting the 3D checkerboard grid from three angles simultaneously, find the correspondence between the world coordinate points and the image coordinate points and then solve the system of equations to estimate the projection matrix of the three high-resolution cameras. A reprojection averaging error of 0.32 pixels can indicate the accuracy of the calibration. However, even minor displacements of the cameras can alter the projection matrix, which may adversely affect the subsequent reconstruction results. Therefore, it is essential to perform periodic calibration to ensure the accuracy and reliability of the projection matrix. Image binarization calibration is achieved by photographing ceramic reference spheres with different diameters of absolute sphericity. Both types of image processing require binarization and particle detection, and high-speed cameras require background removal, enhancement, and denoising before these two steps. The image processing algorithm needs to be evaluated by batch processing of ceramic sphere images, and the average values of the relative errors of $D_{max}$ and $D_{eq}$ are +2.2% and -2.7%, respectively. The issue of matching the same particle by its position in the image can be addressed by using the projection matrix obtained from the calibration of cameras in Sec. 3.1. The preliminary determination of the 3D spatial localization of particles after particle matching can effectively improve the computational efficiency of the 3D reconstruction algorithm, so particle localization is an indispensable step before 3D reconstruction. The snowflake 3D shape is further reconstructed using a visual hull algorithm based on binarized contour images from different angles and projection matrixes (Kleinkort et al., 2017).

The 3D-PPI was installed at Tulihe, China on January 1st, 2024, and the OTT was installed 10 meters apart for comparison. The PSDs, 3D shapes, and fall velocity of snowflakes were preliminarily analyzed. The PSD measured solely by Cam1 and that obtained by OTT exhibit excellent agreement during the typical snowfall case. Several snowflakes with different morphologies were selected and reconstructed in three dimensions, indicating that 3D-PPI is initially capable of reconstructing snowflakes. The horizontal and vertical velocities of snowflakes were calculated to obtain the velocity distribution. Further comparisons were made with the OTT, and overall, the two distributions for fall velocity were similar, however, the diameter

and velocity values measured by 3D-PPI are larger and more dispersed than those measured by OTT. This difference may be attributed to the potential magnification differences of the high-speed camera in the 3D-PPI due to particles being at varying distances from the cameras.

In this paper, The PSD statistics use only one image from a high-resolution camera, and the 3D reconstruction is limited to just one case study. The next step is to optimize the 3D reconstruction algorithms of snowflakes, and statistically characterize the distribution of the 3D shape of a certain number of snowflakes. Also, the accuracy of velocity measurement still needs to be verified and improved. In the future, the 3D-PPI will facilitate more precise and realistic estimations of the snowflake parameters, including the size, volume, mass, and density. Based on the above details parameters, the 3D-PPI has the potential to improve the radar-based estimation for solid precipitation in winter.

*Data availability.* The raw and processed 3D-PPI data mentioned in the paper are available upon request to the authors.

*Author contributions.* JS processed the 3D-PPI data, researched the algorithms, analyzed the data of the case, and drafted the manuscript. XL acquired funding, developed the 3D-PPI instrument, and proofread the manuscript. LL supported funding acquisition, designed the instrument, and conducted field experiments. LL contributed to instrument calibration. PW developed the hardware and software of 3D-PPI. All authors reviewed and edited the draft.

*Competing interests.* The authors declare that they have no conflict of interest.

*Acknowledgements.* Funded by the National Key Research and Development Program of China (Grant 2021YFC2802501) and the National Natural Science Foundation of China (Grant No.42222505).

## Appendix A: Coordinate system transformation

Camera calibration encompasses four key coordinate systems:

1. World coordinate system (WCS): Denoted as $(X_w, Y_w, Z_w)$: This is a user-defined 3D spatial coordinate system that is utilized to describe the location of the target object within the tangible world, with units typically expressed in millimeters.

2. Camera coordinate system (CCS): Denoted as $(X_c, Y_c, Z_c)$: This coordinate system is intrinsic to the camera and is utilized to describe the object's position relative to the camera's perspective. It acts as an intermediary between the WCS and the image (pixel) coordinate system.

3. Image coordinate system (ICS): Denoted as $(x, y)$, this system is employed to articulate the projection and translation of the object from the CCS to the ICS during the imaging process. It facilitates the subsequent extraction of coordinates under the pixel coordinate system, with the unit being millimeters.

4. Pixel coordinate system (PCS): Denoted as ($u$, $v$), this system describes the coordinates of the object's image point post-imaging on the digital image sensor. It is the actual coordinate system from which image information is read from the camera, measured in units of pixels.

The camera imaging process involves the transformation from WCS to PCS. Camera calibration, in essence, is the procedure of determining the transformation relationships between these four coordinate systems.

**A.1 WCS to CCS**

Firstly, the transformation of a camera shot from the WCS to the CCS is a rigid-body transformation, where the object does not deform, but only rotates and translates. Only the rotation matrix $R$ and translation matrix $T$ need to be obtained. The camera coordinate system is obtained by rotating $\theta$, $\alpha$, and $\beta$ angles around the z, y, and x-axes in turn and translating to obtain the rotation matrix in the three dimensions:

$$R_z(\theta) = \begin{bmatrix} \cos(\theta) & \sin(\theta) & 0 \\ -\sin(\theta) & \cos(\theta) & 0 \\ 0 & 0 & 1 \end{bmatrix}$$

$$R_y(\beta) = \begin{bmatrix} \cos(\beta) & 0 & -\sin(\beta) \\ 0 & 1 & 0 \\ \sin(\beta) & 0 & \cos(\beta) \end{bmatrix} \tag{A.1}$$

$$R_x(\alpha) = \begin{bmatrix} 1 & 0 & 0 \\ 0 & \cos(\alpha) & \sin(\alpha) \\ 0 & -\sin(\alpha) & \cos(\alpha) \end{bmatrix}$$

The three matrices are multiplied together to obtain a three-dimensional rotation matrix:

$$R = R_x(\alpha) R_y(\beta) R_z(\theta) =$$
$$\begin{bmatrix} \cos(\beta)\cos(\theta) & \cos(\beta)\sin(\theta) & -\sin(\beta) \\ -\cos(\alpha)\sin(\theta)+\sin(\alpha)\sin(\beta)\cos(\theta) & \cos(\alpha)\cos(\theta)+\sin(\alpha)\sin(\beta)\sin(\theta) & \sin(\alpha)\cos(\beta) \\ \sin(\alpha)\sin(\theta)+\cos(\alpha)\sin(\beta)\cos(\theta) & -\sin(\alpha)\cos(\theta)+\cos(\alpha)\sin(\beta)\sin(\theta) & \cos(\alpha)\cos(\beta) \end{bmatrix} \tag{A.2}$$

Where, for z, y, and x direction of rotation is followed by the right-hand spiral rule, the thumb points to the direction of the axis, and the four-finger direction is the positive direction of rotation.

For the translation matrix, at this point, the coordinates are already in the same direction as CCS, but with the world coordinate system origin coinciding with the coordinates under the coordinates, converting the camera coordinate system also needs to be added to the translation is WCS origin in CCS under the coordinates $T$.

The rotation and translation process can be expressed by the formula:

$$\begin{bmatrix} X_c \\ Y_c \\ Z_c \\ 1 \end{bmatrix} = \begin{bmatrix} R_{3\times3} & T_{3\times1} \\ O & 1 \end{bmatrix} \cdot \begin{bmatrix} X_w \\ Y_w \\ Z_w \\ 1 \end{bmatrix} \tag{A.3}$$

**A.2 CCS to ICS**

The difference between telecentric cameras and traditional pinhole cameras is the difference in projection. A pinhole camera uses a perspective projection to transform from CCS to ICS; a telecentric camera uses an orthogonal projection. The relationship between CCS and ICS is as follows:

$$
\begin{bmatrix} x \\ y \\ 1 \end{bmatrix} = \begin{bmatrix} \beta & 0 & 0 & 0 \\ 0 & \beta & 0 & 0 \\ 0 & 0 & 0 & 1 \end{bmatrix} \cdot \begin{bmatrix} X_c \\ Y_c \\ Z_c \\ 1 \end{bmatrix}
$$

(A.4)

Where $\beta$ is the magnification of the telecentric lens of the telecentric camera. It is not difficult to see that the image coordinates are independent of the camera coordinates $Z_C$, i.e. the distance of the object to be photographed from the lens does not affect the imaging (projection) of the image, which is also in line with the characteristics of telecentric lens imaging.

**A.3 ICS to PCS**

To convert a point in ICS whose origin is at the center of the light in real physical units to a point in an image coordinate system whose origin is at the top left corner of pixels requires two transformations, translation, and scaling, which are affine transformations.

$$
\begin{bmatrix} u \\ v \\ 1 \end{bmatrix} = \begin{bmatrix} \dfrac{1}{S_u} & 0 & u_0 \\ 0 & \dfrac{1}{S_v} & v_0 \\ 0 & 0 & 1 \end{bmatrix} \begin{bmatrix} x \\ y \\ 1 \end{bmatrix}
$$

(A.5)

Where the pixel size is $S_u \times S_v$, $(u_0, v_0)$ is the pixel coordinate of the optical center point.

**A.4 WCS to PCS**

Integration of expressions from the first three sections:

$$
\begin{bmatrix} u \\ v \\ 1 \end{bmatrix} = \begin{bmatrix} \dfrac{1}{S_u} & 0 & u_0 \\ 0 & \dfrac{1}{S_v} & v_0 \\ 0 & 0 & 1 \end{bmatrix} \cdot \begin{bmatrix} \beta & 0 & 0 & 0 \\ 0 & \beta & 0 & 0 \\ 0 & 0 & 0 & 1 \end{bmatrix} \cdot \begin{bmatrix} r_{11} & r_{12} & r_{13} & t_x \\ r_{21} & r_{22} & r_{23} & t_y \\ r_{31} & r_{32} & r_{33} & t_z \\ 0 & 0 & 0 & 1 \end{bmatrix} \cdot \begin{bmatrix} X_w \\ Y_w \\ Z_w \\ 1 \end{bmatrix}
$$

(A.6)

$$
= \begin{bmatrix} \dfrac{\beta}{S_u} & 0 & u_0 \\ 0 & \dfrac{\beta}{S_v} & v_0 \\ 0 & 0 & 1 \end{bmatrix} \cdot \begin{bmatrix} r_{11} & r_{12} & r_{13} & t_x \\ r_{21} & r_{22} & r_{23} & t_y \\ 0 & 0 & 0 & 1 \end{bmatrix} \cdot \begin{bmatrix} X_w \\ Y_w \\ Z_w \\ 1 \end{bmatrix}
$$

Similar to small hole imaging, $\begin{bmatrix} \dfrac{\beta}{S_u} & 0 & u_0 \\ 0 & \dfrac{\beta}{S_v} & v_0 \\ 0 & 0 & 1 \end{bmatrix}$ is the internal parameter of the camera, which only relates to the camera itself

and has nothing to do with the position of the camera. $\begin{bmatrix} r_{11} & r_{12} & r_{13} & t_x \\ r_{21} & r_{22} & r_{23} & t_y \\ 0 & 0 & 0 & 1 \end{bmatrix}$ is an external parameter of the camera, representing

the position of the camera. It has nothing to do with camera manufacturing or lens distortion, but only with the mounting

position and angle of the camera in WCS. R and T represent the rotation and translation process from WCS to CCS, respectively.

Compared to common pinhole lenses, the four quantities $r_{31}$, $r_{32}$, $r_{33}$, and $t_Z$ in the third row of the external reference matrix of

the telecentric camera do not exist. This further confirms the special feature of telecentric camera imaging, i.e. it is a parallel

light projection and the distance of the object from the camera does not affect the size of the object in the image.

## Appendix B: Additional equation-solving process

To solve for $(X_w, Y_w, Z_w)$ based on the known points P($u,v$) and $KM_0$, simplify Eq. (2) by removing 1 from the second

term:

$$KM_0 \cdot \begin{bmatrix} X_w \\ Y_w \\ Z_w \\ 1 \end{bmatrix} = \begin{bmatrix} u \\ v \end{bmatrix} \xrightarrow{\text{Simplify}} A \cdot \begin{bmatrix} X_w \\ Y_w \\ Z_w \end{bmatrix} = B \quad\quad (\text{B.1})$$

Where $A$ is a known 2 × 3 matrix and $B$ is a known 2 × 1 matrix, it is equivalent to underdetermined linear equations. The

solution is not unique and is shown in Eq. (B.2):

$$\begin{bmatrix} X_w \\ Y_w \\ Z_w \end{bmatrix} = U t + V = \begin{bmatrix} U_1 t + V_1 \\ U_2 t + V_2 \\ U_3 t + V_3 \end{bmatrix} \quad\quad (\text{B.2})$$

Where both $U$ and $V$ are 3 × 1 matrixes, $t$ is any real number. Therefore, all solutions form a straight line L in 3D space

WCS. In other words, this process implements the back-projection of P onto the line L.

Furtherly, project the L onto the planes of Cam1 and Cam2 by multiplying the projection matrices $KM_1$ and $KM_2$,

respectively, shown in Eq. (B.3):

$$L_{p1} = KM_1 \cdot \begin{bmatrix} U_1 t + V_1 \\ U_2 t + V_2 \\ U_3 t + V_3 \\ 1 \end{bmatrix} = \begin{bmatrix} f_1(t) \\ g_1(t) \end{bmatrix} \quad L_{p2} = KM_2 \cdot \begin{bmatrix} U_1 t + V_1 \\ U_2 t + V_2 \\ U_3 t + V_3 \\ 1 \end{bmatrix} = \begin{bmatrix} f_2(t) \\ g_2(t) \end{bmatrix} \quad\quad (\text{B.3})$$

Where $L_{p1}$ and $L_{p2}$ denote the point sets of projections of L onto the Cam1 and Cam2 planes respectively. The functions $f_1(t)$, $g_1(t)$, $f_2(t)$, and $g_2(t)$ are all linear functions of $t$. $L_{p1}$ and $L_{p2}$. Therefore, $L_{p1}$ and $L_{p2}$ represent straight lines in the plane. Determine the range of $t$ to ensure that the line is within the image range (4096×3000) to get the corresponding line segments.

## Appendix C: Wind field simulation

To determine the optimal orientation of the 3D-PPI installation (mainly considering the relationship with the prevailing wind direction), we conducted wind field simulations using Solid flow simulation software. The simulation results are shown in Fig. B.1.

When the 3D-PPI is facing the wind (Fig.B.1a), the observation volume experiences an average wind speed of approximately 6.0 m/s. Besides, turbulence may occur within the observation volume. When the 3D-PPI is back facing the

wind (Fig. C.1b), the average wind speed in the observation volume is only about 3.5 m/s, which is obviously due to the shielding of the wind by the instrument. When the 3D-PPI is side facing the wind (Fig. B.1c), the observation volume shows an average wind speed of about 8.5 m/s, exhibiting the smallest difference from 10m/s, compared to the other two situations. However, part of the observation volume close to the instrument is still shielded by the housing, which to some extent also affects the representativeness of the wind field, and subsequent consideration will need to be given to improving the instrument

design to solve this problem.

In addition to 10m/s, we also simulated 5m/s, 20m/s, and 40m/s wind speed fields, and all of them got the consistent conclusion that the wind speed in the observation volume is closest to the simulated wind speed when the instrument is side facing the prevailing wind direction. Therefore, the instrument should be installed sideways to the dominant wind direction in the area, to minimize the disturbance of the instrument to the natural wind field. The prevailing wind direction in the area is

west, so the 3D-PPI is installed facing towards the south.

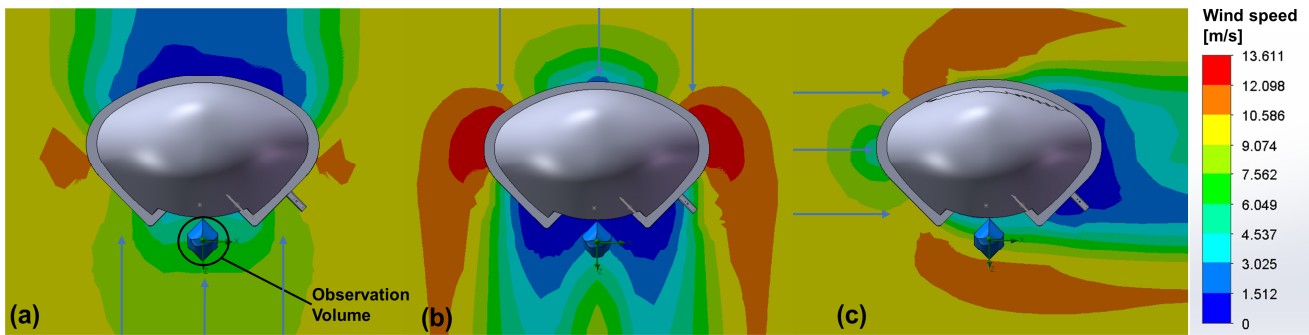

**Figure C.1.** Top view of wind speed distribution in the simulated wind field. 3D-PPI facing 10 m/s wind **(a)**; back facing 10 m/s wind **(b)**; side facing 10 m/s wind **(c)**. The color gradient represents wind speed, with the observation volume indicated in **(a)**.

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
