# Peer review of "An introduction of the Three-Dimensional Precipitation Particles Imager (3D-PPI)"

_Atmospheric Measurement Techniques, 2024_

## Referee Comment (RC1)

Review of "An introduction of Three-Dimensional Precipitation Particles Imager (3D-PPI)"

This manuscript describes a new instrument designed to capture the three dimensional structure of snowflakes. The instrument is composed of three high-resolution cameras and one high-speed camera with an LED light source. Based on the technical specifications listed in Table 1 and the example imagery, this instrument appears to be well designed for capturing snowflakes. The manuscript itself is highly detailed, something this reviewer greatly appreciates in an instrument paper, and is well written. That said, I feel some farther analysis and/or discussion is necessary to ensure future users have a complete grasp of the instrument and its data. I debated whether or not my suggestions constituted a major or minor revision; ultimately, I decided to err on the side of caution and list them as major revisions.

General Comments:

1. How robust are the camera alignment and 3D measurements to thermal expansion shifting the cameras (since the cameras are enclosed in a housing, I assume the effects of wind on camera alignment will be negligible)? Adding a brief discussion of how these might affect the measurements would be helpful. Some tests of how much measurement error is produced from slight changes in camera alignment after the calibration could also be illuminating.
2. Given that the sampling volume is so close to the instrument housing, how will wind flow affect the representativeness of the measurements (e.g., increasing or decreasing the collection efficiency of the sampling volume)? While I understand that a full engineering analysis of the wind effects (similar to what was presented in Fig 3 of Newman et al. 2009, https://doi.org/10.1175/2008JTECHA1148.1) may be outside the scope of this paper, some analysis that compares the 3D-PPI to the PARSEVAL with the wind coming from certain directions relative to the camera pointing direction might help give some insight into how well the instrument handles these conditions.
3. Since LEDs rapidly flash to produce light, it would be worth clarifying how the LED interacts with the cameras. Is the 20-microsecond effective exposure time due to the duration of the LED flash being 20 microseconds? Is there some mechanism to ensure that the LED is synchronized with the camera exposures in some way to ensure each frame has consistent illumination (or is the system sufficiently robust that inconsistencies in LED illumination are not a problem)?
4. How are particles at the edge of the field of view handled? I seem to recall reading somewhere in the manuscript that particles touching the edge are ignored (but I can't seem to find where that was now, sorry if I'm mistaken). If the edge particles are discarded, I think the effective sampling volume would need to be a function of particle size (with the sampling volume being smaller for larger particles).
5. Is the tracking software intended to work for rain drops as well as frozen precipitation? It is clear from the manuscript that the instrument is designed for measuring snowflakes, it might be worth noting in section 5.3 whether or not the tracking software is designed to handle any faster falling precipitation.
6. I have some concerns with the tracking algorithm, although it is possible these concerns are due to misunderstanding on my part and/or a need for further clarification in the manuscript.

When first reading over the tracking method, I was somewhat concerned that particle motion wasn't being considered as, in my personal experience, that is the best way to match an existing particle to its new location (although this isn't possible in the first two frames of the particle and some other criteria must be applied). After having studied the description more closely, it sounds like there might be some shape matching going on. I base this on Line 393, where the pixel coordinates of a particle are mentioned. Are these pixel coordinates being used to match the shape of the particle (e.g., the spatial distribution of these pixels relative to some particle centroid is compared between frames via some method)? If so, this warrants more discussion in the manuscript. If not, how are these pixel positions being used (or am I misunderstanding what is meant by pixel coordinates)? Given that the position difference allows for up to 8 m/s of particle motion, I would expect that the other criteria need to be very robust to avoid mismatches.

Specific Comments:

Title: suggest adding "the" before "Three-Dimensional"

Line 16: change "OTT a good agreement" to "OTT have good agreement"

Line 80: add "a" before "field experiment"

Table 1: Thank you for including this table! Having these technical specifications in one place is very useful.

Lines 107 – 111: These numbers don't seem to agree with those in Table 1. Specifically, the telecentric lens pixel size is listed as 3.45 microns in the table and 42 microns in the paragraph. Similarly, the table lists the non-telecentric lens as having a pixel size of 6.9 microns while the text lists 265 microns. It's very possible that these are referring to two different measurements and if that is the case, I encourage the authors to make that clear either in the text or in the table caption.

Lines 177 – 179: the parenthetical "($D_{max}$ is the distance between the two largest points of the particle profile..." could use rewording. Perhaps replacing "largest" with "farthest" would improve the readability?

Line 180: add "the" before "spheres"

Figure 6: Move the first line of the caption to below panel b.

Lines 208 – 210: These sentences could use some clarification. Are these particle centers as seen from a single camera or are these the particle centers from multiple cameras? If these are from the same camera, it might help clarify things to mention that sometimes a single particle will appear as two particles due to being on the edge of the image processing threshold (or whatever reason is appropriate). Just based on the flow of the manuscript, I assume these are for a single camera, but it might help make things clearer to specify that.

Line 224: Are the ceramic spheres the same as the ones used in section 3.2? If yes, perhaps changing the text from "different diameters in section 3.2 were dropped" to "different diameters, as described in section 3.2, were dropped" would make that clear.

Line 227 – 234: The text mentions that the measurements of the smaller spheres tend to be larger than the true size, but the average error of $D_{eq}$ for the small spheres is negative. I may be missing something, but wouldn't a negative error mean the measurements of the small spheres are smaller than their true value? Also, the authors state that this is the average absolute error, which to me implies that it is the mean of the absolute value of the difference between the measured and true $D_{eq}$ and, therefore, should not be negative. I assume the authors are using "absolute" error in contrast with "relative" error. I'm not sure how best to fix this misunderstanding though.

Figure 8: Are these plots for the high-resolution camera or the high-speed camera? It might be helpful to mention which in the caption.

Lines 389 and 401: Line 389 mentions that the frames need to be adjacent to one another (which I interpret to mean that a missing frame is not allowed), but Line 401 mentions that a missing frame is allowed. This seems like a contradiction, but I suspect I'm just misinterpreting the authors' meaning. Some additional clarification in the text might be needed.

Lines 390 – 391: Is the 200 pixel interval purely in the vertical or is that include the horizontal component of the distance as well. If that includes the horizontal distance, it might be worth noting that this means the instrument will have difficulty producing accurate particle fall speeds at high cross-camera-view wind speeds (which isn't a problem, but is good to know for anyone performing an analysis of the data).

Lines 416 – 417: If westward motion is positive, I assume the 3D-PPI was facing towards the south? It might be useful to mention the direction the instrument is facing in the paragraph starting on Line 324 (preferably as a bearing, but a general direction would do if you don't know the exact bearing).

Lines 426 – 428: Are these outliers real or are they a result of mismatches by the tracking algorithm. If they're a result of mismatches, it might be helpful to provide some statistics regarding how prevalent these outlier are (e.g., what percentage of the total sample population or the a reasonable range of percentages of their size bin populations) so the reader can determine if they are sufficiently infrequent to be considered negligible.

Lines 434: Ensure the 41.5 microns per pixel matches what is said in the text (Lines 107 – 111) and in Table 1

---

## Referee Comment (RC2)

**Manuscript amt-2024-106**

**An introduction of Three-Dimensional Precipitation Particles Imager (3D-PPI)**

The manuscript describes a new instrument, the 3D-PPI, to image snowflakes from three different directions. Conceptually the 3D-PPI is similar to previous instruments, in particular the MASC and the VISSS, however it differs in the number of simultaneous images or the FOV. The authors attempted a detailed description of the instrument. In addition to this description of the 3D-PPI and its calibration, the manuscript presents details of a first measurement campaign including the comparison to a disdrometer (OTT).

Development of instruments to measure snow is important, and I therefore welcome this publication. However, a thorough revision is required to address some shortcomings and missing details or discussions. In addition, a particular issue that needs fixing consists in problems with language and clarity which are present throughout the whole manuscript.

Below are my comments. Firstly, I am addressing important issues that are not only caused by unclear or wrong language. Secondly, I am listing minor issues.
There are many issues with wrong English or otherwise unclear text and statements in the manuscript. This made it difficult to focus the review on other issues and in many places it was difficult to follow and understand the authors' explanations. I am reporting these issues together with the minor issues.

In my comments I will refer to line numbers using simply numbers (e.g. "143" refers to line number 143 in the manuscript). I will also use "um" to indicate micrometer without using special fonts.

**Important points in the order they appear in the manuscript**

1)
Previous instruments are presented such as the VISSS and the MASC. When describing the MASC the following statement is done (73-75): "Nevertheless, only 10^2–10^4 particles were observed during a typical snowfall event (Gergely and Garrett, 2016), which is insufficient to permit the reliable estimation of the particle size distribution (PSD) (Gergely and Garrett, 2016)." I don't see why this sample size would be insufficient and I don't see Gergely and Garrett, 2016 saying that. It is a bit misleading to look at sampled number of particles during specific storms rather than snowfall rates or snow number concentrations. The authors should instead look at the observation

volume or something similar if they want to put the MASC in relation to 3D-PPI. The MASC has three cameras, as the 3D-PPI, but they are all located in one plane. However, the MASC was extended to five cameras with the additional two not being in the same horizontal plane as the original three cameras (Notaros et al. 2016, Kleinkort et al. 2017).

2)
a) What is the actual resolution (combination of optical resolution and illumination that allows details of a certain size to be resolved)? What are the smallest details that can be resolved? This is not discussed but an important detail of a new instrument. I can only guess from the images (mostly Fig. 1c). The images look similar to MASC images of snow, likely due to a similar illumination scheme. The smallest details are very faintly grey and seem to disappear in the black background. Due to this and due to the missing description of the detection algorithm and thresholds (see further comment on image processing (comment 8) below), it is not possible to do a fair judgment of the actual resolution for detection of small details (e.g. thin branches).
b) Judging from Fig 6b, the sizes of the smallest ceramic spheres are underestimated. For each camera, the linear fit has a negative offset (see comment below) with the observed size being on or below that fit. So, there seems to be a systematic bias for sizing of the smallest details. Please comment on this.
c) Related to this is in 207 "Snowflakes that are too small in diameter are ignored". What is this diameter, how did you decide on its value? Also, in the next sentence you state that you connect apparently separate detected regions if they are up to 4mm apart. This indicates that you expect that smaller details that may be connecting these regions are not being detected.

3)
Inconsistent and wrong or confusing use of "resolution", "pixel resolution", "pixel size", and "magnification". Revise and use better and consistent terminology throughout the paper.
Examples:
a) In Sect 2: 92, I would not call this "resolution" as that can be confused with pixel resolution or optical resolution. Say something that is less ambiguous like: "The sensor has 4096 x 3000 pixels."
b) Tab 1: The "pixel resolution" is given as 3.45 um. In the text instead a single pixel size is stated as 0.042mm. This seems wrong or confusing to me. What you mean likely is that the sensor has a pixel size of 3.45 um and the pixel resolution (given by the magnification and pixel size) of 41.5 um/px.
c) Add magnification and pixel resolution to the table.
d) 402, 230um/px is pixel resolution, not "magnification".
e) The magnification 0.026 and pixel size of 6.9um gives a pixel resolution of 265um/px. Why do you state 230um/px here?
f) In Conclusion "resolution of 41.5um/px" is used.

4)
102, "observation volume size of 1505.327cm3": Part of the observation volume is out of focus (as depth of focus with 104mm is smaller than the FOV dimensions). Three digits after the decimal point are not needed.

5)
156 "super-determined": Does this refer to an overdetermined system with more equations than unknowns? What are the unknowns here in that case?
Do "super-determined" and "super-deterministic" refer to the same thing?

6)
163-169, explanations why the planar two-dimensional checkerboard grid cannot be used for calibration:
a) 164-165, "but then the values of all three dimensional world coordinate points are the same": this sounds very general; explain better what you mean, which points/coordinates are the same?
b) The sentence continues that with that A equals a matrix, which is the same matrix as the matrix KMi is multiplied with in Eq (2).
c) The matrix A has not been defined before. Please do that.
d) 165-167, "The third and fourth-row values are the same ..." This sentence is not correct. Please revise. The fourth-row values are all 1, did you want to say that all values in the third row are 1, or are the same but not necessarily 1 but any other value?
e) 167, "determinant of A is 0": as far as I know, the determinant is defined for square matrices, but A is not (unless j=4).
f) 168, "impossible to inverse" should be "impossible to invert"

7)
171-172, in this sentence at the end of Sect 3.1, you present an "average reprojection error". Nowhere in Sect 3.1 you present any calibration results related to the theoretical treatment presented. The theory presented seems needed to determine KMi, which are needed for the matching algorithm. Define what the average reprojection error is and how you have determined it. Here you say 0.32 pixels, in the Conclusion 0.4 pixels.

8)
Determination of pixel resolution (41.5um/px) in Sect 3.2 (Calibration of image binarization) and image processing in Sect. 4.1:
a) 181, "is optimally binarized manually": How has this manual binarization be performed. Is each image treated differently. Is this using the image processing algorithm described in Sect. 4 (where the details of the detection are missing, see comment about "adaptive thresholding below")?

b) You determine the pixel resolution from the reciprocals of the slopes in Fig 6b (Dmax/px vs Dmax/mm). You are not discussing the role of the offset of between 0.5px to 4px in Equations (3)-(5).

c) The pixel resolution should also result from the calibrations using chessboards. Has this been attempted and the values compared to the pixel resolution reported in Sect. 3.2?

d) The pixel resolution using chessboards would be similar to using micrometer or millimeter scales. The advantage over ceramic spheres would be that it would not depend on image processing (selection of specific grey level threshold to detect the contours of the ceramic spheres), which may result in under- or over-sizing of the spheres. The details of this image processing are, however, not disclosed. You only state (Sect. 4.1, 204) that the images are "binarized through adaptive thresholding" without giving any reference or explaining what adaptive thresholding is in your specific case. Please explain this method.

e) In Sect. 4.1 describing the image processing you show results from imaging the ceramic spheres (Fig. 8a and 8b) in terms of measured size vs real size (both in mm). You must have used the pixel resolution resulting from imaging the ceramic spheres (Sect. 3.2, Fig 6b). For the "calibration of image binarization" you must have used the image processing algorithms described in Sect. 4.1. That means that Fig 8 does not show anything new or independent from previously reported Fig 6b. You are presenting the same ceramic sphere measurements in two related ways. Consequently, the error analysis related to Fig 8 is only a re-interpretation of the error analysis of Fig 6b.

f) In any case, the presented error analyses are unclear and confusing:
L196-197, "The estimated random errors from the normalized root square errors, derived from the observed and true size difference …"
Unclear what "observed and true size difference" is.
Unclear what "random" refers to?
Does it refer to variations in 20 observations of the size of the same ceramic sphere?

g) Due to the offset in your linear least squares fits, the reciprocals of the coefficient px/mm (24px/mm here) are not equal to the coefficients that would result from fitting true size vs observed size (mm/px). Redo that fit or comment.

h) 227, "average absolute error of Dmax measurements for all diameters of small spheres is -0.048mm,"
An absolute error is positive, but you state that it is -0.048mm. Same for Deq with -0.33mm.
Then it is unclear what the "average relative error" (229, 230) is.

9)
206-207, "Detecting connected regions … enables the removal of small noise …"
Missing/wrong logic:
What is enabling the removal of noise from image? What is small noise on the image?

10)
207, "Snowflakes that are too small in diameter are ignored"
What is this diameter, how did you decide on its value?

This is related to the comment about actual resolution. Here you seem to have defined a smallest particle that is accepted.

11)
243-252, explanation of matching algorithm:
This is a clever way to use the matrices KMi (i=1,2,3) from camera calibration for matching. The following issues/unclarities result from wrong language or unclear formulation.
a) I would not re-use the index i, that previously was used to refer to the three cameras, to refer now to a particle number. Use another index or no index.
b) 245, "i underdetermined linear equations": Is there one underdetermined linear equation for each particle? Refer to an equation number in the paper or explain better what these equations are (and why/how underdetermined).
c) Would be good to add equation (if appropriate in Appendix), the solution of which is Li. Then also another equation showing the multiplication of KM1 and KM2 to produce the projection of Li onto Cam1 and Cam2 image planes.
d) 246, "ith straight lines" should be "i straight lines"?
e) 247, "the lines Li is the back-projection" should be "the lines Li are the back-projections".
f) 248, "multiplying the projection matrices KM1 and KM2," should be "multiplying the projection matrices KM2 and KM3, respectively, with Li,"
g) 249, "resulting in ith line segments" should be "resulting in i line segments on each of the image planes of Cam1 and Cam2"
h) 251-252: from the text it is unclear if each particle has to be on all three lines, or each particle on its respective line in each of the images. Describe for one particle (to avoid confusion between THREE cameras, and THREE particles/lines).

12)
259-294, 3D reconstruction:
The description of how particle is located to enable/simplify 3D reconstruction is unclear. Here a few things that make it difficult to follow:
a) Add new sub-section (now it is part of 4.2 Particle matching and localization).
b) 259, Not sure what voxels are and why/how they are used traditionally. Provide a reference.
c) 260-261, Explain or discuss how are "traditional methods" "computationally inefficient".
e) 264, "particles' localizations" should be "particles' locations"?
f) 266-271, Unclear why irregular particles pose a problem.  You probably need to be more specific in explaining the issue
g) 277-278, confusing use of indices:
"Lines L1", I thought L1 is one line, the back-projected line from P1 on image of Cam0.
"L2" according to earlier explanations (247), this is the back-projected line from P2 (particle 2) on image of Cam0, but here is redefined.
h) How can L2, a line, be "represented as a 2-row by 1-column matrix"?
279-280, "P2" and "P2'", what is P2 (it is not the P2 on image of Cam0, 243)?

It becomes more and more tedious to follow the remaining explanations. This whole section should be revised for consistency and clarity.

13)
Sect 4.3: It is not immediately clear if and how this connects to explanations in 259-294.
296-297: "silhouettes that have been serially calibrated using multiple viewpoints around the target" The section starts with unclear formulations like this one (are silhouettes contours, multiple viewpoints refer to Cam0,1,2, what does "serially calibrated mean"?).
I did not review the remaining of this section. After revising the previous section, this section should be revised too.

14)
a) Explain better Eq (7).
b) "Nima is the number of particles" should be "Nima is the number of acquired images"?
c) You state that (339) "the probability of capturing the same snowflake in two consecutive frames is very low." How often can the same snowflake be captured twice? Discuss if anything is done to account for this or, if not, how big of an error this may cause.
d) You are getting the PSDs only from Cam0 here. That means Vobservation is different from the observation volume reported in 102 (1505cm3). Vobservation is calculated using the depth of focus (104mm). How well-defined is the depth of focus? Are particles outside depth of focus detected but then rejected reliably?

15)
From Fig16b I would guess that the PSD peaks at Deq between 1 and 2 mm, whereas in Fig 13 the PSD peaks at Deq of about 0.4mm. Is everything consistent here and I am only being confused and misled by something?

16)
Negative velocities:
a) L416-417 "The average value of the horizontal velocity component measured by 3D-PPI is +0.05m/s (positive and negative values indicate westward and eastward velocities, respectively), and the standard deviation is 2.56m/s (Fig. 16a)."
Unclear if your average is an absolute average (or do you consider positive and negative values when averaging)? Consequently, the meaning of the standard deviation is unclear.
b) I would expect horizontal speed to be characterized by speed (absolute value) and direction (0 – 360degrees). Instead, you use positive and negative values to indicate westward and eastward. Why choosing to give info on west-east, and not north-south or the actual direction in degrees?

c) If you choose to include directional information, why are you not analyzing the direction.
I would expect the horizontal speed direction to correlate with local wind direction. Was the local wind measured and compared to horizontal speed? This could be part of a discussion how wind affects measurements.

**Minor issues**

1)
The three cameras are numbered 0,1,2 (Cam0, …), whereas you use the indices i=1,2,3 to indicate the three cameras (e.g. KM1, KM2, KM3).
Can you use the same indices to reduce confusion?

2)
"and" in wrong place in a list (85, 185).

3)
86-87, "capacitive rain sensor is adopted as a trigger, the cameras only work when the precipitation occurs."
Rain sensors detect rain. When you say the cameras only work when precipitation occurs, do you mean when it rains? I would expect you want to measure with snow but perhaps not with rain? Please clarify.

4)
Fig 1a: It would be useful to label the different parts. Without labels it is, for example, not clear which/where is the fourth non-telecentric camera. Labels are a complement to more clarity in the text.

5)
98-99, "45° angle relative to the optical axis of the high-speed camera" Be more specific: two cameras are positioned, at 45 degrees, on either side of the high-speed camera in the same horizontal plane, the third …. at 45 degrees vertically above

6)
100-101, "overlapping region of the LED lighting beams" and intersection of the "three rectangular light columns"???
I don't think you are talking about the LED lightning?! Do you mean the intersection of the FOVs of the three cameras (which may be approximated by rectangular light columns)? But I assume the LED lightning beams are larger than the FOVs (and not exactly rectangular and change in cross section).

7)
Be consistent with units.

102 you state the FOV as 17cm and 12.5cm, later you write 170mm and 125mm.

8)
101-103, the text referring to Fig 2 describes light reflected and scattered by snow particles, which is not shown in Fig 2.

9)
Fig 2: You have talked about rectangular columns earlier; you should now show these (rather than the circular columns you do show).
"Optical structure" doesn't seem to be the appropriate expression. Some more info in the caption would be useful.

10)
114-115, "which leads to a difference in the method of performing 3D reconstruction later in Sec. 4". Are you referring to a difference of the method". How is the method different from what?

11)
127 "The LED light sources are arranged in a parallel configuration, leading to a unidirectional power supply interface."
Are you talking about electrical set-up or spatial placement of LEDs?
Are you talking about LEDs or LED arrays?
What do you mean with "unidirectional power supply interface"?

12)
128: specify (or re-word) "consistent light output" as it is unclear.

13)
Fig 4a is not needed, you have the same information as Fig 4b.

14)
139-140: "projection matrix KMi of the transformation relationship between the 3D spatial points and each pixel plane pixel point in the world coordinate system"
a) Re-formulate this for clarity.
What is a "pixel plane pixel point" if it is not a mistake?
A pixel point is not "in the world coordinate system".
b) Also, define the World Coordinate System when you first use it.

15)
142-144: Something is missing or wrong in this sentence (in particular " and the apparent 3D …").

16)
a) 153-154: Define/describe the 3D checkerboard.
b) I would be consistently using only "checkerboard" or only "chessboard".
c) Reconsider sentence: "from the same localization using three cameras" is wrong. The three cameras image from three different views/locations.

17)
157 "such as Eq. (2)": should that be "shown in Eq. (2)"? (Eq (2) is not an example but shows exactly the equations you want)

18)
170-171, "The two ... define a common ...WCS"
Two planes do not define a WCS. Should it be "The three..."?

19)
Eq.s (3)-(5): "Dmax [um]" should be "Dmax [mm]"

20)
Fig 6a and 6b: You show 13 spheres in Fig 6a and 15 points on the plot Fig 6b. How many spheres did you use?

21)
209, "is necessary is an essential step" Check and correct.

22)
239, "which poses a challenge for particle identification from the images captured by three cameras"
Did you mean a "challenge for particle matching"?

23)
Comparison between 3D-PPI and OTT measurements
Fig 16, What are the numbers on the colour scale?
Would be interesting to see speed distribution for a few particle sizes (you mentioned such a distribution in 402 "snowflake velocity distribution with diameter was calculated").

24)
343-344, "PSDs are described ..."
a) What is the meaning of this sentence?
b) "across a larger range of sizes"?

25)
344, "The peaks of Deq..."
Deq does not peak. Be more precise and correct with your formulations.

26)
a) 348-352, "Comparison of temporal plots..."
Long sentence with several statements that are unclear.
b) Same next sentence (meaning of "which means the aggregation of snowflakes was weakened"). Please revise.

27)
Eq. (8): What is surface area S? Surface area of the 3D reconstructed hull?

28)
381, "so blurring that particle motion is insignificant" Check sentence for correct English.

29)
383, "and the same particle is merged into a single image in Fig. 15a"
Describe better what was merged into a single image (and that it is done only to better visualize something in the paper/Fig).

30)
In Sect. 6 Conclusion:
446, "pre-calibration" is used only once so that it is unclear what it refers to (add reference to section in paper and use consistent names).
Revise the whole Section after revising the manuscript.

31)
Comparisons without clear reference:
342, "across a larger range of sizes"
345, "more and more concentrated small particles"
345-346, "average particle size was consistently smaller" smaller than? (also unclear what "consistently" means)
391, "generally not more than 20%." What is 20%?
435-436, "estimate PSD more accurately"
436, "calculation of velocity more accurately"

---

## Author Comment (AC1)

**An introduction of Three-Dimensional Precipitation Particles Imager (3D-PPI)**
**Response to the reviewers**

Jiayi Shi, Xichuan Liu, Lei Liu, Liying Liu, Peng Wang

*Original Referee comments are in italic*

manuscript text is indented, with added text underlined and

We would like to thank the reviewers for their very helpful comments. We revised the manuscript thoroughly and responded to all of the reviewers' comments.

**General Comments:**

*1.How robust are the camera alignment and 3D measurements to thermal expansion shifting the cameras (since the cameras are enclosed in a housing, I assume the effects of wind on camera alignment will be negligible)? Adding a brief discussion of how these might affect the measurements would be helpful. Some tests of how much measurement error is produced from slight changes in camera alignment after the calibration could also be illuminating.*

Thanks for your comments. To mitigate the effects of wind on camera alignment, we have designed a protective housing of 3D-PPI that effectively shields the cameras from wind disturbances. Additionally, to further address potential temperature-related impacts, we have added a semiconductor air conditioner in the housing of 3D-PPI. This will help maintain a stable temperature around the cameras and minimize any thermal expansion effects. We have added a brief discussion in the revised manuscript.

Line 182:

Even small movements of the high-resolution camera can alter the projection matrix. This requires the instrument to be more robust. To mitigate the effects of wind on camera alignment, the instrument housing has been specifically designed for stability. Additionally, a semiconductor air conditioner has been installed in the housing, which will prevent minor camera expansion caused by temperature fluctuations.

Line 460:

However, even minor displacements of the high-resolution cameras can alter the projection matrix, which may adversely affect the subsequent reconstruction results.

Therefore, it is essential to perform periodic calibration to ensure the accuracy and reliability of the projection matrix.

*2.Given that the sampling volume is so close to the instrument housing, how will wind flow affect the representativeness of the measurements (e.g., increasing or decreasing the collection efficiency of the sampling volume)? While I understand that a full engineering analysis of the wind effects (similar to what was presented in Fig 3 of Newman et al. 2009, https://doi.org/10.1175/2008JTECHA1148.1) may be outside the scope of this paper, some analysis that compares the 3D-PPI to the PARSEVAL with the wind coming from certain directions relative to the camera pointing direction might help give some insight into how well the instrument handles these conditions.*

Thanks for your comments. We have added the wind field simulation in the Appendix C.

Appendix C: Wind field simulation

To determine the optimal orientation of the 3D-PPI installation (mainly considering the relationship with the prevailing wind direction), we conducted wind field simulations using Solid flow simulation software. The simulation results are shown in Fig. B.1.

When the 3D-PPI is facing the wind (Fig.B.1a), the observation volume experiences an average wind speed of approximately 6.0 m/s. Besides, turbulence may occur within the observation volume. When the 3D-PPI is back facing the wind (Fig. C.1b), the average wind speed in the observation volume is only about 3.5 m/s, which is obviously due to the shielding of the wind by the instrument. When the 3D-PPI is side facing the wind (Fig. B.1c), the observation volume shows an average wind speed of about 8.5 m/s, exhibiting the smallest difference from 10m/s, compared to the other two situations. However, part of the observation volume close to the instrument is still shielded by the housing, which to some extent also affects the representativeness of the wind field, and subsequent consideration will need to be given to improving the instrument design to solve this problem.

In addition to 10m/s, we also simulated 5m/s, 20m/s, and 40m/s wind speed fields, and all of them got the consistent conclusion that the wind speed in the observation volume is closest to the simulated wind speed when the instrument is side facing the prevailing wind direction. Therefore, the instrument should be installed sideways to the dominant wind direction in the area, to minimize the disturbance of the instrument to the natural wind field. The prevailing wind direction in the area is west, so the 3D-PPI is installed facing towards the south.

[Figure]

Figure C.1. Top view of wind speed distribution in the simulated wind field. 3D-PPI facing 10 m/s wind (a); back facing 10 m/s wind (b); side facing 10 m/s wind (c). The color gradient represents wind speed, with the observation volume indicated in (a).

*3.Since LEDs rapidly flash to produce light, it would be worth clarifying how the LED interacts with the cameras. Is the 20-microsecond effective exposure time due to the duration of the LED flash being 20 microseconds? Is there some mechanism to ensure that the LED is synchronized with the camera exposures in some way to ensure each frame has consistent illumination (or is the system sufficiently robust that inconsistencies in LED illumination are not a problem)?*

Thank you for your comments. To clarify, our LEDs remain continuously illuminated once triggered, eliminating the concern about a specific exposure time related to the LED flash. Therefore, there is no defined exposure duration for the LEDs. Furthermore, we have proposed an adaptive thresholding method in our image processing algorithms, it help us to mitigate variations in illumination and ensures robust data analysis. We appreciate your attention to detail, and we have added this clarification in the revised manuscript Line 135:

> Once triggered, the LEDs will continue to illuminate, providing consistent lighting throughout the exposure period.

*4.How are particles at the edge of the field of view handled? I seem to recall reading somewhere in the manuscript that particles touching the edge are ignored (but I can't seem to find where that was now, sorry if I'm mistaken). If the edge particles are discarded, I think the effective sampling volume would need to be a function of particle size (with the sampling volume being smaller for larger particles).*

Thank you for your comments. We indeed discard the particles at the edge of the field of view, we have added the following paragraph in the revised manuscript Line 227:

> If the connected region of a particle contains points located at the edges of the image, the particle is considered not to be fully captured, and it should be discarded.

We recognize that discarding particles at the edges of the field of view does influence the effective sampling volume, so we modified Eq. (7) and Eq. (8) for the calculation of the average PSD as follows:

Where PSD ($D_i$) ($mm^{-1} \cdot m^{-3}$) is the number concentration of particles per unit volume per unit size interval $\Delta D_i$ for snowflake size $D_i$ (mm); $n_{ij}$ is the number of snowflakes within size bin $i$ and velocity bin $j$; $T$ (s) is the sampling time (60 s in this study), and $V_j$ (m/s) is the falling speed for velocity bin $j$; $S$ ($m^2$) is the effective sampling area (0.18 m×0.03 m).

The time-averaged PSD calculated from 3D-PPI counting over a specified period is as follows:

$$PSD(D_i) = \frac{N_i}{\triangle D_i \cdot \sum_{j=1}^{N_{ima}} V_j} \tag{7}$$

Where $N_i$ is the number of particles in the $i^{th}$ size bin; $N_{ima}$ is the number of acquired images over a period of time; The size descriptor $D$ for 3D-PPI is $D_{max}$ or $D_{eq}$ in this paper; $V_j$ ($m^3$) is the valid OV of the Cam0 at the $j^{th}$ moment. Since the we discard particles at the edges of the image in Sec. 4.1, $V_j$ is a function of the average particle size $\overline{D_j}$ at the $j^{th}$ moment, shown in Eq. (8). The $a$, $b$, and $d$ represent the length (0.17m), width (0.125m), and depth (0.1043m) of field of view respectively.

$$V_j = \begin{cases} \left(a - \overline{D_j}\right) \times b \times d & \text{,when the particles at left or right edges discarded;} \\ a \times \left(b - \overline{D_j}\right) \times d & \text{,when the particles at top or bottom edges discarded;} \\ \left(a - \overline{D_j}\right) \times \left(b - \overline{D_j}\right) \times d & \text{,both conditions exists at the same time;} \\ a \times b \times d & \text{, no edge particles are discarded.} \end{cases} \tag{8}$$

*5. Is the tracking software intended to work for rain drops as well as frozen precipitation? It is clear from the manuscript that the instrument is designed for measuring snowflakes, it might be worth noting in section 5.3 whether or not the tracking software is designed to handle any faster falling precipitation.*

Yes, the tracking software is primarily designed to measure precipitation particles including raindrops as well as frozen precipitation. The sampling area of high-speed camera is 72mm (width) × 54mm (height), and the sampling rate is 200 fps. For a raindrop with diameter 5 mm, the maximum speed that can be measured according to two consecutive images between 5ms is 8.8m/s.

$$\frac{(54 - 5 - 5)\,mm}{5\,ms} = 8.8\,m/s$$

*6. I have some concerns with the tracking algorithm, although it is possible these concerns are due to misunderstanding on my part and/or a need for further clarification in the manuscript. When first reading over the tracking method, I was somewhat concerned that particle motion wasn't being considered as, in my personal experience, that is the best way to match an existing particle to its new location (although this isn't possible in the first two frames of the particle and some other criteria must be applied). After having studied the description more closely, it sounds like there might be some shape matching going on. I base this on Line 393, where the pixel coordinates of a particle are mentioned. Are these pixel coordinates being used to match the shape of the particle (e.g., the spatial distribution of these pixels relative to some particle centroid is compared between frames via some method)? If so, this warrants more discussion in the manuscript. If not, how are these pixel positions being used (or am I misunderstanding what is meant by pixel coordinates)? Given that the position*

*difference allows for up to 8 m/s of particle motion, I would expect that the other criteria need to be very robust to avoid mismatches.*

Thank you for your comments. To clarify, our tracking algorithm relies on matching the same particle across consecutive frames based on their size ($D_{max}$ and $D_{eq}$), and pixel coordinates of the centroid of particle contours in the image (including horizontal and vertical coordinates). Our approach focuses on extracting the centroid pixel coordinates of the particles, as well as $D_{max}$ and $D_{eq}$. Considering that the differences in centroid pixel coordinates of the same particle between consecutive frames, as well as $D_{max}$ and $D_{eq}$, do not vary significantly, the possible mismatch particles can be filtered out, as stated in the revised manuscript Line 411.

We sincerely apologize for the mistakes in our manuscript. We only extracted the centroid pixel coordinates of the particle contours to perform matching and calculations based on centroid pixel coordinates changes, rather than using the pixel locations. Additionally, we found that the upper limit of 8 m/s was too high, therefore, we have revised it to 4 m/s to reduce the likelihood of matching errors. We have modified the corresponding sentence in the revised manuscript Line 408:

> (ii) the pixel coordinates of the centroid of the contours of the same particle in consecutive frames are similar (the vertical velocity of the snowflake is generally not more than 4m/s, so the  distance between the vertical pixel coordinates of the same particle centroid in two adjacent image frames  is not more than 100 pixels);

**Specific Comments:**

*1.Title: suggest adding "the" before "Three-Dimensional"*

Changed as suggested.

*2.Line 16: change "OTT a good agreement" to "OTT have good agreement"*

Changed as suggested.

*3.Table 1: Thank you for including this table! Having these technical specifications in one place is very useful.*

Thanks for your positive feedback!

*4.Lines 107 – 111: These numbers don't seem to agree with those in Table 1. Specifically, the telecentric lens pixel size is listed as 3.45 microns in the table and 42 microns in the paragraph. Similarly, the table lists the non-telecentric lens as having a pixel size of 6.9 microns while the text lists 265 microns. It's very possible that these are referring to two different measurements and if that is the case, I encourage the authors to make that clear either in the text or in the table caption.*

We apologize for the unclear explanation in the manuscript, which may have led to

some misunderstandings. The pixel size of the telecentric lens (non-telecentric lens) is 3.45 (6.9) microns, the magnification rate is 0.083 (0.026), the corresponding field of view size is 41.5 (265) microns. We have rephrased in the revised manuscript Line 116:

For the high-resolution camera, the single pixel size is 3.45 μm, and the magnification of the lens is 0.083, meaning that the pixel resolution is 41.2 μm·px$^{-1}$. Telecentric lens distortion is 0.044% and allowed to be ignored. For the high-speed camera, the single pixel size is 3.45 μm, and the magnification is 0.026 at a working distance of 450mm, meaning that the pixel resolution is 265.4 μm·px$^{-1}$.

5.Lines 177 – 179: the parenthetical "(Dmax is the distance between the two largest points of the particle profile…" could use rewording. Perhaps replacing "largest" with "farthest" would improve the readability?

Thank you for your advice, "largest" is indeed inappropriate and caused misunderstandings. Changed as suggested.

6.Line 180: add "the" before "spheres"

Thank you for your advice, changed as suggested.

7.Figure 6: Move the first line of the caption to below panel b.

Thank you for your advice, changed as suggested.

8.Lines 208 – 210: These sentences could use some clarification. Are these particle centers as seen from a single camera or are these the particle centers from multiple cameras? If these are from the same camera, it might help clarify things to mention that sometimes a single particle will appear as two particles due to being on the edge of the image processing threshold (or whatever reason is appropriate). Just based on the flow of the manuscript, I assume these are for a single camera, but it might help make things clearer to specify that.

Thank you for your advice. These particle centers are indeed seen from a single camera. The image processing algorithms in section 4.1 were all about images from a single camera. We have modified the sentences in the revised manuscript, Line 222:

Secondly, combine regions into a single particle when the centroids of connected regions in a single image are detected to be less than 4 mm apart. This step is necessary because a single particle may sometimes be perceived as two separate particles due to its position near the edge of the image processing threshold.

9.Line 224: Are the ceramic spheres the same as the ones used in section 3.2? If yes, perhaps changing the text from "different diameters in section 3.2 were dropped" to "different diameters, as described in section 3.2, were dropped" would make that clear.

Thank you for your advice, we did use the same ceramic spheres as in section 3.2. Changed as suggested.

10.Line 227 – 234: The text mentions that the measurements of the smaller spheres tend

*to be larger than the true size, but the average error of Deq for the small spheres is negative. I may be missing something, but wouldn't a negative error mean the measurements of the small spheres are smaller than their true value? Also, the authors state that this is the average absolute error, which to me implies that it is the mean of the absolute value of the difference between the measured and true Deq and, therefore, should not be negative. I assume the authors are using "absolute" error in contrast with "relative" error. I'm not sure how best to fix this misunderstanding though.*

Thank you for your comments and sorry for our mistakes. The "absolute error" actually refers to the "error". The error is defined as: the measured value - the true value, where a positive value indicates that the size of the particle tends to be overestimated, and a negative value indicates that it tends to be underestimated. Similarly, the relative error is calculated as: (measured value - true value) / true value.

We have modified the sentences in the revised manuscript Line 245-249.

> Regarding the Dmax measurement results (Fig. 8a), smaller spheres (8 mm and below) tend to show measurements that are slightly greater than the true values, while larger particles exhibit measurements that are slightly lower than the true values. The average error for all spheres across different diameters is -0.048 mm, and the average relative error is +2.2%. As for Deq measurement results (Fig. 8b), all diameter measurements underestimate the true values. The average error for all spheres is -0.33 mm, and the average relative error is -2.7%.

*11. Figure 8: Are these plots for the high-resolution camera or the high-speed camera? It might be helpful to mention which in the caption.*

Thank you for your advice, these plots are for the high-resolution cameras. Changes are as follows:

> **Figure 8**  The average values of measurements of $D_{max}$ **(a)** and $D_{eq}$ **(b)** for ceramic spheres of different diameters from three high-resolution cameras.

*12. Lines 389 and 401: Line 389 mentions that the frames need to be adjacent to one another (which I interpret to mean that a missing frame is not allowed), but Line 401 mentions that a missing frame is allowed. This seems like a contradiction, but I suspect I'm just misinterpreting the authors' meaning. Some additional clarification in the text might be needed.*

Thank you for your advice. We would like to clarify our previous statements from the perspective of the processing algorithm flow. The process begins with high-speed image processing, where all particles in the images are detected. A particle can only be considered the same if it appears in adjacent frames; it is not possible for a particle to be captured in one frame, not seen in the next few, and then reappear later. This is an obvious point that I failed to state clearly, which may have led to some misunderstandings. Perhaps I should consider removing the first principle stated in Line 403 to avoid any confusion. Thank you for your understanding.

*13. Lines 390 – 391: Is the 200 pixels interval purely in the vertical or is that include the horizontal component of the distance as well. If that includes the horizontal distance, it might be worth noting that this means the instrument will have difficulty producing accurate particle fall speeds at high cross-camera-view wind speeds (which isn't a problem, but is good to know for anyone performing an analysis of the data).*

Thank you for your comments. The 200 pixels interval represents a purely vertical speed measurement. The horizontal velocity of snowflakes is heavily influenced by wind speed, which can exceed 10 m/s, while the vertical velocity typically does not exceed 4 m/s. We appreciate your reminder regarding the 8 m/s speed; indeed, it is somewhat higher. We have reformulated that statement to reflect that the maximum vertical speed is 4 m/s:

> (ii) the pixel coordinates in the particle images are similar (the falling velocity of the snowflake is generally not more than 4m/s, so the interval between neighboring snowflakes is not more than 100 pixels);

*14. Lines 416 – 417: If westward motion is positive, I assume the 3D-PPI was facing towards the south? It might be useful to mention the direction the instrument is facing in the paragraph starting on Line 324 (preferably as a bearing, but a general direction would do if you don't know the exact bearing)*

Yes, your assumption is correct; the instrument was indeed facing towards the south. We appreciate your attention to detail and have added the description about the instrument orientation in the revised manuscript Line 340.

*15. Lines 426 – 428: Are these outliers real or are they a result of mismatches by the tracking algorithm. If they're a result of mismatches, it might be helpful to provide some statistics regarding how prevalent these outlier are (e.g., what percentage of the total sample population or the a reasonable range of percentages of their size bin populations) so the reader can determine if they are sufficiently infrequent to be considered negligible.*

Thank you for your advice. We apologize for the error in the velocity calculations, which led to the mistakes in Figure 16. It has been corrected, the modified Figure 16 are shown as follows:

[Figure]

The modified Figure 16 shows a significant reduction in outliers, only less than 3% of the total samples are identified as anomalous particles caused by incorrect matching,

we believe that less than 3% is negligible.

*16.Lines 434: Ensure the 41.5 microns per pixel matches what is said in the text (Lines 107 – 111) and in Table 1.*

Thank you for your advice. The 3.45 microns in Table 1 refers to the size of a single pixel, while the resolution of high-resolution camera in Lines 434 is the actual object size that occupies only one pixel in the image, that is the smallest particle size can theoretically be measured is 3.45/0.083=41.5 microns.

To avoid the misunderstanding, we have modified the corresponding content in Lines 116- 119 and Table 1 in the revised manuscript.

**Table 1:** Technical specifications of the cameras.

|  |  | High-resolution camera | High-speed camera |
|---|---|---|---|
| Image sensor | Type | CMOS, Global shutter | CMOS, Global shutter |
|  | Model | Sony IMX304 | Sony IMX287 |
|  | Single pixel size [μm] | 3.45 × 3.45 | 6.9 × 6.9 |
|  | Resolution [px] | 4096×3000 | 720×540 |
|  | Pixel resolution [μm·px$^{-1}$] | 41.6 | 265.4 (450mm distance) |
|  | Size of the field of view (a × b) [mm] | 170 × 125 | 191 × 143 (450mm distance) |
|  | Frame rate [fps] | 5 | 200 |
|  | Effective exposure time [μs] | 20 | 20 |
| Lens | Type | Telecentric lenes | 25mm non-telecentric lenes |
|  | Aperture | F6.5 | F2.4 to F16 |
|  | Magnification of lens | 0.083 (constant) | 0.026 (450mm distance) |
|  | Distortion | 0.044% | 0.16% |

---

## Author Comment (AC2)

**An introduction of the Three-Dimensional Precipitation Particles Imager (3D-PPI)**

**Response to the reviewers**

Jiayi Shi, Xichuan Liu, Lei Liu, Liying Liu, Peng Wang

*Original Referee comments are in italic*

manuscript text is indented, with added text underlined and

We would like to thank the reviewers for their very helpful comments. We revised the manuscript thoroughly and responded to all of the reviewers' comments.

**Important points in the order they appear in the manuscript:**

*1) Previous instruments are presented such as the VISSS and the MASC. When describing the MASC the following statement is done (73-75): "Nevertheless, only 10^2–10^4 particles were observed during a typical snowfall event (Gergely and Garrett, 2016), which is insufficient to permit the reliable estimation of the particle size distribution (PSD) (Gergely and Garrett, 2016)." I don't see why this sample size would be insufficient and I don't see Gergely and Garrett, 2016 saying that. It is a bit misleading to look at sampled number of particles during specific storms rather than snowfall rates or snow number concentrations. The authors should instead look at the observation volume or something similar if they want to put the MASC in relation to 3D-PPI. The MASC has three cameras, as the 3D-PPI, but they are all located in one plane. However, the MASC was extended to five cameras with the additional two not being in the same horizontal plane as the original three cameras (Notaros et al. 2016, Kleinkort et al. 2017).*

Thank you for your comments. We apologize for our mistakes. The reference we cited does not mention the relevant information that we stated. We did intend to compare 3D-PPI to MASC in terms of the particle capture efficiency. The sampling area of MASC is 2.5 cm² with a frame rate of 2 fps, while the 3D-PPI has a larger sampling (observation) volume of 1464 cm³ (due to the use of a telecentric lens) and a higher frame rate of 5 fps.

The MASC captured 10,000 images during nearly 2 months, excluding out-of-focus particles, which were captured at very few particles per minute. On the contrary, one of the advantages of 3D-PPI is that in just 13 hours we recorded over 880,000 snowflakes. Therefore, for 3D-PPI, it is sufficient to reliably estimate a PSD on the minute temporal scales needed to capture changes in precipitation particle properties. Additionally, we

are not comparing the two instruments in terms of camera position design here. We have modified the sentences in the revised manuscript (72-77).

 Incorporating the rich dataset from the Multi-Angle Snowflake Camera (MASC), the 3D-GAN model is adeptly trained to reconstruct the intricate three-dimensional architecture of snowflakes, thereby unlocking new dimensions in the study of snowfall microphysics (Leinonen et al.,2021). Furthermore, the MASCDB, is a comprehensive database of images, descriptors, and microphysical properties of individual snowflakes in free fall, as presented by, showcases the MASC's exceptional potential for contributing to the field of atmospheric science by providing an extensive and detailed resource for studying the microphysical properties of snowflakes (Grazioli et al.,2022).

*2) a) What is the actual resolution (combination of optical resolution and illumination that allows details of a certain size to be resolved)? What are the smallest details that can be resolved? This is not discussed but an important detail of a new instrument. I can only guess from the images (mostly Fig. 1c). The images look similar to MASC images of snow, likely due to a similar illumination scheme. The smallest details are very faintly grey and seem to disappear in the black background. Due to this and due to the missing description of the detection algorithm and thresholds (see further comment on image processing (comment 8) below), it is not possible to do a fair judgment of the actual resolution for detection of small details (e.g. thin branches).*

Thank you for your comments. The pixel resolution of 3D-PPI is 41.6 $\mu m \cdot px^{-1}$, which theoretically means that the smallest recognizable details (e.g. thin branches) of a large snowflake is 0.0416mm. The actual resolution is influenced by both the optical resolution and the illumination scheme. The illumination provided by the LED array is designed to enhance the contrast and visibility of snowflakes, but it can also affect the perception of smaller details, particularly those that are faint against a dark background. The faint gray details you observed in Fig. 1c may indeed be challenging to discern, which can complicate the assessment of small features like thin branches. We are currently refining our detection algorithm and thresholds of image processing algorithms to improve the identification of these subtle details.

*b) Judging from Fig 6b, the sizes of the smallest ceramic spheres are underestimated. For each camera, the linear fit has a negative offset (see comment below) with the observed size being on or below that fit. So, there seems to be a systematic bias for sizing of the smallest details. Please comment on this.*

Thank you for your comments. We acknowledge that the reflectivity of ceramic spheres differs from that of snowflakes. The smooth surface of the ceramic spheres can lead to variations in brightness across different areas in the ceramic spheres' image. Some regions appear very bright, while others may appear quite dark. The darker areas are more susceptible to being misidentified as background, which can contribute to an underestimation of their sizes. Before the linear fitting in Fig. 6b, we performed manual

adjustments to mitigate the software's misidentifications (illustrated in the following figure). However, this systematic bias in the sizing of the smallest details remains a challenge.

[Figure]

**The image of ceramic spheres**     **Detect by software**     **Detect manually**

In contrast, when capturing real snowflakes with rough surfaces, their diffuse reflection may result in a more uniform brightness distribution on a single snowflake image, Therefore, the systematic bias of the real snowflakes is expected to be much lower than that of the ceramic spheres.

*c) Related to this is in 207 "Snowflakes that are too small in diameter are ignored". What is this diameter, how did you decide on its value? Also, in the next sentence you state that you connect apparently separate detected regions if they are up to 4mm apart. This indicates that you expect that smaller details that may be connecting these regions are not being detected.*

Thank you for your comments. Diameter means $D_{max}$, the detected connected regions in binarized images with the $D_{max}$ less than 20 pixels (equivalent to approximately 0.035 mm², $D_{max}$ is about 0.2mm) are ignored. This threshold is the optimal value chosen through testing. The value cannot be too large, or more snowflakes will be ignored; nor can it be too small, or some small noise spots will be recognized as snowflakes mistakenly. For the second problem, the apparently separate detected regions should be connected firstly if they are up to 4mm apart, and then discard particles that are too small. We have modified the corresponding text in the revised manuscript (221-227).

(ii) Particles detection. Firstly, detect the connected regions in binarized images. Secondly, combine regions into a single particle when the centers of connected regions in a single image are detected to be less than 4 mm apart. This step is necessary because a single particle may sometimes be perceived as two separate particles due to its position near the edge of the image processing threshold. Thirdly, discarding the particles with an area greater than 20 pixels (Equivalent to 0.035 mm², $D_{max}$ is about 0.2mm) enables the removal of small noises from the image, to prevent these noises from being mistakenly identified as small snowflakes.

*3) Inconsistent and wrong or confusing use of "resolution", "pixel resolution", "pixel size", and "magnification". Revise and use better and consistent terminology throughout the paper.*

Thank you for your advice. We have modified the terminology in the revised manuscript as follows: resolution: 4096×3000 or 720×540; pixel resolution: 41.6 or 265.4 µm·px−1; pixel size: 3.45µm × 3.45µm or 6.9µm × 6.9µm; magnification: 0.083 or 0.026. These have been clearly specified in Table 1 of the revised manuscript.

*4) 102, "observation volume size of 1505.327cm3": Part of the observation volume is out of focus (as depth of focus with 104mm is smaller than the FOV dimensions). Three digits after the decimal point are not needed.*

Thank you for your advice. The effective observation volume (OV) of 3D-PPI should be the intersection of the OVs of three high-resolution cameras, which is smaller than the OV of a single camera (170mm ×125mm × 104mm). The observation volume has been recalculated, which is 1464 cm$^3$. We have modified the corresponding descriptions in the revised manuscript (104-108).

To clarify, the observation volume (OV) of one high-resolution camera is the interior rectangle of the observation volume of one telecentric lens with three dimensions a × b × d (170mm × 125mm × 104mm), which represents the length, width, and depth of field of view respectively. The intersection of three OVs of three high-resolution cameras forms the effective OV of 3D-PPI, the volume size is 1464 cm$^3$. Only the particles falling within the effective OV of 3D-PPI can be simultaneously captured by the three high-resolution cameras.

*5) 156 "super-determined": Does this refer to an overdetermined system with more equations than unknowns? What are the unknowns here in that case? Do "super-determined" and "super-deterministic" refer to the same thing?*

Sorry for our mistakes. Both the "super-determined" and "super-deterministic" refer to the overdetermined linear system, where the number of equations exceeds the number of unknowns. In Eq. (2), the other two matrices are known except $KM_i$, which is an unknown $2 \times 4$ matrix. The number of columns of the two known matrices is the number of chosen checkerboard corner points and is much more than 4. Therefore, Eq. (2) solving $KM_i$ is equivalent to solving two overdetermined linear systems with the number of equations larger than the number of unknowns. The unknowns of the two equations are the first and second rows of $KM_i$ respectively. We have modified the corresponding sentences in the revised manuscript (175-176).

As can be seen from Fig. 5b, the value of $J$ is much larger than 4, so Eq. (2) is equivalent to overdetermined linear systems. Further, the least squares method is used to optimally estimate the projection matrices ($KM$) for each high-resolution camera.

*6) 163-169, explanations why the planar two-dimensional checkerboard grid cannot be used for calibration:*

*a) 164-165, "but then the values of all three-dimensional world coordinate points are the same": this sounds very general; explain better what you mean, which points/coordinates are the same?*

What we mean is that there is only one WCS. We have deleted this sentence to avoid confusion.

*b) The sentence continues that with that A equals a matrix, which is the same matrix as the matrix KMi is multiplied with in Eq (2).*

The matrix *A* has been deleted.

*c) The matrix A has not been defined before. Please do that.*

The matrix *A* has been deleted.

*d) 165-167, "The third and fourth-row values are the same ..." This sentence is not correct. Please revise. The fourth-row values are all 1, did you want to say that all values in the third row are 1, or are the same but not necessarily 1 but any other value?*

Sorry for our mistakes. We intended to state that the third-row values are the same. Furthermore, since the fourth row consists entirely of 1s, if the values in the third row are all the same, then the third and fourth rows are linearly dependent, which means the matrix *A* is impossible to invert. We have replaced it with a simpler statement, shown in the next reply.

*e) 167, "determinant of A is 0": as far as I know, the determinant is defined for square matrices, but A is not (unless j=4). f) 168, "impossible to inverse" should be "impossible to invert"*

Thank you for your advice. Actually, *A* is not a square matrix and has no determinant. Since the third and fourth rows are linearly related, the rank of *A* is 3. Changed as suggested. To be understood clearly, we have simplified the descriptions in the revised manuscript (176-178), which can also answer the above comments a) to e).

> It is worth noting that during the solution (optimal estimation) it is important to make sure that no row of $W_J$ (except the row with value 1) can be the same value. In other words, the selected points cannot be in the same plane in the WCS. That is why we select a 3D checkerboard rather than a 2D checkerboard.

*7) 171-172, in this sentence at the end of Sect 3.1, you present an "average reprojection error". Nowhere in Sect 3.1 you present any calibration results related to the theoretical treatment presented. The theory presented seems needed to determine KMi, which are needed for the matching algorithm. Define what the average reprojection error is and how you have determined it. Here you say 0.32 pixels, in the Conclusion 0.4 pixels.*

Thank you for your comments and sorry for our mistakes. The average reprojection error is a key metric used to evaluate the accuracy of our estimated projection matrix. It is calculated as the mean distance between the observed 2D image points and their corresponding projected points obtained from the estimated projection matrix. This error indicates how well our model captures the actual image data. The calibration result referred to here is the estimated projection matrix *KM*. We have modified and improved this section in the revised manuscript (179-181).

To assess the accuracy of the estimated projection matrices (*KM*) of three cameras, we calculated the average reprojection error, which is calculated as the mean distance between the identified 2D image points (uj, vj) and their corresponding projected points $KM \cdot (X_{wj}, Y_{wj}, Z_{wj})^T$. The  results show that the average reprojection error for three high-resolution cameras is 0.32 pixels.

Besides, we have changed to 0.32 pixels in the conclusion of the revised manuscript.

*8) Determination of pixel resolution (41.5um/px) in Sect 3.2 (Calibration of image binarization) and image processing in Sect. 4.1:*

*a) 181, "is optimally binarized manually": How has this manual binarization been performed. Is each image treated differently? Is this using the image processing algorithm described in Sect. 4 (where the details of the detection are missing, see comment about "adaptive thresholding below")?*

Thanks for your comments. The meaning of manual binarization has been illustrated in the previous figure in 2b). We performed manual adjustments to mitigate the obvious software misidentifications. Each image was treated differently. The image processing algorithm mentioned in Sect. 4 is not used here. In this part, accurate pixel resolution is obtained by capturing ceramic spheres and linear fitting, so we need to treat each image differently and perform perfect processing. In other words, since the perfect binarization result of each ceramic sphere in the image is circular theoretically, optimal manual binarization is to make the ceramic sphere in the image more circular, to avoid missing detection details.

*b) You determine the pixel resolution from the reciprocals of the slopes in Fig 6b (Dmax/px vs Dmax/mm). You are not discussing the role of the offset of between 0.5px to 4px in Equations (3)-(5).*

Thank you for your comments. The non-zero intercepts observed in the linear fits (0.5 px to 4 px) might be attributed to several factors: systematic errors introduced during the calibration process due to lens distortion or misalignment; image processing artifacts from the binarization method that may affect edge detection. Additionally, the pixel resolution of the cameras and the physical characteristics of the ceramic spheres, such as surface texture, can also cause these offsets. We have added these discussions in the revised manuscript (211-214).

*c) The pixel resolution should also result from the calibrations using chessboards. Has this been attempted and the values compared to the pixel resolution reported in Sect 3.2?*

Thank you for your comments. To clarify, the word "calibration" in Sec. 3.1 and Sec. 3.2 has different purposes. Sec.3.1 is only for estimating the projection matrix, which contains the internal and external parameters of the camera, and we are not estimating the full range of internal and external parameters, as having the projection matrix is sufficient for the 3D reconstruction later on. Sec.3.2 is for estimating more accurately the pixel resolution, which is a crucial internal parameter.

The reasons for not using the checkerboard grid to obtain image resolution are as follows: pixel resolution can only be estimated when the checkerboard plane is parallel to the camera image plane, which is difficult to guarantee. The advantage of the ceramic sphere in comparison is that its image taken at any angle is round.

*d) The pixel resolution using chessboards would be similar to using micrometer or millimeter scales. The advantage over ceramic spheres would be that it would not depend on image processing (selection of specific grey level threshold to detect the contours of the ceramic spheres), which may result in under- or over-sizing of the spheres. The details of this image processing are, however, not disclosed. You only state (Sect. 4.1, 204) that the images are "binarized through adaptive thresholding" without giving any reference or explaining what adaptive thresholding is in your specific case. Please explain this method.*

Thank you for your comments. According to your comments 8) a) and our response, the checkerboard was used to estimate the projection matrix, rather than using micrometer or millimeter scales. Regarding the adaptive threshold method, we have added the following description in the revised manuscript (219-221).

> Adaptive thresholding dynamically calculates the threshold for smaller regions of the image, allowing for better handling of varying lighting conditions. This method enhances the accuracy of foreground particle detection, particularly in images with complex backgrounds and uneven illumination.

*e) In Sect. 4.1 describing the image processing you show results from imaging the ceramic spheres (Fig. 8a and 8b) in terms of measured size vs real size (both in mm). You must have used the pixel resolution resulting from imaging the ceramic spheres (Sect. 3.2, Fig 6b). For the "calibration of image binarization" you must have used the image processing algorithms described in Sect. 4.1. That means that Fig 8 does not show anything new or independent from previously reported Fig 6b. You are presenting the same ceramic sphere measurements in two related ways. Consequently, the error analysis related to Fig 8 is only a re-interpretation of the error analysis of Fig 6b.*

Thank you for your comments. As explained above, in Sec.3.2 we did not use the image processing algorithm in Sec. 4.1, the measurements were processed by optimal manual binarization, and the section was only intended to get a more accurate pixel resolution. Instead, the measurements in Fig.8 were obtained by batch processing using the image processing algorithm mentioned in Sec.4.1. The purpose of Fig. 8 is to evaluate the reliability of the image processing algorithm as its error is mainly caused by the image processing algorithm.

*f) In any case, the presented error analyses are unclear and confusing: L196-197, "The estimated random errors from the normalized root square errors, derived from the observed and true size difference …" Unclear what "observed and true size difference" is. Unclear what "random" refers to? Does it refer to variations in 20 observations of the size of the same ceramic sphere?*

Thank you for your comments and sorry for our unclear descriptions. Actually, no error

analysis is needed for this section, we just want to estimate the pixel resolution of each high-resolution camera by the linear fitting.

We removed the text:

*g) Due to the offset in your linear least squares fits, the reciprocals of the coefficient px/mm (24px/mm here) are not equal to the coefficients that would result from fitting true size vs observed size (mm/px). Redo that fit or comment.*

Thank you for your comments. Actually, the inverse of the slope of the fitted straight line is very close to the pixel resolution after unit conversion, as follows:

$$\frac{1}{24\,\text{px/mm}} = \frac{1000\,\mu\text{m}}{24\,\text{px}} \approx 41.67\,\mu\text{m} \cdot \text{px}^{-1}$$

*h) 227, "average absolute error of Dmax measurements for all diameters of small spheres is -0.048mm," An absolute error is positive, but you state that it is -0.048mm. Same for Deq with - 0.33mm. Then it is unclear what the "average relative error" (229, 230) is.*

Thank you for your comments and sorry for our mistakes. The "absolute error" actually refers to the "error". The error is defined as: the measured value - the true value, where a positive value indicates that the size of the particle tends to be overestimated, and a negative value indicates that it tends to be underestimated. Similarly, the relative error is calculated as: (measured value - true value) / true value.

*9) 206-207, "Detecting connected regions ... enables the removal of small noise ..." Missing/wrong logic: What is enabling the removal of noise from image? What is small noise on the image?*

Thank you for your comments. Actually, there might be individual noise points or artifacts in the binarized image that were misidentified as particles. By connecting the regions within an area greater than 20 pixels (equivalent to approximately 0.035 mm², $D_{max}$ is about 0.2mm), we can effectively filter out these small noises and the particles smaller than 0.2 mm. Considering that the 3D shape reconstruction of snowflakes mainly focuses on larger particles, the above noise removal process does not have a significant impact on the observations.

*10) 207, "Snowflakes that are too small in diameter are ignored" What is this diameter, how did you decide on its value? This is related to the comment about actual resolution. Here you seem to have defined a smallest particle that is accepted.*

Thank you for your comments. The diameter refers to $D_{max}$, the detected connected regions in binarized images with the $D_{max}$ less than 20 pixels (equivalent to

approximately 0.035 mm²) are ignored. This value is determined through pre-testing. It cannot be too large, or more snowflakes will be ignored; nor can it be too small, or some small noise spots will be mistakenly recognized as snowflakes.

Actually, the smallest acceptable particle $D_{max}$ is 0.2mm does not contradict that the pixel resolution of 3D-PPI is 41.6 µm·px$^{-1}$. The pixel resolution means that the smallest recognizable detail of a large snowflake is 0.0416mm, rather than the smallest recognizable snowflake.

*11) 243-252, explanation of matching algorithm: This is a clever way to use the matrices KMi (i=1,2,3) from camera calibration for matching. The following issues/unclarities result from wrong language or unclear formulation.*

*a) I would not re-use the index i, that previously was used to refer to the three cameras, to refer now to a particle number. Use another index or no index.*

Thank you for your advice. The index *i* was removed in the revised manuscript, because we then modified the manuscript to describe the matching process for only one particle.

*b) 245, "i underdetermined linear equations": Is there one underdetermined linear equation for each particle? Refer to an equation number in the paper or explain better what these equations are (and why/how underdetermined).*

Thank you for your comments. Actually, there is one underdetermined linear equation for each particle centroid. In Eq. (2), when $J = 1$, $KM$ and $C_1$ are known and $W_1$ is solved for, i.e., 3D points are solved from 2D points, this is equivalent to solving the underdetermined equation, which does not have unique solutions.

*c) Would be good to add equation (if appropriate in Appendix), the solution of which is Li. Then also another equation showing the multiplication of KM1 and KM2 to produce the projection of Li onto Cam1 and Cam2 image planes.*

Thank you for your comments. We have supplemented Appendix B with a more detailed description of the equations:

**Appendix B: Additional equation solving process**

To solve for $(X_w, Y_w, Z_w)$ based on the known points $P(u,v)$ and $KM_0$, simplify Eq. (2) by removing 1 from the second term:

$$KM_0 \cdot \begin{bmatrix} X_w \\ Y_w \\ Z_w \\ 1 \end{bmatrix} = \begin{bmatrix} u \\ v \end{bmatrix} \xrightarrow{\text{Simplify}} A \cdot \begin{bmatrix} X_w \\ Y_w \\ Z_w \end{bmatrix} = B \qquad (B.1)$$

Where $A$ is a known $2 \times 3$ matrix and $B$ is a known $2 \times 1$ matrix, so it is equivalent to underdetermined linear equations. And the solution is not unique and shown in Eq. (C.2):

$$\begin{bmatrix} X_w \\ Y_w \\ Z_w \end{bmatrix} = U t + V = \begin{bmatrix} U_1 t + V_1 \\ U_2 t + V_2 \\ U_3 t + V_3 \end{bmatrix} \qquad (B.2)$$

Where, both $U$ and $V$ are $3 \times 1$ matrixes, $t$ is all real number. Therefore, all solutions form a straight line L in 3D space WCS. In other words, this process implements the back-projection of P onto the line L.

Furtherly, project the L onto the planes of Cam1 and Cam2 by multiplying the projection matrices $KM_1$ and $KM_2$, respectively, shown in Eq. (C.3):

$$L_{p1} = KM_1 \cdot \begin{bmatrix} U_1 t + V_1 \\ U_2 t + V_2 \\ U_3 t + V_3 \\ 1 \end{bmatrix} = \begin{bmatrix} f_1(t) \\ g_1(t) \end{bmatrix} \qquad L_{p2} = KM_2 \cdot \begin{bmatrix} U_1 t + V_1 \\ U_2 t + V_2 \\ U_3 t + V_3 \\ 1 \end{bmatrix} = \begin{bmatrix} f_2(t) \\ g_2(t) \end{bmatrix} \qquad (B.3)$$

Where, $L_{p1}$ and $L_{p2}$ denote the point sets of projections of L onto the Cam1 and Cam2 planes respectively. The functions $f_1(t)$, $g_1(t)$, $f_2(t)$, and $g_2(t)$ are all linear functions of $t$. $L_{p1}$ and $L_{p2}$. Therefore, $L_{p1}$ and $L_{p2}$ represent straight lines in the plane. Determine the range of $t$ to ensure that the line is within the image range (4096×3000) to get the corresponding line segments.

*d) 246, "ith straight lines" should be "i straight lines"?*

Thank you for your comments. The index *"i"* has been removed in the revised manuscript.

*e) 247, "the lines Li is the back-projection" should be "the lines Li are the back-projections".*

Thank you for your advice. We have modified as suggested in the revised manuscript.

*f) 248, "multiplying the projection matrices KM1 and KM2," should be "multiplying the projection matrices KM2 and KM3, respectively, with Li,"*

Thank you for your advice. We have modified as suggested in the revised manuscript.

*g) 249, "resulting in ith line segments" should be "resulting in i line segments on each of the image planes of Cam1 and Cam2"*

Thank you for your advice. We have modified as suggested in the revised manuscript.

*h) 251-252: from the text it is unclear if each particle has to be on all three lines, or each particle on its respective line in each of the images. Describe for one particle (to avoid confusion between THREE cameras, and THREE particles/lines).*

Thank you for your advice. It refers to each particle on the respective line in each of the images. We have described one particle matching process to avoid confusion.

To be understood clearly, we have revised the descriptions in the manuscript, which can also answer the above comments a) to g).

The revised text is as follows (259-273):

For one particle, the matching algorithm is implemented in detail as follows (Fig. 9):

(i) Detect the centroid coordinates of the particle P ($u$, $v$) in the PCS from Cam0 (Fig. 10a).

(ii) Using the projection matrix $KM_0$ of Cam0, the underdetermined linear equations corresponding to P are solved to obtain a straight line L in WCS. Specifically, it is equivalent to solving $W_1$ in Eq. (2) with $KM$ and $C_1$ known, where the solution to $W_1$ is not unique and all solutions are the L in the WCS. The L represents all points in 3D space that can be projected onto P by $KM_0$, in other words, the line L is the back-projection of the point P in WCS.

(iii) Project the L onto the planes of Cam1 and Cam2 by multiplying the projection matrices $KM_1$ and $KM_2$, respectively, with L, resulting in the line segments on each of the image planes of Cam1 and Cam2 (Fig. 10b, c). The exact derivation of the formulae in (ii) and (iii) are described in detail in Appendix B.

(iv) Detect the particles that the line segments pass through on Cam1 and Cam2 respectively. If the line segments do not pass through any particles in Cam1 or Cam2, it is a failed matching, meaning that the particle does not appear in the effective OV.

By performing the above matching for each particle detected by cam0, the location of this particle in Cam1 and Cam2 can be found. Fig. 10 shows the three particles detected in Cam0 and the matching of each particle in Cam1 and Cam2.

*12) 259-294, 3D reconstruction: The description of how particle is located to enable/simplify 3D reconstruction is unclear. Here a few things that make it difficult to follow:*

*a) Add new sub-section (now it is part of 4.2 Particle matching and localization).*

Thank you for your advice. We have modified it as suggested in the revised manuscript.

*b) 259, Not sure what voxels are and why/how they are used traditionally. Provide a reference.*

Thank you for your comments and sorry for our mistakes. Our method is not comparable to traditional methods, so we decided to remove this sentence.

*c) 260-261, Explain or discuss how are "traditional methods" "computationally inefficient".*

Sorry for our mistake. Our method is not comparable to traditional methods, so we decided to remove this sentence.

*e) 264, "particles' localizations" should be "particles' locations"?*

Thank you for your advice. We have modified it as suggested in the revised manuscript.

*f) 266-271, Unclear why irregular particles pose a problem. You probably need to be*

*more specific in explaining the issue.*

Thank you for your comments. We were originally trying to state that the localization is relatively straightforward for particles with regular geometries, on the contrary, it is somewhat more complex and requires the methods described below for complex snowflakes. However, we later found out that it is not true. Therefore, this sentence was removed in the revised manuscript.

We removed the text:

~~For particles with regular geometries, back-projection of the contour centroids suffices to determine the centroid of the 3D reconstructed object. However, snowflakes often exhibit highly irregular shapes. The center of the three-dimensional body projected onto the contour may not coincide with the center of the two-dimensional contour, rendering the straightforward intersection of back-projected lines inapplicable and complicating the reconstruction process.~~

*g) 277-278, confusing use of indices: "Lines L1", I thought L1 is one line, the back-projected line from P1 on image of Cam0. "L2" according to earlier explanations (247), this is the back-projected line from P2 (particle 2) on image of Cam0, but here is redefined.*

Thank you for your comments and sorry for our mistakes. We have modified the previous text, $L_1$, $L_2$, $P_1$, $P_2$ are no longer existed. So, in this section, we still use these symbols.

*h) How can L2, a line, be "represented as a 2-row by 1-column matrix"? 279-280, "P2" and "P2'", what is P2 (it is not the P2 on image of Cam0, 243)? It becomes more and more tedious to follow the remaining explanations. This whole section should be revised for consistency and clarity.*

Thank you for your advice. We have modified the section in the revised manuscript (280-303):

After completing camera calibration and particle matching, 3D reconstruction for each particle needs to be performed. However, since each particle only occupies a small space in the effective OV, we propose a method that involves preliminarily locating particles in WCS before proceeding with subsequent 3D reconstruction. This method leverages the positions of single particles in three images to identify the minimal cuboid capable of containing the particles, thereby accurately pinpointing the particles' localizations.

For a single particle, the pixel coordinates of the centroid point of the particle in Cam0, Cam1, and Cam2, respectively, have been detected and the subsequent 3D spatial localization steps in the WCS are as follows (Fig. 11):

(i) Find the back-projection line $L_1$ of point $P_1$ by $KM_0$. The underdetermined linear equation corresponding to $P_1$ is solved to obtain a straight line $L_1$ in WCS. This implementation principle is similar to the second step of particle matching

mentioned above. Eq. (B.1) and Eq. (B.2) in Appendix B explain this process.

(ii) The lines $L_1$ are projected onto the planes of Cam1 by multiplying the projection matrices KM1, resulting in line segment $L_2$, which is represented as a 2-row by 1-column matrix. This corresponds to the segment of the Lp1 within the image boundaries as described in Eq. (B.3).

(iii) Find the point $P_2'$ on $L_2$ that is closest to $P_2$. Due to the irregular shape of the particle, $P_2'$ does not necessarily coincide with $P_2$.

(iv) Following the same approach as in step 1, determine the back-projection line L3 of point $P_2'$ by $KM_1$.

(v) Localize the 3D coordinates of the intersection of $L_1$ and $L_3$ in the WCS, that is Pc, which is the centroid of the target cuboid, and further determine the side lengths of the cuboid. From the previous steps, $L_1$ and $L_3$ are destined to intersect in the WCS, and the intersection point is regarded as the centroid of the rectangle, whose side lengths can be determined by converting from the pixel dimensions in the particle image to the actual physical dimensions in the WCS.

(vi) Finally, verify that the projection of the Pc point through $KM_2$ in Cam2 is near the P3 point and within the particle contour, otherwise, it is a failed localization.

The particle's position in the WCS should be inside the region of the cuboid determined by localization, which will next be discretized into numerous smaller voxel grids to finely perform 3D reconstruction.

*13) Sect 4.3: It is not immediately clear if and how this connects to explanations in 259-294. 296-297: "silhouettes that have been serially calibrated using multiple viewpoints around the target" The section starts with unclear formulations like this one (are silhouettes contours, multiple viewpoints refer to Cam0,1,2, what does "serially calibrated mean"?). I did not review the remaining of this section. After revising the previous section, this section should be revised too.*

Thank you for your constructive comments. The Visual Hull (VH) method is employed for reconstructing the 3D shape of snowflakes by utilizing silhouettes—defined here as the contour outlines of the snowflakes as viewed from various perspectives. Multiple viewpoints, such as cameras positioned at angles Cam0, Cam1, and Cam2, are used to capture the silhouettes. Each silhouette is carefully calibrated to ensure accuracy in the reconstruction process. The term "serially calibrated" refers to the process of calibrating the camera parameters and positions for each viewpoint.

We have revised this part (307-311):

 The Visual Hull (VH) method is a technique used to reconstruct the three-dimensional shape of snowflakes by utilizing silhouettes, which are the

outlines of the snowflakes as seen from different camera angles. In this process, multiple cameras are positioned around the snowflake at various viewpoints (Cam0, Cam1, and Cam2 in 3D-PPI). Each camera captures the silhouette of the snowflake, and these silhouettes are carefully calibrated to ensure that they accurately represent the snowflake's geometry.

*14)*

*a) Explain better Eq (7).*

*b) "Nima is the number of particles" should be "Nima is the number of acquired images"?*

Thank you for your advice. We have modified it as suggested in the revised manuscript.

*c) You state that (339) "the probability of capturing the same snowflake in two consecutive frames is very low." How often can the same snowflake be captured twice? Discuss if anything is done to account for this or, if not, how big of an error this may cause.*

Thank you for your advice. Since the high-resolution camera samples at a frame rate of 5fps (with a time interval $\Delta t$=0.2s), particles cannot be captured twice when the horizontal velocity of the particle $v_h > 0.85$m/s (0.17m / 0.2s) or the vertical velocity $v_v > 0.625$m/s (0.125m / 0.2s).

When $v_h < 0.85$m/s and $v_v < 0.625$m/s, the probability that the particle is caught twice in the horizontal and vertical directions are respectively:

$$\begin{cases} \text{Prob}_h = 1 - \dfrac{\Delta t \cdot v_h}{0.17} \\ \text{Prob}_v = 1 - \dfrac{\Delta t \cdot v_v}{0.125} \end{cases}$$

Horizontal and vertical capture probabilities are 50% when $v_h$ and $v_v$ of the particles are approximately 0.4 m/s and 0.3 m/s respectively. Particles with small velocities have a higher probability of being captured more than twice, which is indeed an important factor contributing to the inaccuracy of PSD measurements. Subsequently, we will consider increasing $\Delta t$ (decreasing the frame rate of the high-resolution camera) to alleviate this problem.

*d) You are getting the PSDs only from Cam0 here. That means $V_{observation}$ is different from the observation volume reported in 102 (1505cm3). $V_{observation}$ is calculated using the depth of focus (104mm). How well-defined is the depth of focus? Are particles outside depth of focus detected but then rejected reliably?*

Thank you for your comments. In the revised manuscript, $V_{observation}$ refers to the volume of OV, which is the interior rectangle of the observed volume of the telecentric lens, with three dimensions a × b × d (170mm × 125mm × 104mm), representing the length, width, and depth of field of view respectively. To clarify, the "104mm" refers to the

depth of field rather than the depth of focus, referring to the range that can be clearly imaged perpendicular to the direction of the lens. It has been tested that particles outside the depth of field cannot be clearly imaged and therefore cannot be detected.

*15) From Fig16b I would guess that the PSD peaks at Deq between 1 and 2 mm, whereas in Fig 13 the PSD peaks at Deq of about 0.4mm. Is everything consistent here and I am only being confused and misled by something?*

Thank you for your comments. We would like to clarify that Fig.16 represents the particle velocity distribution measured on April 6, while Fig.13 shows measurements taken on March 28 and 29. The differing periods account for the variations in the observed PSD peak values.

We apologize for any confusion caused by this discrepancy. Unfortunately, due to data storage limitations during the field experiments, we were unable to retain high-resolution camera images and high-speed camera images from the same period for direct comparison. We will continue to conduct the field experiments in the next winter and analyze the observation data, thank you for your understanding, .

*16) Negative velocities:*

*a) L416-417 "The average value of the horizontal velocity component measured by 3D-PPI is +0.05m/s (positive and negative values indicate westward and eastward velocities, respectively), and the standard deviation is 2.56m/s (Fig. 16a)." Unclear if your average is an absolute average (or do you consider positive and negative values when averaging)? Consequently, the meaning of the standard deviation is unclear.*

Thank you for your comments. In calculating these statistics, both positive and negative values were taken into account, so they are not absolute averages. The average velocity and standard deviation values have been updated to +0.41 m/s and 0.73 m/s in the revised manuscript (432-433).

*b) I would expect horizontal speed to be characterized by speed (absolute value) and direction (0 – 360degrees). Instead, you use positive and negative values to indicate westward and eastward. Why choosing to give info on west-east, and not north-south or the actual direction in degrees?*

Thank you for your comments. The 3D-PPI instrument was installed with its camera facing south, side facing the prevailing west wind direction. As a result, the measurements from this orientation primarily capture the east-west horizontal velocity component, represented by positive values for westward velocity and negative values for eastward velocity. We acknowledge that a single high-speed camera cannot measure speeds in all directions from 0 to 360 degrees.

*c) If you choose to include directional information, why are you not analyzing the direction. I would expect the horizontal speed direction to correlate with local wind direction. Was the local wind measured and compared to horizontal speed? This could be part of a discussion how wind affects measurements.*

Thank you for your comments. Actually, the horizontal speed direction has a high correlation with the local wind direction. Unfortunately, we have not collected the local wind observation data during this study, we acknowledge the significance of wind effects on measurements and will consider this in future observations.

**Minor issues**

*1) The three cameras are numbered 0,1,2 (Cam0, …), whereas you use the indices i=1,2,3 to indicate the three cameras (e.g. KM1, KM2, KM3). Can you use the same indices to reduce confusion?*

Thank you for your advice. We have mapped the three camera projection matrices $KM_0$, $KM_1$ and $KM_2$ to the camera numbers (Cam0, Cam1 and Cam2) to reduce confusion.

*2) "and" in wrong place in a list (85, 185).*

Thank you for your advice. Changed as suggested.

*3) 86-87, "capacitive rain sensor is adopted as a trigger, the cameras only work when the precipitation occurs." Rain sensors detect rain. When you say the cameras only work when precipitation occurs, do you mean when it rains? I would expect you want to measure with snow but perhaps not with rain? Please clarify.*

Thank you for your comments. To clarify the point in Lines 86-87, we would like to emphasize that the capacitive rain sensor is designed to trigger the operation of the 3D-PPI instrument whenever precipitation occurs, regardless of whether it is rain or snow. The sensor detects any moisture on the instrument's surface, and its heating element ensures that snow is melted and then sensed. We have added the following description in the revised manuscript (88-90).

> To improve the instrument's working efficiency, a capacitive rain sensor is adopted as a trigger, the cameras only work when the precipitation occurs. The sensor detects any moisture on the instrument's surface, and its heating element ensures that snow is melted and then sensed.

*4) Fig 1a: It would be useful to label the different parts. Without labels it is, for example, not clear which/where is the fourth non-telecentric camera. Labels are a complement to more clarity in the text.*

Thank you for your advice, we have added the labels in Figure 1.

[Figure]

Thank you for your advice, we have revised the corresponding sentences in 100-102:

> The high-speed camera is positioned horizontally at the center, while  the Cam1 and Cam2 are positioned, at 45 degrees, on either side of the high-speed camera in the same horizontal plane, and the Cam0 is positioned at 45 degrees vertically above (Fig. 1a).

*6) 100-101, "overlapping region of the LED lighting beams" and intersection of the "three rectangular light columns"??? I don't think you are talking about the LED lightning?! Do you mean the intersection of the FOVs of the three cameras (which may be approximated by rectangular light columns)? But I assume the LED lightning beams are larger than the FOVs (and not exactly rectangular and change in cross section).*

Sorry for our unclear descriptions. To clarify, the observation volume (OV) of one high-resolution camera is the interior rectangle of the observation volume of one telecentric lens with three dimensions a × b × d (170mm × 125mm × 104mm), which represent the length, width, and depth of field of view respectively. The intersection of three OVs of three high-resolution cameras forms the effective OV of 3D-PPI, the volume size is 1464 cm$^3$.   The high-brightness LED array light sources are situated on the same side as the cameras to illuminate the observation volume.   We have revised the corresponding sentences in revised manuscript (102-107):

> The high-brightness LED array light sources are situated on the same side as the cameras to illuminate the observation volume. The cylindrical observation volume of the three telecentric lenses and LED lighting beams of 3D-PPI is illustrated in Fig. 2. To clarify, the observation volume (OV) of one high-resolution camera is the interior rectangle of the observation volume of one telecentric lens with three

dimensions a × b × d (170mm × 125mm × 104mm), which represent the length, width, and depth of field of view respectively. The intersection of three OVs of three high-resolution cameras forms the effective OV of 3D-PPI, the volume size is 1464 cm$^3$.

*7) Be consistent with units. 102 you state the FOV as 17cm and 12.5cm, later you write 170mm and 125mm.*

Thank you for your advice. Changed as suggested.

*8) 101-103, the text referring to Fig 2 describes light reflected and scattered by snow particles, which is not shown in Fig 2.*

Thank you for your advice. Changed as suggested in revised manuscript (103-104):

The cylindrical observation volume of the three telecentric lenses and LED lighting beams of 3D-PPI is illustrated in Fig. 2.

*9) Fig 2: You have talked about rectangular columns earlier; you should now show these (rather than the circular columns you do show). "Optical structure" doesn't seem to be the appropriate expression. Some more info in the caption would be useful.*

Thank you for your comments. We have modified the sentences in revised manuscript (110):

Figure 2. The observation volume of the three telecentric lenses and LED lighting beams.

*10) 114-115, "which leads to a difference in the method of performing 3D reconstruction later in Sec. 4". Are you referring to a difference of the method". How is the method different from what?*

Thank you for your comments. We would like to clarify that the different imaging schemes will result in slight variations in the methods used for 3D reconstruction. We have modified the sentences in revised manuscript (120-123):

Unlike non-telecentric lenses, which produce larger images of nearby objects and smaller images of distant objects (Fig. 3a), telecentric lenses are based on the principle of parallel light imaging, resulting in identical objects at different distances from the lens having the same size in the image (Fig. 3c). This difference in imaging scheme will lead to distinct methods for three-dimensional reconstruction (Fig. 3b, d), which will be discussed in detail in Sec. 4.

*11) 127 "The LED light sources are arranged in a parallel configuration, leading to a unidirectional power supply interface." Are you talking about electrical set-up or spatial placement of LEDs? Are you talking about LEDs or LED arrays? What do you mean with "unidirectional power supply interface"?*

Thank you for your comments.

**Electrical Setup vs. Spatial Placement:** We are referring to the electrical setup of the

LED light sources, which are arranged in a parallel configuration. This configuration allows for a single power supply interface to power the LEDs.

**LEDs vs. LED Arrays:** We are specifically discussing LED arrays, where multiple LEDs are grouped together for improved illumination.

**Unidirectional Power Supply Interface:** The term "unidirectional power supply interface" refers to a single power connection that supplies power to all LEDs in the parallel configuration, ensuring that the power is distributed uniformly across the array.

*12) 128: specify (or re-word) "consistent light output" as it is unclear.*

Thank you for your comments. To clarify, we have rephrased this to specify that it refers to a stable and uniform light intensity produced by the LEDs. We have modified the sentences as follows in response to 11) and 12) in revised manuscript (135-139):

>
>
> The LED light sources are configured in parallel, allowing for a single power supply connection that distributes power to the entire array. Each LED can operate in either constant current mode or trigger mode. In constant current mode, the LEDs provide stable and uniform light intensity, which enhances the uniformity of illumination within the OVs.

*13) Fig 4a is not needed, you have the same information as Fig 4b.*

Thank you for your advice. Changed as suggested.

*14) 139-140: "projection matrix KMi of the transformation relationship between the 3D spatial points and each pixel plane pixel point in the world coordinate system"*

*a) Re-formulate this for clarity. What is a "pixel plane pixel point" if it is not a mistake? A pixel point is not "in the world coordinate system".*

Thank you for your comments. We have rephrased in revised manuscript (147-151):

> The geometric model for telecentric lens imaging is described in detail in Appendix A, where four coordinate systems (WCS, CCS, ICS, and PCS) are defined and their transformations are derived. The purpose of camera calibration in this section is to estimate the projection matrix  ,$KM_0$, $KM_1$, and $KM_2$ for three high-resolution cameras, which enables the coordinates transformation of 3D spatial points in the WCS to their corresponding projection points in the PCS.

*b) Also, define the World Coordinate System when you first use it.*

Thank you for your advice. In the previous sentence, we have already mentioned that

WCS has been defined in Appendix A.

*15) 142-144: Something is missing or wrong in this sentence (in particular "and the apparent 3D …").*

Thank you for your comments. Considering that the 3D checkerboard has been introduced later, this sentence has been deleted

>

*16) a) 153-154: Define/describe the 3D checkerboard.*

Thank you for your advice. The 3D checkerboard has been defined in the revised manuscript.

*b) I would be consistently using only "checkerboard" or only "chessboard".*

Thank you for your comments. It should be "checkerboard" rather than "chessboard". It has been changed to 'checkerboard' throughout the manuscript.

*c) Reconsider sentence: "from the same localization using three cameras" is wrong. The three cameras image from three different views/locations.*

Thank you for your advice. We have rephrased as follows in the revised manuscript (161-167):

> To obtain the projection matrices, the following steps are executed:  Firstly, make a 3D checkerboard and establish the WCS. Attach three high-precision 2D checkerboards to three mutually perpendicular flat boards to form a 3D checkerboard. The three plane intersections are used as the WCS origin O, and the two plane intersections are used as the X, Y, and Z axes respectively (Fig. 5b). The 3D checkerboard is placed in a position that defines the WCS. This position ensures that three high-resolution cameras can capture checkerboard corner points. Secondly, physically measure the precise coordinates of all checkerboard corner points in the WCS ($X_{wj}$, $Y_{wj}$, $Z_{wj}$) ($j$ denotes the number of corner points), and identify the pixel coordinates ($u_j$, $v_j$) of  $j^{th}$ corner points in the PCS of each camera image.

*17) 157 "such as Eq. (2)": should that be "shown in Eq. (2)"? (Eq (2) is not an example but shows exactly the equations you want)*

Thank you for your comments. Changed as suggested.

*18) 170-171, "The two … define a common …WCS" Two planes do not define a WCS. Should it be "The three…"?*

Thank you for your comments. This sentence has been removed in the revised

manuscript.

*19) Eq.s (3)-(5): "Dmax [um]" should be "Dmax [mm]"*

Thank you for your comments. Changed as suggested.

*20) Fig 6a and 6b: You show 13 spheres in Fig 6a and 15 points on the plot Fig 6b. How many spheres did you use?*

Sorry for our mistakes. We used a total of 15 spheres, we have made the necessary corrections in Fig. 6a.

[Figure]

(a)

*21) 209, "is necessary is an essential step" Check and correct.*

Thank you for your comments. Changed as suggested.

*22) 239, "which poses a challenge for particle identification from the images captured by three cameras" Did you mean a "challenge for particle matching"?*

Yes, we mean "challenge for particle matching" rather than identification. We have modified it in the revised manuscript.

*23) Comparison between 3D-PPI and OTT measurements Fig 16, What are the numbers on the colour scale? Would be interesting to see speed distribution for a few particle sizes (you mentioned such a distribution in 402 "snowflake velocity distribution with diameter was calculated").*

Thank you for your comments. The color scale denotes the number of snowflakes measured in the corresponding bins. (Sorry for the error in the previous Fig. 13, please refer to the new one.)

[Figure]

(a)  (b)  (c)

We have revised the corresponding description and added some necessary information

in revised manuscript (418-421):

From 0800 UTC to 0830 UTC on 6 April 2024, the Cam3 of 3D-PPI recorded  77042 valid snowflakes, and the snowflake velocity distribution with diameter was calculated. for each snowflake, horizontal and vertical velocities were calculated. The distributions of horizontal  and vertical velocity components as a function of $D_{eq}$ are further plotted as a scatter density plot and compared to the results measured by OTT at the same period, which is shown in Fig.16. The color scale denotes the number of snowflakes measured in the corresponding bins.

*24) 343-344, "PSDs are described …" a) What is the meaning of this sentence? b) "across a larger range of sizes"?*

Thank you for your comments. We try to convey that the PSD described by $D_{max}$ covers a broader range of particle sizes compared to that described by $D_{eq}$. However, using $D_{eq}$ to describe the PSD appears to be more valuable in terms of analysis. b) "across a larger range of sizes" means covering a broader range of particle sizes in the PSD plot for the horizontal coordinates. To avoid misunderstanding, we have removed this sentence in the revised manuscript.

*25) 344, "The peaks of Deq…" Deq does not peak. Be more precise and correct with your formulations.*

Thank you for your comments. To clarify, the highest number density of particles is found at $D_{eq}$ 0.4 mm for both chosen days. In the revised manuscript (359-360):

 The PSDs of particles with a $D_{eq}$ of about 0.4 mm were highest for both chosen days.

*26) a) 348-352, "Comparison of temporal plots…" Long sentence with several statements that are unclear. b) Same next sentence (meaning of "which means the aggregation of snowflakes was weakened"). Please revise.*

Thank you for your advice. We have revised this part in the revised manuscript (362-367).

~~Comparison of temporal plots, some periods (19:00 to 19:50, 20:50 to 22:00, and after 23:30 UTC on March 28; 01:30 to 04:30, and 06:00 to 07:59 on March 29) have fewer snowflakes per unit volume, while the average size is larger and the deviation of Deq and Dmax is larger, which may be a period when the snowflakes are sparsely distributed in space, with a high degree of aggregation of individual snowflakes and a more complex shape. On the contrary, in other periods (19:50 to 20:50 and 22:00 to 23:30 UTC on March 28; 00:00 to 01:30, and 04:30 to 06:00 on March 29), the particle counts per unit volume were smaller, while the average size and the deviation of Deq and Dmax was larger, which means the aggregation of snowflakes was weakened.~~

In comparing the temporal plots (Fig. 13c, d, e, f), certain periods (19:00 to 19:50, 20:50 to 22:00, and after 23:30 UTC on March 28; 01:30 to 04:30, and 06:00 to 07:59 on March 29) exhibited a smaller number of particles per unit volume, with larger average sizes and greater difference between $D_{eq}$ and $D_{max}$. This indicates a higher degree of aggregation and potentially more complex shapes of individual snowflakes during these times. Conversely, other periods (19:50 to 20:50 and 22:00 to 23:30 UTC on March 28; 00:00 to 01:30, and 04:30 to 06:00 on March 29) showed a larger number of particles per unit volume, smaller average sizes, and reduced difference between $D_{eq}$ and $D_{max}$.

*27) Eq. (8): What is surface area S? Surface area of the 3D reconstructed hull?*

Thank you for your comments. The surface area *S* indeed refers to the surface area of the 3D reconstructed hull of the observed particles. This value is calculated based on the 3D model generated during the reconstruction process. In the revised manuscript (377):

*Sp* is derived from the *V* and *S* (surface area of 3D reconstructed hull) and characterizes the to which 3D particles approach the sphere:

*28) 381, "so blurring that particle motion is insignificant" Check sentence for correct English.*

Sorry for our unclear description. We have revised the sentence in the revised manuscript (394-396):

 The exposure time for a single frame is 20 µs, which renders the motion blur of the particles negligible. Additionally, the time interval between two consecutive frames is 5 ms, allowing the same particle to be captured multiple times, thus enabling accurate velocity calculations.

*29) 383, "and the same particle is merged into a single image in Fig. 15a" Describe better what was merged into a single image (and that it is done only to better visualize something in the paper/Fig).*

Thank you for your comments. For better visualization of the particle movement, we have merged several consecutive frames of the same particle into a single image in Fig. 15a. This approach is only for improved clarity in illustrating the particle's motion. We have revised the corresponding sentence in the revised manuscript (396-397):

 The same particle from consecutive frames is merged into a single image in Fig. 15a to enhance the visualization of its movement. The speed calculation schematic is shown in Fig. 15b.

*30) In Sect. 6 Conclusion: 446, "pre-calibration" is used only once so that it is unclear what it refers to (add reference to section in paper and use consistent names). Revise the whole Section after revising the manuscript.*

Thank you for your comments. It refers to the camera calibration mentioned in section 3.1 rather than pre-calibration. Changed as suggested.

*31) Comparisons without clear reference: 342, "across a larger range of sizes" 345, "more and more concentrated small particles" 345-346, "average particle size was consistently smaller" smaller than? (also unclear what "consistently" means) 391, "generally not more than 20%." What is 20%? 435-436, "estimate PSD more accurately" 436, "calculation of velocity more accurately"*

Thank you for your comments.

342, "across a larger range of sizes" means PSDs described by $D_{max}$ cover a broader range of particle sizes compared to those described by Deq, which has been mentioned in 25). We have removed it in the revised manuscript.

345-346, The sentence "There were more and more concentrated small particles on March 28 compared to March 29 (Fig. 13a), and the average particle size was consistently smaller (Fig. 13e)." is not sufficiently clear and does not add substantial meaning to the discussion. Therefore, we have removed it in the revised manuscript.

391, "generally not more than 20%." We added some necessary explanations in the revised manuscript (405-407):

> Given that the size of the particles captured in two consecutive frames does not change significantly, the $D_{max}$ and $D_{eq}$ of the particles are similar, generally not more than 20% (the $D_{max}$ or $D_{eq}$ value of the particle in the next frame deviates within ±20% of the previous particle).

435-436, "estimate PSD more accurately". There is no comparison here, so we removed "more" in the revised manuscript.

436, "calculation of velocity more accurately". There is no comparison here, so we removed "more" in the revised manuscript.

---

## Referee Report (RR1)

Review of "An introduction of Three-Dimensional Precipitation Particles Imager (3D-PPI)"

Thank you to the reviewers for providing careful and detailed responses to my comments. In particularly, I was impressed with the addition of the wind simulations. Overall, I am satisfied with the responses with one exception: I still have a concern about the PSD calculation (see below). As such, I recommend minor revisions be made before accepting this manuscript for publication.

**Specific Comments:**

Equation 8: First off, I'm not an expert on the theory behind PSD calculations. That said, from what I understand, I think the image edge correction needs based on the PSD bin centers rather than for individual particles. Using the bin center in the edge correction would essentially be accounting for the potential volume in which a particle from that size bin could be observed rather than accounting for this volume difference only when a particle is observed. So the sample volume for a PSD bin centered on Di would be

$$v_i = (a - 2D_i) * (b - 2D_i) * d,$$

where v, a, b, and d are defined as per Equation 8 of the manuscript. The depth might also vary with particle size (I know it does with PIP), but exactly how this might vary with particle size is beyond my knowledge.

---

## Referee Report (RR2)

**Manuscript amt-2024-106**

**An introduction of Three-Dimensional Precipitation Particles Imager (3D-PPI)**

amt-2024-106-manuscript-version3

Thank you for carefully responding to all raised issues. The changes to the manuscript resolved many of these issues. However, a few questions remain or issues have not been resolved completely. In the following I am describing a few issues that I think still need attention.

**Observation volume OV and effective OV**

The effective OV is the intersection of the three individual OVs, which are 170mmx 125mm x 104mm. The intersection of the two OVs of Cam1 and Cam2 is a cuboid with the dimensions 104mm x 104mm x 125mm, i.e. it has the volume of 1352cm3. Intersecting this with the OV of Cam0 will result in the effective OV, which then must have a smaller volume. That means the volume that you provide with 1464cm3 is wrong. Alternatively, your description is wrong and the effective OV is not the intersection oif the individual OVs.

Rephrase: "the observation volume (OV) ... is the interior rectangle of the observation volume of ..."
-   'Interior rectangle' is wrong term to describe a volume
-   It is not clear to use 'observation volume' to describe 'observation volume'

Rephrase: "The cylindrical observation volume of the three telecentric lenses and LED lighting beams of 3D-PPI is illustrated in Fig. 2."
-   Does this refer to effective OV?
-   Fig 2 does not show any observation volume (shows beams)
-   No observation volume is cylindrical

**Optical resolution / resolving power / smallest resolved detail**

Thank you for adopting a better terminology. Pixel resolution and resolution (number of pixels of sensor) of the different cameras are important parameters in describing your instrument. I am missing another paramter describing the optical performance of your cameras, that is the resolving power achieved on images of Cam0-2 and on images of Cam3. You could determine it by calibration or by identifying the smallest features that can be detected on your images. I do not agree with a statement that the "smallest recognizable detail ... is 0.0416 mm" (in response to my point 9). Of course, this is theoretically the best limit, however, it is not backed up by evidence. Can you show

images of snow particles with small details to illustrate the smallest features that can be detected (under optimal illumination and in-focus position)?
You seem to focus on "larger particles" (response to my point 9), but you don't mention this clearly in the manuscript.

**Removal of noise and smaller particles**

You are rejecting features with Dmax less than about 0.2mm. I would recommend stating this limit in the Abstract/Conclusions (something like "...measure particles > 0.2 mm").

You call it "removal of small noises from image". I would rephrase this as I don't consider 'small noises' a proper term, it sounds colloquial.

Prior to rejecting small features, you join connected regions that have their centroids separated by less than 4mm. This still fells like a large distance to me. I don't understand why the centroid distance has been chosen here rather than the actual gap separating any two connected regions. The centeroid distnce depends on the size of the connected regions whereas the gap represents the size of a potentially undetected region of the particle. When joining small connected regions you accept a gap of almost 4mm. For larger connected regions, the accepted gap is smaller by roughly the average size of these connected regions. It would be good, after justifying better your joice of criterion, to show examples of particles resulting from joining connected regions. This could be done in Fig.7 by adding two or three such examples including a length scale for reference. A propoer description of what is shown now in the figure is missing (you indicate (in red numbers) detected particles, I think).

**Pixel resolution**

You determine the pixel resolution using ceramic spheres of known diameters, as described in Sect 3.2, which you call "Calibration of image binarization". I still find it unclear how you describe the process.
You say that "we perform manual adjustments to mitigate the software's misidentifification" sugesting that you use a software or algorithm. However, you state that the algorithm described in Sect 4 is not used. So, it is unclear what software or algorithm is used here.
I think, for the purpose of determining the pixel resolution, it would have been better, and easier to describe, to do a completely manual analysis of the images of the ceramic spheres, for example by adding best-fitting circles and determining their diameters in pixels.
I also still believe that taking images of a millimeter scale would have been better for the purpose. Aligning such a millimeter scale would not have been difficult with the required accuracy. Using a millimeter scale would have avoided the dependenc on the used algorithm and/or manual analysis (e.g. different thresholds result in over- or undersizing).

In fact, the use of spheres of know sizes is more suitable for testing an image processing algorithm. You actually do that in Sect 4.1 and Fig. 8. As a consequence, your Figures 6b) and 8 look almost identical, and it si difficult to see what difference there is between the two tests. In one figure you show error bars in the other not. What are the error bars in Fig 8? I guess they are related to that you image 20 spheres falling through in fornt of each camera. For Fig 6b, did you image one sphere for each diameter? Were these spheres dropped as for Fig 8?

For Fig.6b, I would suggest to plot the distance of two points (in mm) over the distance of these two points on the image in pixels (rather than the opposite as you are doing). Then you end up with fitting your data to the linear expression of the form $Dmax = d*Px + e$ rather than, as you do now, $Px = b*Dmax + c$. In these expressions Dmax represents the actual size or distance in mm and Px represents the measured distance on the image in pixels. The fitting coefficients are d and b, respectively, and the intercepts are e and c, respectively. The resulting value for d is the pixel resolution. Whereas you take the inverse of the slope b. It is, however not generally true that $d = 1/b$, where d and b result from fitting the same data to the above two expressions. So, do the fit of $Dmax = d*Px + e$ to your data in Sect 3.2 and then report d as pixel resolution. You likley get very similar results, but you avoid the confusion of how to comparer or convert your results properly to pixel resolution.

Fig 6b from one sphere (for each size), or the average of several images of spheres (at each size)?

**Image processing algorithm – resulting errors**

You use adaptive thresholding to binarize images. You also state (in the response to my point 14d) that "It has been tested that particles outside the depth of field cannot be clearly imaged and therefore cannot be detected." What does this statement mean? Are particles outside the DOF not properly binarized (because they are fuzzy, not in focus)? My question remains: How well-defined is the depth of field? The PSD values and OVs depend on it.

You state that "The average error for all spheres across different diameters is -0.048 mm." In your responses you explain that this error is "measured-true" and can be positive or negative. Taking the average of two errors where one is, for example, +5 and the other -5 would result in a zero average, which would wrongly describe the error. I think this is what happened when you report the very small "average error", it is a misleadingly small value, the average of larger positive and negative values. Consider another way to describe the errors and relative errors.

**Horizontal speed**

"The average value (consider positive and negative) of the horizontal velocity component..." What does that mean? Is this a suitabel average (see also error of image processing above).
It is not obviously clear that the horizontal speed corresponds to the East-West wind direction. Mention that installing the instrument facing south means that the horizontal speed seen by the high-speed camera Cam3 corrsiponds to East-West.

**Size-dependent OV in Eq. 7**

Excluding particles at the edge of the image means a reduction of OV that depends on particle size. You account for that in the modified eq (7). It now features $V_j$, the valid OV at the "jth moment". $V_j$ has already been used in eq (6) for speed, potentially leading to confusion. However, what confuses me more is this "jth moment", not sure what that means. Should that not be "in the jth image"?

---

## Referee Report (RR3)

**Comments in response to amt-2024-106-ATC3 and amt-2024-106-author\_response-version3**

Thank you for your careful responses and revisions of the manuscript. I have checked them and still found a few points I would like to highlight. In the following I will first point out a few issues that can be easily fixed followed by a few points that I think need further discussion and/or need more than an easy fix. I am referring to numbers in the revised manuscript (version4), in particular line numbers as, for example, "L100" for line 100 in that manuscript version.

**Easily fixed issues**

**Pu et al 2021**

The new reference to Pu et al. 2021 (L29) is about snow darkening and melting as a consequence of it. As such it is not suitable as reference for "importance of snowflake shapes for our understanding of atmospheric science.

**Fig 2**

The beams extending from the four cameras in Fig 2 are confusing. They are not mentioned neither in the sentence (L104) referring to Fig 2, nor in the caption of Fig 2. Fig 2 shows also the high-speed camera Cam3, apparently with a conical beam extending from it. That camera is not mentioned in L104.

For clarity, I would add labels in Fig 2 pointing to the three telecentric lenses, the highspeed camera or lens, and the LED light beams. If you still want to keep the additional beams extending from the lenses, then I would mention them somehow in the caption.

**Three dimensions of OV**

The sentence in L104-106 has several issues and should be rephrased. The "three dimensions ... is a x b x d" is grammatically and mathematically wrong. The three dimensions are a, b, and d, not a x b x d. The previous issue is better fixed with using the appropriate term instead of "interior rectangle". I think "cuboid" would be correct here (OV of one camera is a cuboid defined by the three dimensions ...).

**Da vs Dp**

Thank you for adopting this way of showing results of ceramic spheres. Remove sentence L209-210 which doesn't apply anymore.

**Size-dependent OV**

The new Eq (10) to calculate the effective OV as a function of snowflake size seems wrong to me. At each border or edge only D/2 needs to be removed. If a particle of size D is at least D/2 away from the border (distance between particle centre and border), then it will not touch it. So:  $Vi = (a-Di)(b-Di)^*d$

**Further discussions**

**REVISED Sec. 3.2 Estimation of pixel resolution (previously Calibration of image binarization) and NEW Sec. 3.3 Calibration of image binarization**

For me the estimation of pixel resolution comes too early (before image processing) and requires image binarization (so that that is done twice). Image binarization is optimized, not calibrated.

In your response you try to justify why spherical targets are better than planar micrometer scales, where there would be an issue with orienting them perfectly. The disadvantage of the spherical targets is the necessity to do a manual binarization in addition to the adaptive thresholding. The apparent size of spherical objects depends on any binarization. I believe that this is introduces a larger uncertainty than what would be related to imperfect alignment of a micrometer slide. Some more detailed comments below:

Image processing is described in separate sections (3 and 4):

The image binarization (now described in the new Sect 3.3) is for me not a "calibration" as the title of the subsection suggests (but the determination of optimal binarization). I would consider image binarization as one of the steps in image processing. Subsection 4.1 is now called "Image processing" and describes the steps of noise removal and segmentation.

It would be a clearer structure to first describe the complete image processing and only then results (pixel resolution) and algorithms that use this image processing. Sect 3.3 should be part of image processing (Sect 4.1).

For doing the above, you would need to do the binarization only once (with adaptive thresholding). Now it is done twice:

In Sect. 3.2, Dp is "counted manually" to determine pixel resolution, which means that a manual binarization is done. Then, in Sect 3.3, binarization is done again with adaptive thresholding. The "optimal" sensitivity coefficient c is effectively the coefficient where adaptive thresholding approximates closest the manual binarization from Sect 3.2.

(It is understandable and fine that you will not change from your spheres to anything else now. But consider the following:)

It would be more transparent to find "manually" the best sensitivity coefficient c (i.e. decide visually what looks to be a good binarization) and make this part of the description of image processing. Then the estimation of pixel resolution could be done with images processed according to that image processing. That would also avoid having twice similar analyses (related to 6b and 7c/d).

**MTF method to determine the depth of field**

I appreciate the details on how you determined the depth of field and that you checked and corrected it.

There are two things I didn't understand though:

- What are "different spatial frequencies"? This is probably a minor thing or me not understanding.
- How are snowflakes below the MTF threshold "deemed fuzzy and considered
- outside the depth of field"? You claim that, consequently, your "algorithm effectively excludes these particles from identification". How does this work? Snowflakes are detected/identified if they are binarized as a connected region of more than 20 pixels. What is the relation to the MTF threshold?

**Description of image binarization Sec. 3.3**

I appreciate the added details about the binarization method (adaptive thresholding). However, I have some questions and see a few issues:

- Could you cite some description of this method in the literature? Without that I think that I need some more information.
- L221 and Eq (6): what is the "local mean  $\mu(u,v)$ "? Mean of what? How is it calculated? What is the specified neighborhood? How is that adjusted by the sensitivity coefficient C?
- MRE defined by Eq (7): this seems to be the average of the two means of the absolute relative errors. Being based on absolute values it is always positive.
  - Later you refer to the "MRE of Dmax" (L243) and "MRE of Deq" (L244).
    According to Eq (7) there is only one MRE, which is based on both Dmax and Deq.
  - Then, you also refer to "relative errors" of Dmax and Deq. These are not absolute values but positive or negative. From Fig 7 I assume that they are determined as (Dmaxi-Dai)/Dai, which is different from what Eq (7) would suggest ((Dai-Dmaxi)/Dai).
  - If the worst relative error is -7% then it is strictly speaking wrong to call that the "maximum relative error", which would be +2% in case of Deq (Fig 7f).
- The definition of Dmax (L231) is different from the definition of Dmax later used in Sect. 5.2 (L397). The definition in L397 is the one I would expect here. I would call Dmax "maximum dimension" not "maximum size" as doen in Sect 5.2.

• The definition of Deq (L232) is linguistically wrong (a circle cannot be equal to an area).

**Pixel noise (L274-275)**

Referring to detected regions with less than 20 connected pixels as "pixel noise (no larger than 20 pixels)" is now clearer than "small noises" previously. It implies that all regions with less than 20 connected pixels are indeed noise, i.e. not related to actual snowflakes. I am not sure this is true in general. Could these "noise" features be caused by snowflakes that are outside the depth of field, or by small snowflakes that are too small to be detected by 3D-PPI? So, rather than and/or in addition to "prevent these noises from being mistakenly detected as small snowflakes" it should say "exclude features of small snowflakes that cannot be detected from analysis"?

**2-mm gap criterion**

Your new criterion for joining regions is better than the previous one. It, however, still allows that small regions would be joined across a gap that can be larger than these regions. A 20-pixel region has about 14 to 5 pixels across, and a 2-mm gap corresponds to almost 50 pixels. I.e., two 5-pixel regions could be joined even if they are separated by a about ten times larger gap. I think two such regions should rather be excluded. Would they indeed be belonging to the same snowflake, then that would mean that a large part of this snowflake would have been missed (not been detected by binarization) likely due to being out of depth of field.

---

## Author Response (AR2)

**An introduction of Three-Dimensional Precipitation Particles Imager (3D-PPI)**

**Response to the referees**

Jiayi Shi, Xichuan Liu, Lei Liu, Liying Liu, Peng Wang

**Response to Anonymous Referee #1**

*Original Referee comments are in italic*

   manuscript text is indented, with added text underlined and

Our responses are in regular font.

Thank you very much for your thorough review and insightful comments on our manuscript. We appreciate the time and effort you have dedicated to evaluating our work and your constructive feedback. Your suggestions have been invaluable in helping us improve the quality and clarity of our manuscript. Below, you will find our point-by-point responses to your comments, along with the revisions made to the manuscript.

**Regarding the PSD Calculation Concern:**

*Equation 8: First off, I'm not an expert on the theory behind PSD calculations. That said, from what I understand, I think the image edge correction needs based on the PSD bin centers rather than for individual particles. Using the bin center in the edge correction would essentially be accounting for the potential volume in which a particle from that size bin could be observed rather than accounting for this volume difference only when a particle is observed. So the sample volume for a PSD bin centered on Di would be*

$$V_i = (a - 2D_i) \cdot (b - 2D_i) \cdot d$$

*where v, a, b, and d are deffned as per Equation 8 of the manuscript. The depth might also vary with particle size (I know it does with PIP), but exactly how this might vary with particle size is beyond my knowledge.*

Thank you for your valuable advice about the calculation of Vi, particularly regarding the image edge correction. We have revised Eq. (8) in the manuscript to incorporate the bin center for the edge correction, as you suggested. Because of the previous addition of equations, Eq. 7 and 8 in the original manuscript correspond to Eq. 9 and 10 in the revised manuscript, as follows:

The time-averaged PSD calculated from 3D-PPI counting over a specified period is as follows:

$$PSD(D_i) = \frac{N_i}{\triangle D_i \cdot \sum_{j=1}^{N_{ima}} V_j} \quad PSD(D_i) = \frac{N_i}{\triangle D_i \cdot N_{ima} \cdot V_i} \qquad (7)\ \underline{(9)}$$

Where $N_i$ is the number of particles in the $i^{th}$ size bin; $N_{ima}$ is the number of acquired images over a period of time; The size descriptor $D$ for 3D-PPI is $D_{max}$ or $D_{eq}$ in this paper; $V_i$ (m³) is the valid OV of the Cam0 after edge correction. Since we discard particles at the edges of the image in Sec. 4.1, $V_i$ is a function of $D_i$, shown in Eq. (10). The $a$, $b$, and $d$ represent the length (0.17m), width (0.125m), and depth (0.0.088m) of the field of view respectively.

$$V_j = \begin{cases} \left(a - \overline{D_j}\right) \times b \times d & \text{,when the particles at left or right edges discarded;} \\ a \times \left(b - \overline{D_j}\right) \times d & \text{,when the particles at top or bottom edges discarded;} \\ \left(a - \overline{D_j}\right) \times \left(b - \overline{D_j}\right) \times d & \text{,both conditions exists at the same time;} \\ a \times b \times d & \text{, no edge particles are discarded.} \end{cases}$$

$$(8)$$

$$V_i = (a - 2D_i) \cdot (b - 2D_i) \cdot d \qquad\qquad (10)$$

**Response to Anonymous Referee #2**

*Original Referee comments are in italic*

manuscript text is indented, with added text underlined and

Our responses are in regular font.

Thank you very much for your thorough review and insightful comments on our manuscript. We appreciate the time and effort you have dedicated to evaluating our work and your constructive feedback. Your suggestions have been invaluable in helping us improve the quality and clarity of our manuscript. Below, you will find our point-by-point responses to your comments, along with the revisions made to the manuscript.

**Observation volume OV and effective OV**

*The effective OV is the intersection of the three individual OVs, which are 170mmx 125mm x 104mm. The intersection of the two OVs of Cam1 and Cam2 is a cuboid with the dimensions 104mm x 104mm x 125mm, i.e. it has the volume of 1352cm3. Intersecting this with the OV of Cam0 will result in the effective OV, which then must have a smaller volume. That means the volume that you provide with 1464cm3 is wrong. Alternatively, your description is wrong and the effective OV is not the intersection oif the individual OVs.*

Thank you for your comments. We apologize for our mistakes. Upon retesting, we found that the depth of field should indeed be 88mm, rather than 104mm, and the effective OV volume should indeed be 775 cm³, rather than 1464 cm³. We have promptly corrected this in the revised manuscript.

*Rephrase: "the observation volume (OV) ... is the interior rectangle of the observation volume of ..." - 'Interior rectangle' is wrong term to describe a volume - It is not clear to use 'observation volume' to describe 'observation volume'*

Thank you for your comments. The"interior rectangle" is indeed a misnomer. We have removed these misrepresentations.

*Rephrase: "The cylindrical observation volume of the three telecentric lenses and LED lighting beams of 3D-PPI is illustrated in Fig. 2." - Does this refer to effective OV? - Fig 2 does not show any observation volume (shows beams) - No observation volume is cylindrical*

It doesn't refer to effective OV. Fig. 2 shows is the beams of cylindrical lens rather than effective OV.

We have revised this part to make it clearer in the revised manuscript Line 105 and 110:

The cylindrical observation volume of the three telecentric lenses and LED lighting beams of 3D-PPI is illustrated in Fig. 2. To clarify, the three dimensions of observation volume (OV) of one high-resolution camera is  a × b × d (170mm × 125mm × 88mm), which represent the length, width, and depth of field of view respectively.

Figure 2. The two views of  three telecentric lenses and LED light beams.

**Optical resolution / resolving power / smallest resolved detail**

*Thank you for adopting a better terminology. Pixel resolution and resolution (number of pixels of sensor) of the different cameras are important parameters in describing your instrument. I am missing another paramter describing the optical performance of your cameras, that is the resolving power achieved on images of Cam0-2 and on images of Cam3. You could determine it by calibration or by identifying the smallest features that can be detected on your images. I do not agree with a statement that the "smallest recognizable detail … is 0.0416 mm" (in response to my point 9). Of course, this is theoretically the best limit, however, it is not backed up by evidence. Can you show images of snow particles with small details to illustrate the smallest features that can be detected (under optimal illumination and in-focus position)? You seem to focus on "larger particles" (response to my point 9), but you don't mention this clearly in the manuscript.*

Thank you for your comments. We apologize that 'smallest recognizable detail can be detected' was an inaccurate description in the previous reply. Actually, given that the 3D-PPI uses telecentric lenses, it is approximated that all particles within the depth of field (88mm) are well illuminated and clearly imaged. In our manuscipt, we don't take into account the concepts of optical resolution, resolving power and smallest resolved detailed that you mentioned (we know they have different meanings from pixel resolution). The 0.0416 mm just refers to the pixel resoltion indeed the theoretical limit based on the camera's specifications. i.e. a single pixel representing the actual length of 0.0416 mm. The following figure illustrate this point.

Thank you for your understanding!

[Figure]

**Figure 1**. Evidence of pixel resolution

**Removal of noise and smaller particles**

*You are rejecting features with Dmax less than about 0.2mm. I would recommend stating this limit in the Abstract/Conclusions (something like "...measure particles > 0.2 mm").*

Thank you for your advice. We have added that our instrument is designed to measure particles with $D_{max}$ greater than 0.2 mm in Conclusion:

> The high-resolution cameras feature a pixel resolution of 41.5 µm·px-1 and are precisely synchronized by clock control which is sufficient to obtain fine shapes of snowflakes larger than 0.2mm, and the large field of view of 170 mm×125 mm enables it to capture enough snowflakes to estimate PSD accurately.

*You call it "removal of small noises from image". I would rephrase this as I don't consider 'small noises' a proper term, it sounds colloquial.*

Thank you for your advice. We have replaced "small noises " with "pixel noise (no larger than 20 pixels)" to avoid any colloquial implications.

*Prior to rejecting small features, you join connected regions that have their centroids separated by less than 4mm. This still fells like a large distance to me. I don't understand why the centroid distance has been chosen here rather than the actual gap separating any two connected regions. The centeroid distnce depends on the size of the connected regions whereas the gap represents the size of a potentially undetected region of the particle. When joining small connected regions you accept a gap of almost 4mm. For larger connected regions, the accepted gap is smaller by roughly the average size of these connected regions. It would be good, after justifying better your joice of criterion, to show examples of particles resulting from joining connected*

*regions. This could be done in Fig.7 by adding two or three such examples including a length scale for reference. A propoer description of what is shown now in the figure is missing (you indicate (in red numbers) detected particles, I think).*

Thank you for your advice. We have also reconsidered our image processing algorithms to joining connected regions. Using centroid distance as a criteria for judgement does bring these problems you mentioned. Instead of using the centroid distance, we have now adopted the distance between the closest points of connected regions as the standard criterion. Given the adoption of the new distance criterion, the minimum threshold for combining should be set at 2 mm rather than 4 mm. The revised Figure 7 now illustrates an example where several closely spaced connected regions are combined into one, and the 2 mm length scale has been added.

[Figure]

**Figure 2**. The revised Fig. 8 in the revised manuscipt.

We have revised this part in manuscipt:

Secondly, combine regions into a single particle when the  distance between the closest points of connected regions in a single image  is detected to be less than  2 mm apart. This step is necessary because a single particle may sometimes be perceived as two separate particles due to its position near the edge of the image processing threshold. Thirdly, discard the particles with an area smaller than 20 pixels (Equivalent to 0.035 mm2, Dmax is about 0.2mm), which enables the removal of  pixel noise (no larger than 20 pixels) from the image, to prevent there pixel noises from being mistakenly detected as small snowflakes.

**Pixel resolution**

*You determine the pixel resolution using ceramic spheres of known diameters, as described in Sect 3.2, which you call "Calibration of image binarization". I still find it unclear how you describe the process. You say that "we perform manual adjustments*

*to mitigate the software's misidentifification" sugesting that you use a software or algorithm. However, you state that the algorithm described in Sect 4 is not used. So, it is unclear what software or algorithm is used here. I think, for the purpose of determining the pixel resolution, it would have been better, and easier to describe, to do a completely manual analysis of the images of the ceramic spheres, for example by adding best-fitting circles and determining their diameters in pixels. I also still believe that taking images of a millimeter scale would have been better for the purpose. Aligning such a millimeter scale would not have been difficult with the required accuracy. Using a millimeter scale would have avoided the dependenc on the used algorithm and/or manual analysis (e.g. different thresholds result in over- or undersizing).*

Thank you for your comments. We apologize for our mistakes.

Firstly, As you say we just use ceramic spheres of known diameters to calculate a more accurate pixel resolution for each high-resolution camera. This process is not called "Calibration of image binarization", it is just an essential step before the "Calibration of image binarization". In the revised manuscript, we have placed this process in the added Sec. 3.2: **Estimation of pixel resolution**. And revised Sec. 3.3: **Calibration of image binarization**.

Secondly, actually, we don't use any software or algorithms, as you suggest, we process it entirely manually: taking the approach of counting the number of pixels that the diameter of the sphere occupies in the image.

Thirdly, using millimeter scale indeed a good choice as it comes with its own scale avoiding manual counting. However, the reason for not using the millimeter scale to obtain image resolution are as follows: pixel resolution can only be estimated when the millimeter scale plane is parallel to the camera image plane, which is difficult to guarantee. The advantage of the ceramic sphere in comparison is that its image taken at any angle is round.

*In fact, the use of spheres of know sizes is more suitable for testing an image processing algorithm. You actually do that in Sect 4.1 and Fig. 8. As a consequence, your Figures 6b) and 8 look almost identical, and it is difficult to see what difference there is between the two tests. In one figure you show error bars in the other not. What are the error bars in Fig 8? I guess they are related to that you image 20 spheres falling through in fornt of each camera. For Fig 6b, did you image one sphere for each diameter? Were these spheres dropped as for Fig 8? For Fig.6b, I would suggest to plot the distance of two points (in mm) over the distance of these two points on the image in pixels (rather than the opposite as you are doing). Then you end up with fitting your data to the linear expression of the form Dmax = d\*Px + e rather than, as you do now, Px = b\*Dmax + c. In these expressions Dmax represents the actual size or distance in mm and Px represents the measured distance on the image in pixels. The fitting coefficients are d and b, respectively, and the intercepts are e and c, respectively. The resulting value for d is the pixel resolution. Whereas you take the inverse of the slope b. It is, however not generally true that d = 1/b, where d and b result from fitting the same data to the above*

*two expressions. So, do the fit of Dmax = d\*Px + e to your data in Sect 3.2 and then report d as pixel resolution. You likley get very similar results, but you avoid the confusion of how to comparer or convert your results properly to pixel resolution.*

*Fig 6b from one sphere (for each size), or the average of several images of spheres (at each size)?*

Thank you for your comments.

Firstly, for Fig. 6b, we used these spheres to estimate the pixel resolution. For Fig. 8b (Fig. 7 of the revised manuscript), we still used these spheres to test the accuracy of the image binarization. These two figures look similar and both use the same spheres, but for different purposes.

Secondly, for Fig. 6b, only one set of measurements was taken for each sphere diameter to fit the slope of the line. In contrast, for Fig. 8b (Fig. 7 of the revised manuscript) shows that each sphere was measured 20 times, with the average of those measurements represented as the center point of the error bars for each diameter. The length of the error bars indicates the uncertainty, expressed as the standard deviation. For Fig 6b, we only image one sphere for each diameter. These spheres indeed dropped as for Fig 8.

Finally, thank you for your advice to switch the horizontal and vertical coordinates of Fig. 6b. We have revised our manuscipt as your suggestions.

**Image processing algorithm – resulting errors**

*You use adaptive thresholding to binarize images. You also state (in the response to my point 14d) that "It has been tested that particles outside the depth of field cannot be clearly imaged and therefore cannot be detected." What does this statement mean? Are particles outside the DOF not properly binarized (because they are fuzzy, not in focus)? My question remains: How well-defined is the depth of field? The PSD values and OVs depend on it. You state that "The average error for all spheres across different diameters is -0.048 mm." In your responses you explain that this error is "measured-true" and can be positive or negative. Taking the average of two errors where one is, for example, +5 and the other -5 would result in a zero average, which would wrongly describe the error. I think this is what happened when you report the very small "average error", it is a misleadingly small value, the average of larger positive and negative values. Consider another way to describe the errors and relative errors.*

Thank you for your comments.

Firstly, we have conducted laboratory tests to determine the depth of field (DOF) of our imaging system, shown in Figure 3. We used the Modulation Transfer Function (MTF), which measures the system's ability to resolve contrast at different spatial frequencies. Specifically, we consider the range where the MTF value exceeds 50% of its maximum as the effective depth of field (shown in Figure 4, the DOF is 88mm). As such, any

snowflakes with MTF values below this threshold are deemed fuzzy and considered outside the depth of field. Therefore, our algorithm effectively excludes these particles from identification.

[Figure]

**Figure 3**. The DOF test scene.

[Figure]

**Figure 4**. The MTF plot.

Secondly, as you mentioned, averaging the positive and negative errors can lead to misleadingly small values. To provide a clearer and more intuitive presentation of the errors, we have calculated the mean relative error (MSE) for $D_{max}$ and $D_{eq}$ and plotted the relative error bars in Fig. 7 of the revised manuscript, shown in Figure 5

Regarding the Dmax measurement results (Fig. 7c, e), smaller spheres (9 mm and below) tend to show that the measurements are slightly greater than the true values, while larger particles exhibit that the measurements are slightly lower than the true values. The maximum relative error is about 14%, and the MRE of Dmax is about 4%. As for Deq measurement results (Fig. 7d, f), almost all diameter measurements underestimate the true values. The maximum relative error is about -7% and the

MRE of Deq is about 3%. Since the measurement errors of Deq for all spheres are lower than the true values, they can be utilized for systematic error correction.

[Figure]

**Figure 5**. The revised Fig. 7 in the revised manuscipt.

**Horizontal speed**

*"The average value (consider positive and negative) of the horizontal velocity component…" What does that mean? Is this a suitable average (see also error of image processing above).*

The statement means that the average horizontal velocity component (+0.41m/s), including both positive and negative values, represents the net directional movement of the particles. In my personal view, this average calculation is appropriate as it provides a direct measure of the overall trend in particle motion, which is essential for our analysis.

*It is not obviously clear that the horizontal speed corresponds to the East-West wind direction. Mention that installing the instrument facing south means that the horizontal speed seen by the high-speed camera Cam3 corrsiponds to East-West.*

Thank you for your advice, we have mentioned this at the beginning of this paragraph:

Installing the instrument facing south means that the horizontal velocity seen by the high-speed camera Cam3 corresponds to East-West. The average value  of the horizontal velocity component measured by 3D-PPI is +0.41m/s , and the standard deviation is 0.73m/s (Fig. 16a). The overall distribution of particle horizontal velocities ranges between ±3m/s, and more than

70% of the snowflakes have a horizontal velocity distribution between $\pm 1$m/s. Positive velocities predominate over negative ones, largely influenced by the prevailing westward winds.

**Size-dependent OV in Eq. 7**

*Excluding particles at the edge of the image means a reduction of OV that depends on particle size. You account for that in the modified eq (7). It now features Vj, the valid OV at the "jth moment". Vj has already been used in eq (6) for speed, potentially leading to confusion. However, what confuses me more is this "jth moment", not sure what that means. Should that not be "in the jth image"?*

Thank you for your comments. We apologize for our mistakes. $V_j$ here does create confusion with the representation of velocity. And it should be 'in the jth image' rather than 'jth moment'.We have revised Eq. 7 and 8, and all related text, to eliminate the use of " $V_j$ " and "the $j$th moment," which should now prevent any ambiguity in the presentation of our methodology.

Because of the previous addition of equations, Eq. 7 and 8 in the original manuscript correspond to Eq. 9 and 10 in the revised manuscript, as follows:

The time-averaged PSD calculated from 3D-PPI counting over a specified period is as follows:

[revised manuscript text omitted]

Where n denotes the number of spheres of different diameters, which is 15; $D_{ai}$ denotes the $i^{\text{th}}$ diameter of the sphere;

$\overline{D}_{\max i}$ and $\overline{D}_{eqi}$ denote the average value of $D_{\max}$ and $D_{eq}$ for 20 times for $i^{\text{th}}$ diameter respectively.

With $C$ set to 0.4, the 15 spheres in Fig. 6a were effectively binarized, as shown in Fig. 7b. Regarding the $D_{\max}$ measurement results (Fig. 7c, e), smaller spheres (9 mm and below) tend to show that the measurements are slightly greater than the true values, while larger particles exhibit that the measurements are slightly lower than the true values. The maximum relative error is about 14%, and the MRE of $D_{\max}$ is about 4%. As for $D_{eq}$ measurement results (Fig. 7d, f), almost all diameter measurements underestimate the true values. The maximum relative error is about -7% and the MRE of $D_{eq}$ is about 3%. Since the measurement errors of $D_{eq}$ for all spheres are lower than the true values, they can be utilized for systematic error correction.

[Figure]

**Figure 7 (a)** Variation of MRE with sensitivity coefficient $C$; **(b)** The binarized images of 15 spheres; The average values of measurements of $D_{\max}$; **(c) - (f)** The average values and relative errors of 20 $D_{\max}$ and $D_{eq}$ measurements for per diameter sphere.

---

## Author Response (AR3)

**An introduction of Three-Dimensional Precipitation Particles Imager (3D-PPI)**

**Response to the reviewers**

Jiayi Shi, Xichuan Liu, Lei Liu, Liying Liu, Peng Wang

Original Referee comments are in italic

manuscript text is indented, with added text underlined and <del>removed text crossed</del> <del>out.</del>

Our responses are in regular font.

Thank you very much for your thorough review and insightful comments on our manuscript. We appreciate the time and effort you have dedicated to evaluating our work and your constructive feedback. Your suggestions have been invaluable in helping us improve the quality and clarity of our manuscript. Below, you will find our pointby-point responses to your comments, along with the revisions made to the manuscript.

**Easily fixed issues:**

**Pu et al 2021**

The new reference to Pu et al. 2021 (L29) is about snow darkening and melting as a consequence of it. As such it is not suitable as reference for "importance of snowflake shapes for our understanding of atmospheric science.

Thank you for your advice. We have removed this reference in the revised manuscript.

**Fig 2**

The beams extending from the four cameras in Fig 2 are confusing. They are not mentioned neither in the sentence (L104) referring to Fig 2, nor in the caption of Fig 2. Fig 2 shows also the high-speed camera Cam3, apparently with a conical beam extending from it. That camera is not mentioned in L104. For clarity, I would add labels in Fig 2 pointing to the three telecentric lenses, the high-speed camera or lens, and the LED light beams. If you still want to keep the additional beams extending from the lenses, then I would mention them somehow in the caption.

Thank you for your advice. We have added labels in Fig 2 and also referred to them in the caption and in L104.

The three telecentric lenses and LED lighting beams of 3D-PPI are illustrated in Fig. 2.

The four cameras, lenses, and LED lights, including the additional beams of 3D-PPI, are illustrated in Fig. 2.

**Figure 2. The two views of three telecentric lenses and LED light beams.**

Figure 2. The two views of four cameras and lenses, including the additional beams extending from the lenses and the LED light beams.

**Three dimensions of OV**

The sentence in L104-106 has several issues and should be rephrased. The "three dimensions ... is a x b x d" is grammatically and mathematically wrong. The three dimensions are a, b, and d, not a x b x d. The previous issue is better fixed with using the appropriate term instead of "interior rectangle". I think "cuboid" would be correct here (OV of one camera is a cuboid defined by the three dimensions ...)

Thank you for your advice. We have revised this part:

To clarify, the three dimensions of observation volume (OV) of one high resolution camera is a  $\times$  b  $\times$  d (170mm  $\times$  125mm  $\times$  88mm), which represent the length, width, and depth of field of view respectively.

To clarify, the observation volume (OV) of one high-resolution camera is a cuboid defined by the three dimensions: a, b, and d (170mm, 125mm and 88mm), which represent the length, width, and depth of field of view respectively.

**Da vs Dp**

Thank you for adopting this way of showing results of ceramic spheres. Remove sentence L209-210 which doesn't apply anymore.

Thank you for your advice. We have removed this sentence in the revised manuscript.

**Size-dependent OV**

The new Eq (10) to calculate the effective OV as a function of snowflake size seems wrong to me. At each border or edge only D/2 needs to be removed. If a particle of size D is at least D/2 away from the border (distance between particle centre and border), then it will not touch it. So: Vi = (a-Di)(b-Di)\*d

Thank you for your comments, we have modified the formula as you suggested.

**Further discussions**

**REVISED** Sec. 3.2 Estimation of pixel resolution (previously Calibration of image binarization) and NEW Sec. 3.3 Calibration of

**image binarization**

For me the estimation of pixel resolution comes too early (before image processing) and requires image binarization (so that that is done twice). Image binarization is optimized, not calibrated. In your response you try to justify why spherical targets are better than planar micrometer scales, where there would be an issue with orienting them perfectly. The disadvantage of the spherical targets is the necessity to do a manual binarization in addition to the adaptive thresholding. The apparent size of spherical objects depends on any binarization. I believe that this is introduces a larger uncertainty than what would be related to imperfect alignment of a micrometer slide. Some more detailed comments below:

Image processing is described in separate sections (3 and 4): The image binarization (now described in the new Sect 3.3) is for me not a "calibration" as the title of the subsection suggests (but the determination of optimal binarization). I would consider image binarization as one of the steps in image processing. Subsection 4.1 is now called "Image processing" and describes the steps of noise removal and segmentation. It would be a clearer structure to first describe the complete image processing and only then results (pixel resolution) and algorithms that use this image processing. Sect 3.3 should be part of image processing (Sect 4.1).

For doing the above, you would need to do the binarization only once (with adaptive thresholding). Now it is done twice: In Sect. 3.2, Dp is "counted manually" to determine pixel resolution, which means that a manual binarization is done. Then, in Sect 3.3, binarization is done again with adaptive thresholding. The "optimal" sensitivity coefficient c is effectively the coefficient where adaptive thresholding approximates closest the manual binarization from Sect 3.2. (It is understandable and

fine that you will not change from your spheres to anything else now. But consider the following:

It would be more transparent to find "manually" the best sensitivity coefficient c (i.e. decide visually what looks to be a good binarization) and make this part of the description of image processing. Then the estimation of pixel resolution could be done with images processed according to that image processing. That would also avoid having twice similar analyses (related to 6b and 7c/d).

Thank you for your comments and advice.

Firstly, we have removed the section "3.2 pixel resolution estimation" for the following reasons. We have completed an accurate estimation of the pixel resolution by shooting standard calibration plates including micrometer slide (shown in Figure 1), which yields a value of  $41.6 \mu m \cdot px^{-1}$ . As you say, estimating camera resolution using spheres relies on image binarization and manual processing, and the estimated pixel resolution will not necessarily be more accurate than the  $41.6 \mu m \cdot px^{-1}$  given by the manufacturer.

Secondly, we removed the section "3.3 calibration of image binarization " and the content about image binarization were added to section "4.1 Image Processing".

Thirdly, we photographed the ceramic spheres after image processing. Further image processing such as binarization was performed and the errors was calculated.

Fourthly, the previous method of determining the sensitivity coefficient C seems to be a bit complicated and not very meaningful. It is indeed a simple and effective method to find the optimal C through visual assessment. We have revised this part as your advice:

The sensitivity coefficient C is crucial; it adjusts how the local mean is used to set the threshold. A smaller C favors classifying pixels as foreground, while a larger Cfavors background classification. We have manually adjusted the sensitivity coefficient C to determine the optimal value of 0.4. This process involved visually assessing the binarization outcomes for various C values to identify which value best distinguishes between foreground objects and background.

Figure 1. The images of standard calibration plates.

**MTF method to determine the depth of field**

I appreciate the details on how you determined the depth of field and that you checked and corrected it.

There are two things I didn't understand though:

• What are "different spatial frequencies"? This is probably a minor thing or me not understanding.

Thank you for your comments. To clarify, spatial frequency describes the rate that a stimulus changes across space. For images recorded in this manuscript, high - frequency components in an image refers to the sharp edges, fine textures and other detailed information, while low - frequency components refer to large, smooth areas and overall contours.

When using the Modulation Transfer Function (MTF) to determine the depth of field (DOF), we consider different spatial frequencies because a lens's imaging ability for different levels of detail varies with the object distance. Higher spatial frequencies reflect the lens's ability to resolve fine details, and lower spatial frequencies represent its performance in depicting large - scale features.

Reference:

(https://evidentscientific.com/en/microscope-resource/knowledge-hub/anatomy/mtfintro) Ashoor, M. and Khorshidi, A.: Modeling modulation transfer function based on analytical functions in imaging systems, The European Physical Journal Plus, 138, https://doi.org/10.1140/epjp/s13360-023-03884-8, 2023.

• How are snowflakes below the MTF threshold "deemed fuzzy and considered outside the depth of field"? You claim that, consequently, your "algorithm effectively excludes these particles from identification". How does this work? Snowflakes are

**detected/identified if they are binarized as a connected region of more than 20 pixels. What is the relation to the MTF threshold?**

Thank you for your comments. In our previous response, we mentioned that the depth of field of the lens was determined by calculating the MTF function. Due to its high accuracy, this MTF method is commonly employed by lens manufacturers to determine the depth of field. MTF is time-consuming and not suitable for rapidly processing a large number of snowflake images to exclude those deemed fuzzy that are out of focus. In the revised manuscript, we have explained how we exclude deemed fuzzy snowflakes outside the depth of field by calculating the variance of the Laplacian:

Secondly, combine regions into a single region of interest (ROI) when the distance between the closest points of connected regions in a single image is detected to be less than 0.5 mm apart. This step is necessary because a single particle may sometimes be perceived as two separate particles due to its position near the edge of the image processing threshold. Thirdly, discard the blurred particles outside the depth of field. To avoid detecting the particles completely out of focus, in the greyscale image before binarization, the mean grey value of the ROI region must be at least 20 greater than the mean grey value of the image and the variance of the Laplacian of the ROI grey value must be at least 10. Fourthly, discard the particles at the edge of the image. If the connected region of a particle contains points located at the edges of the image, the particle is considered not to be fully captured, and it should be discarded.

**Description of image binarization Sec. 3.3**

I appreciate the added details about the binarization method (adaptive thresholding). However, I have some questions and see a few issues:

• Could you cite some description of this method in the literature? Without that I think that I need some more information.

Thank you for your advice. We apologize for our oversight. We have added literature in the revised manuscript.

Bataineh, B., Abdullah, S. N. H. S., and Omar, K.: An adaptive local binarization method

for document images based on a novel thresholding method and dynamic windows, Pattern

Recognit. Lett., 32, 1805-1813, https://doi.org/10.1016/j.patrec.2011.08.001, 2011.

• L221 and Eq (6): what is the "local mean  $\mu(u,v)$ "? Mean of what? How is it calculated? What is the specified neighborhood? How is that adjusted by the sensitivity coefficient C?

Thank you for your comments. The local mean  $\mu(u,v)$  refers to the average brightness of pixels within a specified neighborhood around the pixel (u,v). This neighborhood can be defined in various ways, such as a fixed-size window or a dynamically sized region based on the local image characteristics. The specific method of calculation depends on the implementation, but typically involves averaging the brightness values of the pixels within the chosen neighborhood.

• *MRE* defined by Eq (7): this seems to be the average of the two means of the absolute relative errors. Being based on absolute values it is always positive.

Later you refer to the "MRE of Dmax" (L243) and "MRE of Deq" (L244). According to Eq (7) there is only one MRE, which is based on both Dmax and Deq.

Then, you also refer to "relative errors" of Dmax and Deq. These are not absolute values but positive or negative. From Fig 7 I assume that they are determined as (Dmaxi-Dai)/Dai, which is different from what Eq (7) would suggest ((Dai-Dmaxi)/Dai).

If the worst relative error is -7% then it is strictly speaking wrong to call that the "maximum relative error", which would be +2% in case of Deq (Fig 7f).

Thank you for your comments. We apologize for the confusion caused by the addition of Eq (7) in the last response. We have removed the Eq (7) in the revised manuscript. Therefore, in the revised manuscript there is no MRE, only "relative errors", which are not absolute values but positive or negative.

Thank you for your advice. The "maximum error" is indeed inappropriate and we have changed it to "worst error".

• The definition of Dmax (L231) is different from the definition of Dmax later used in Sect. 5.2 (L397). The definition in L397 is the one I would expect here. I would call Dmax "maximum dimension" not "maximum size" as doen in Sect 5.2.

Thank you for your comments. We apologize for our mistakes. To avoid redundancy, we have revised the notation in L397 by renaming  $D_{\text{max}}$  to  $DV_{\text{max}}$ . The corresponding content in Fig. 14 has also been updated accordingly. Additionally, we have revised the definitions of both variables, as detailed below:

L231: the  $D_{\text{max}}$  is the distance between the two farthest points of the particle profile

the diameter of the smallest enclosing circle

maximum dimension  $DV_{max}$  (distance between the two farthest points on the surface of the particle)

**Pixel noise (L274-275)**

Referring to detected regions with less than 20 connected pixels as "pixel noise (no larger than 20 pixels)" is now clearer than "small noises" previously. It implies that all regions with less than 20 connected pixels are indeed noise, i.e. not related to actual snowflakes. I am not sure this is true in general. Could these "noise" features be caused

by snowflakes that are outside the depth of field, or by small snowflakes that are too small to be detected by 3D-PPI? So, rather than and/or in addition to "prevent these noises from being mistakenly detected as small snowflakes" it should say "exclude features of small snowflakes that cannot be detected from analysis"?

Thank you for your comments. As you pointed out, these so - called "pixel noises" could indeed be small snowflakes or snowflakes outside the depth of field (We have already mentioned the snowflakes outside the depth of field in the revised manuscript.) Given their small size, it is extremely difficult to extract the features of these tiny regions. Thus, we excluded them from the analysis. We have revised the sentence as your advice:

Thirdly, discard the particles with an area smaller than 20 pixels (Equivalent to 0.035 mm2,  $D_{max}$  is about 0.2mm), which enables the removal of pixel noise or small snowflakes (no larger than 20 pixels) from the image, to prevent these noises from being mistakenly detected as small snowflakes exclude features of small snowflakes that cannot be detected from analysis.

**2-mm gap criterion**

Your new criterion for joining regions is better than the previous one. It, however, still allows that small regions would be joined across a gap that can be larger than these regions. A 20-pixel region has about 14 to 5 pixels across, and a 2-mm gap corresponds to almost 50 pixels. I.e., two 5-pixel regions could be joined even if they are separated by a about ten times larger gap. I think two such regions should rather be excluded. Would they indeed be belonging to the same snowflake, then that would mean that a large part of this snowflake would have been missed (not been detected by binarization) likely due to being out of depth of field.

Thank you for your comments. We fully recognize that our previous criterion for joining regions might not be reasonable, which could lead to the situation that you described. After careful consideration, we have raised the standard for combining. Now, only connected regions with an area larger than 20 pixels and a distance of less than 0.5 mm between their closest points will be combined. As a result, we have reordered the sentences in the manuscript by first discarding connected regions with an area larger than 20 pixels, and then combine regions into a single particle when the distance between the closest points of connected regions in a single image is detected to be less than 0.5 mm apart. This adjustment allows us to preserve only the shape features of the main part of the particles and effectively avoids the problem that you mentioned. We have revised the sentence in the manuscript:

(ii) Particle detection. Firstly, detect the connected regions in binarized images. Secondly, combine regions into a single particle when the distance between the closest points of connected regions with an area larger than 20 pixels in a single image is detected to be less than 0.5 mm apart. This step is necessary because a single particle may sometimes be perceived as two separate particles due to its position near the edge of the image processing threshold. Thirdly, discard the particles with an area smaller than 20 pixels (Equivalent to 0.035 mm2, Dmax is about 0.2mm), which enables the removal of pixel noise or small snowflakes (no larger than 20 pixels) from the image, to exclude features of small snowflakes that cannot be detected from analysis.

(ii) Particle detection. Firstly, in binarized images, detect the connected regions with an area larger than 20 pixels (Equivalent to 0.035 mm2, Dmax is about 0.2mm), which enables the removal of pixel noise or small snowflakes (no larger than 20 pixels) from the image, to exclude features of small snowflakes that cannot be detected from the analysis. Secondly, combine regions into a single region of interest (ROI) when the distance between the closest points of connected regions in a single image is detected to be less than 0.5 mm apart. This step is necessary because a single particle may sometimes be perceived as two separate particles due to its position near the edge of the image processing threshold. This method enhances the accuracy of foreground particle detection, particularly in images with complex backgrounds and uneven illumination. Thirdly, discard the blurred particles outside the depth of field. To avoid detecting the particles completely out of focus, in the greyscale image before binarization, the mean grey value of the ROI region must be at least 20 greater than the mean grey value of the image and the variance of the Laplacian of the ROI grey value must be at least 10.

---

## Author Response (AR4)

**An introduction of Three-Dimensional Precipitation Particles Imager (3D-PPI)**

**Response to the reviewers**

Jiayi Shi, Xichuan Liu, Lei Liu, Liying Liu, Peng Wang

Original Referee comments are in italic

manuscript text is indented, with added text underlined and <del>removed text crossed</del> <del>out.</del>

Our responses are in regular font.

Thank you very much for your thorough review and insightful comments on our manuscript. We appreciate the time and effort you have dedicated to evaluating our work and your constructive feedback. Your suggestions have been invaluable in helping us improve the quality and clarity of our manuscript. Below, you will find our pointby-point responses to your comments, along with the revisions made to the manuscript.

1. In subsection 4.1 Image processing, all has improved and is fine with the exception of how you seem to do normal arithmetic means of relative errors that are positive and negative. I find it odd to do that, since you could have cases where half of the errors are +5% and half -5%, in which the average would be 0% (could lead to think that on average there is no error). For Dmax the average is, you say +2.2%. If I would determine the average of the absolute values of all Dmax errors it would be close to 5%.

Thank you for your recognition. We have revised the text to clarify that while the arithmetic mean of  $D_{\text{max}}$  relative errors is +2.2%, the average absolute relative error is 5.0%, indicating a systematic overestimation tendency. This distinction ensures readers understand both the net bias and the overall error spread.

2. Still in subsection 4.1 and still discussing errors in Fig 7, you say that "the measurement errors of Deq for all spheres are lower than the true values ...". Did you want to compare measurements (rather than "measurement errors") to true values? In that case, it would be almost all (rather than "all", there is one exception). Maybe there is something else here I didn't understand.

Thank you for your advice. We apologize for our mistake. The expression "all" is wrong, and there is indeed an exception here. The revised text now states: "nearly all estimates are below the true values, with a single exception at 10 mm (+0.8%)", which aligns with the data in Fig. 7d.

What follows is lines 233-239 in the revised manuscript, which should address your first two comments:

Regarding the  $D_{\text{max}}$  measurement results (Fig. 7c, e), smaller spheres ( $\leq 9 \text{ mm}$ ) exhibit slight overestimations of the true values, while larger particles show underestimations. The maximum relative error is approximately 14%. The arithmetic mean of relative errors across all diameters is +2.2%, though the average absolute relative error (i.e., magnitude regardless of sign) is 5.0%, reflecting a systematic overestimation tendency. For  $D_{eq}$  measurements (Fig. 7d, f), nearly all estimates are below the true values, with a single exception at 10 mm (+0.8%). The worst relative error is -7%, and the arithmetic mean of relative errors is -2.7%. The consistent underestimation of  $D_{eq}$  (except for the 10 mm case) suggests its utility for systematic error correction. Overall, the image processing methods demonstrate effectiveness, with errors remaining minimal in practical terms.

3. In Section 5, you have introduced DVmax to denominate the distance between the two farthest points on the surface of the particle. I cannot guess why you called it DVmax. It is also fine that you use something different from Dmax here, even if I don't see the motivation. Regardeless, If you want to keep this DVmax, then update the Figure that follows with examples listing particle properties (I guess Dmax in that figure should now be DVmax).

Thank you for your advice. We apologize for the lack of clarity. The  $DV_{\text{max}}$  (short for "Dimensional Maximum in Volume") is defined as the maximum distance between any two surface points of the 3D-reconstructed particle. This terminology distinguishes it from the 2D  $D_{\text{max}}$  (smallest enclosing circle diameter). To clarify, in the manuscript, " $DV_{\text{max}}$ " is only used in Section 5.2, while " $D_{\text{max}}$ " is used in all other parts of the manuscript. We have checked and confirmed that " $DV_{\text{max}}$ " is consistently used in all examples in Figure 13. The specific revisions are as follows:

To characterize the 3D shape of each snowflake, four parameters are calculated: volume V, dimensional maximum dimension in volume  $DV_{max}$  (distance between the two farthest points on the surface of the 3D-reconstructed particle).